# OPEN

# CDK1–cyclin-B1-induced kindlin degradation drives focal adhesion disassembly at mitotic entry

Nan-Peng Chen ⬤ ✉, Jonas Aretz and Reinhard Fässler ⬤ ✉

**The disassembly of integrin-containing focal adhesions (FAs) at mitotic entry is essential for cell rounding, mitotic retraction fibre formation, bipolar spindle positioning and chromosome segregation. The mechanism that drives FA disassembly at mitotic entry is unknown. Here, we show that the CDK1–cyclin B1 complex phosphorylates the integrin activator kindlin, which results in the recruitment of the cullin 9–FBXL10 ubiquitin ligase complex that mediates kindlin ubiquitination and degradation. This molecular pathway is essential for FA disassembly and cell rounding, as phospho-inhibitory mutations of the CDK1 motif prevent kindlin degradation, FA disassembly and mitotic cell rounding. Conversely, phospho-mimetic mutations promote kindlin degradation in interphase, accelerate mitotic cell rounding and impair mitotic retraction fibre formation. Despite the opposing effects on kindlin stability, both types of mutations cause severe mitotic spindle defects, apoptosis and aneuploidy. Thus, the exquisite regulation of kindlin levels at mitotic entry is essential for cells to progress accurately through mitosis.**

Adherent animal cells dramatically change morphology at mitotic entry. In interphase, cells are flat and firmly adhere to the extracellular matrix (ECM) substrate via integrin-containing FAs. With the activation of the cyclin-dependent kinase 1 (CDK1)–cyclin B1 complex at mitotic entry[1–3], cells isochronally undergo a series of morphological events, which commence with the disassembly and shrinkage of FAs that enable the retraction of the cell margins and the formation of mitotic retraction fibres. These events are completed by the rounding up of the cell body whose near spherical shape provides the space and geometry required for mitotic spindle assembly, positioning and function[4–14]. Inhibition of FA disassembly prevents mitotic cell rounding and impedes the attachment of the bipolar spindle to the cell cortex for the accurate capture and segregation of chromosomes[8,15]. The mechanism underlying mitotic cell rounding has been deciphered as an interplay between cortical actomyosin contractility and osmotic outward pressure[7]; however, the mechanism underlying FA disassembly and remodelling remains enigmatic. Interestingly, CDK1 activity has been linked to FA remodelling in S and G2 phases. In S phase, CDK1 in complex with cyclin A2 increases FA size and stress fibre formation by phosphorylating formin-like protein 2 (FMNL2) and probably other FA and actin-associated proteins[15,16]. By contrast, in G2 phase, CDK1 is inactivated, which leads to a decrease in FA size[15]. Since inhibition of the CDK1–cyclin B1 complex blocks the entire process of mitosis, it is impossible to define with loss-of-function experiments whether activation of the CDK1–cyclin B1 complex directly induces FA disassembly at mitotic entry or whether this function is mediated by other, M-phase-dependent and CDK1-independent signalling pathways.

Integrins assemble different adhesion sites that attach cells to the ECM[17–19]. During cell spreading, integrins cluster in small nascent adhesions at cell margins that can mature into stable, large and stress-fibre-associated FAs[20–22]. A hallmark of integrins is that ligand binding requires an integrin-activation step that is induced by the cytoplasmic adaptor proteins talin and kindlin[22–28]. Disruption of talin or kindlin binding to integrins induces rapid FA disassembly and curbs cell-substrate adhesion, essentially as observed at mitotic entry. In this context, it was shown that talin and other canonical

FA components, including paxillin and zyxin, are absent from activated integrin-containing adhesion sites of mitotic retraction fibres and cell bodies[14,29]. Although these observations challenge the current model of integrin activation and suggest that cells use different mechanisms to activate integrins and anchor to ECM substrates in interphase and mitosis, the absence of talin from mitotic adhesions could potentially be accomplished by downregulation of Rap1 GTPase activity[30], which facilitates talin binding to and activation of integrins[31]. However, the normal development of mice that lack the expression of both Rap1 isoforms (Rap1A and Rap1B) up to mid-gestation[32] indicates that Rap1 per se is dispensable for mitotic cell rounding.

Kindlins, the loss of expression of which has been associated with mitotic spindle defects in the human cancer-prone disease Kindler syndrome[33,34], consist of three cell-type-specific isoforms[35–37]. Kindlin-1 (KIND1; encoded by *FERMT1*) is predominantly expressed in epithelial cells[28,35], whereas kindlin-2 (KIND2; encoded by *FERMT2*) is expressed in almost all cells except haematopoietic cells[27,35], which exclusively express kindlin-3 (KIND3; encoded by *FERMT3*)[26,35]. In this study, we examine the abundance of core components of integrin adhesion sites during the cell cycle and show that KIND1 and KIND2 are almost entirely degraded at mitotic entry followed by the disassembly or shrinkage of interphase FAs. The remaining, non-degraded kindlins localize together with talin and several canonical FA proteins to small, activated integrin-containing adhesion sites in mitotic retraction fibres to secure substrate attachment, proper spindle positioning and accurate chromosome segregation in dividing cells.

## Results

**KIND1 and KIND2 are degraded at mitotic entry.** To identify core FA proteins that decrease in abundance during FA disassembly at mitotic entry (Supplementary Video 1), we measured protein levels of $\beta_1$ integrin, talin-1, KIND1, KIND2, vinculin, paxillin, integrin-linked kinase (ILK), PINCH and FAK in HeLa cells synchronized by double thymidine block (D-THY; Extended Data Fig. 1a). In mitosis (M phase; 9 h after release and indicated by phospho-histone 3), KIND1 and KIND2 underwent a mobility

Department of Molecular Medicine, Max Planck Institute of Biochemistry, Martinsried, Germany. ✉e-mail: nchen@biochem.mpg.de; faessler@biochem.mpg.de

shift to an apparently higher molecular weight and concomitantly decreased in abundance by ~80%, and returned to pre-mitotic levels in the subsequent G1 phase (13 h after release; Fig. 1a,b and Extended Data Fig. 1b). *FERMT2* mRNA levels remained unchanged in mitotic HeLa cells (Extended Data Fig. 1c). A mobility shift without change in protein level was observed for paxillin, whereas none of the other FA proteins that were measured changed their expression pattern during mitosis, including the integrin subunits on the cell surface, with the exception of $\beta_5$ integrin, which was slightly increased in M phase (Fig. 1a,b and Extended Data Fig. 1b,d). Endogenous KIND2 or overexpressed EGFP-tagged KIND2 protein also showed a mobility shift and had reduced levels in HAP1, HEK293T, HeLa, U2OS and RPE1 cells and in mouse fibroblasts arrested using *S*-trityl-L-cysteine (STLC) (Extended Data Fig. 1e–g). The reduction in KIND2 levels in STLC-arrested, mitotic HeLa cells occurred on surfaces coated with fibronectin (FN), vitronectin (VN) and fetal bovine serum, and after embedment in Matrigel only or mixed with collagen type I (Extended Data Fig. 1h,i). Furthermore, KIND1 and KIND2 protein levels were decreased in mitotic cell populations collected by shake-off of untreated cycling HeLa as well as U2OS cells (Extended Data Fig. 1j). Notably, STLC-arrested human erythroleukaemic K562 cells, which express KIND3, did not have upshifted or changed KIND3 levels (Extended Data Fig. 1k).

Next, we arrested HeLa cells in G2 by inhibiting CDK1 activity with RO3306 (ref. [38]). As soon as 5 min after RO3306 washout, the KIND2 protein upshift became visible, increased further with time and was accompanied by a large decrease in KIND2 protein levels (Fig. 1c). After mitotic exit (2 h after washout) KIND2 returned to pre-mitotic molecular weight and levels. Images of fixed D-THY-synchronized and still images of live-recorded RO3306-released HeLa cells expressing EGFP–KIND2, Lifeact–mRuby and histone 2B (H2B)–CFP revealed that KIND2 signals decreased and shifted from FAs in interphase to the cytosol in prophase and metaphase. Moreover, KIND2 signals increased and accumulated in lamellipodia in anaphase and returned to pre-mitotic levels and FAs in G1 again (Fig. 1d,e and Extended Data Fig. 1l).

**Non-degraded KIND2 localizes in mitotic adhesion sites.** To explore the subcellular fate of non-degraded KIND2 during M phase, we performed live-cell imaging of D-THY-synchronized HeLa cells expressing EGFP–KIND2-, Lifeact–mRuby or H2B–CFP seeded on FN using total internal reflection fluorescence (TIRF) microscopy. EGFP–KIND2 localized to F-actin stress-fibre-associated FAs in interphase. In M phase, KIND2 was found in puncta arranged like a string of pearls along the F-actin bundles in retraction fibres and in the cell body beneath the condensed chromosomes (Fig. 2a). YPet-tagged talin-1 showed the same localization as EGFP–KIND2 in mitotic HeLa cells (Fig. 2b).

Structured illumination microscopy (SIM) revealed that EGFP–KIND2 colocalized with the $\beta_1$ integrin activation-associated 12G10 epitope in FAs of interphase HeLa cells and in puncta along the F-actin bundles of retraction fibres and in the round mitotic cell body (Fig. 2c,d). The EGFP–KIND2 signal in retraction fibres was weak but clearly visible after enhancing the fluorescence intensity,

and it colocalized with paxillin, vinculin, the 12G10 epitope and the TS2/16 epitope (exposed on active and inactive $\beta_1$ integrins; Fig. 2c,d and Extended Data Fig. 2a–f). Immunostaining of HeLa cells confirmed that the low levels of endogenous KIND2 also localized to the mitotic cell body and puncta in retraction fibres (Fig. 2e). Notably, YPet–talin-1 also located to 12G10-positive and TS2/16-positive puncta of retraction fibres and the mitotic cell body (Extended Data Fig. 2g–j), whereas zyxin, the recruitment of which to FAs is force-dependent[39,40], was absent in mitotic adhesions of retraction fibres (Extended Data Fig. 2k,l). This observation was consistent with the absence of phospho-myosin light chain (pMLC) staining in retraction fibres (Extended Data Fig. 2m). Optical cell-sectioning using confocal microscopy (Fig. 2f) revealed 12G10-positive integrins on the surface and EGFP–KIND2 and YPet–talin-1 throughout the cytosol of STLC-treated HeLa cells seeded on FN (Fig. 2g). This observation, together with the inability of STLC-arrested mitotic HeLa cells collected by mechanical shake-off to re-adhere to ECM substrates (Fig. 2h), suggests that the small adhesions along mitotic retraction fibres serve as major mitotic cell–ECM attachment sites.

**CDK1–cyclin B1 mediates KIND1 and KIND2 phosphorylation at mitotic entry.** To test whether the KIND1 and KIND2 upshift in M phase was caused by phosphorylation, we treated STLC-arrested HeLa cell lysates with alkaline and lambda phosphatases, which almost completely reversed the KIND1 and KIND2 upshift (Extended Data Fig. 3a). Next, we compared the phosphoproteomes of STLC-arrested and untreated HeLa cells by mass spectrometry (MS). Among the >25,000 identified phosphorylation events, large increases in the phospho-Ser179 (pS179) of KIND1 and the phospho-Ser181 (pS181) of KIND2 were observed in M phase (Fig. 3a and Supplementary Table 1). Consistent with the mobility shift of paxillin in M phase (Fig. 1a), paxillin was also phosphorylated (at S106 and S109) after STLC arrest (Fig. 3a and Supplementary Table 1). MS of KIND1 and KIND2 immunoprecipitates from STLC-arrested HeLa cells confirmed the phosphorylation of KIND1-S179 and KIND2-S181 and revealed phosphorylation of closely adjacent serine residues (KIND1: SSphS[174]GphS[176]PVphS[179]PGLYSK; KIND2: GphS[175]GphS[177]IYphS[180]phS[181]PGLYSK), although to a weaker extent (Supplementary Table 2).

KIND2-S181 and KIND1-S179 are located at the disordered, flexible peptide loop of the F1 domain and are conserved in vertebrates (Fig. 3b). Notably, the corresponding serine residue is absent from KIND3, which explains why KIND3 levels remained unchanged in mitotic haematopoietic cells (Extended Data Fig. 1j). The sequence of the phospho-motif in KIND1 and KIND2 (Fig. 3b) pointed to a potential proline-directed CDK1 consensus motif[41,42]. Consistently, MS confirmed the occurrence of strong direct phosphorylation of recombinant KIND2 at S181 and to a lesser extent adjacent serine residues (GphS[175]GphS[177]IYphS[180]phS[181]PGLYphS[186]K) by the recombinant CDK1–cyclin B1 complex in vitro (Extended Data Fig. 3b and Supplementary Table 3). Moreover, immunoprecipitation of KIND2 in cell lysates revealed an association with CDK1 and cyclin B1 in lysates from STLC-arrested mitotic but not untreated HeLa cells (Fig. 3c). By contrast, cyclin A2 neither associated with EGFP–KIND2 in untreated nor STLC-arrested cells (Fig. 3c).

**Fig. 1 | KIND1 and KIND2 are degraded in HeLa cells at mitotic entry. a**, Western blot (WB) of core FA proteins in HeLa cells released from D-THY block and collected at the indicated times. **b**, Abundance relative to GAPDH and normalized to $t = 0$ of indicated FA proteins in D-THY-released HeLa cells determined by WB analysis ($n = 3$ independent experiments). See Extended Data Fig. 1b for details of quantifications. **c**, KIND2 levels in RO3306-synchronized HeLa cells collected by mitotic shake-off at the indicated times. **d**, Representative images recorded by wide-field fluorescence microscopy showing the localization and relative fluorescence of EGFP–KIND2, Lifeact–mRuby and H2B–CFP in different cell cycle phases of D-THY-synchronized HeLa cells. Arrowheads indicate lamellipodia in anaphase cells. Scale bar, 10 μm. **e**, Dynamics of EGFP–KIND2 fluorescence intensity normalized to $t = 0$ throughout mitosis in HeLa cells after RO3306 release recorded by live-cell imaging using wide-field fluorescence microscopy (mean ± s.d.; $n = 5$ cells). Nuclear envelope breakdown (NEBD) and metaphase-to-anaphase transition (MAT) are indicated. All WB experiments were repeated three times, with similar results obtained. Phospho-histone 3 (pH3) served as an indicator for mitosis and GAPDH as the loading control.

Next, we generated a polyclonal antibody against the pS181-motif of KIND2 (Extended Data Fig. 3c,d) that bound phosphorylated endogenous KIND2 and overexpressed EGFP-tagged wild-type KIND2 (EGFP–KIND2-WT) in mitotic HeLa cell lysates but not

EGFP–KIND2 carrying a S181 for alanine substitution (S181A) (Fig. 3d). Immunoblotting with this antibody indicated that KIND2-pS181 was absent in interphase HeLa cells (Fig. 3d), in mitotic HeLa cell lysates treated with alkaline and lambda phosphatases (Fig. 3e)

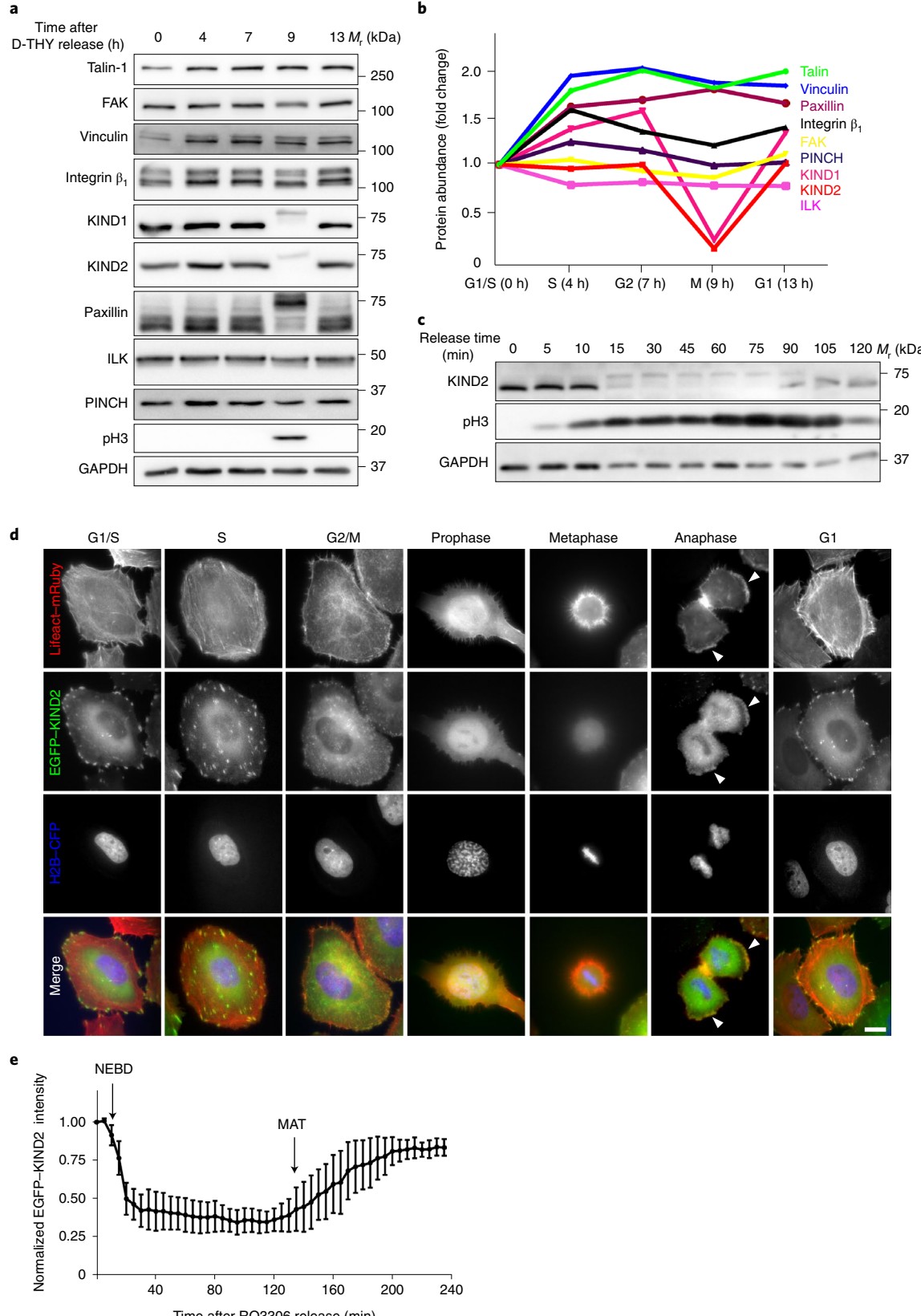

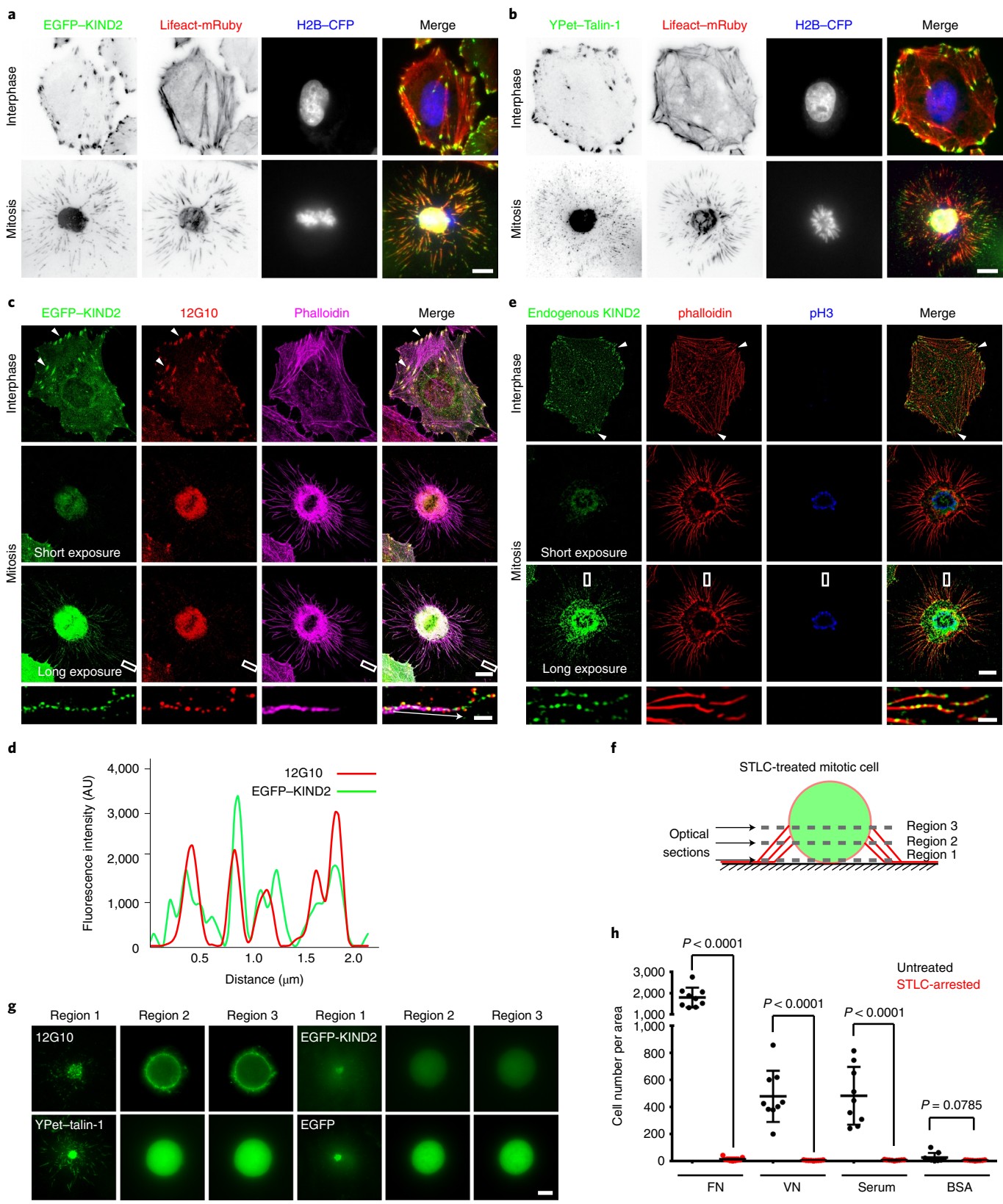

**d**

Fluorescence intensity plot with legend: 12G10 (red), EGFP–KIND2 (green). Y-axis: Fluorescence intensity (AU), 0 to 4,000. X-axis: Distance (μm), 0.5 to 2.0.

**f** STLC-treated mitotic cell
Optical sections → Region 3, Region 2, Region 1

**g** Region 1, Region 2, Region 3 — 12G10; Region 1, Region 2, Region 3 — EGFP-KIND2; YPet–talin-1; EGFP

**h** Cell number per area. Untreated (black), STLC-arrested (red). FN: P < 0.0001; VN: P < 0.0001; Serum: P < 0.0001; BSA: P = 0.0785.

and in cytosols of mitotic mouse fibroblasts deficient in both KIND1 and KIND2 (ref. [43]) that re-express EGFP–KIND2-S181A (Extended Data Fig. 3e). In cycling HeLa cells, the anti-KIND2-pS181 immuno-fluorescence signal was first visible in the cytosol of prophase cells, peaked in metaphase, decreased in anaphase, decreased further in telophase and vanished at cytokinesis (Fig. 3f). The KIND2-pS181 signal was also visible by SIM in puncta of retraction fibres (Fig. 3g). The KIND2-pS181 oscillation (Fig. 3h) resembles the oscillatory behaviour of CDK1–cyclin B1 activity during cell division[44,45] and negatively correlates with KIND2 abundance.

**Fig. 2 | Localization of non-degraded KIND2 in mitotic adhesions. a,b,** Stills of live-cell recordings generated by TIRF microscopy showing a representative interphase and mitotic (30 min after RO3306 release) HeLa cell stably expressing EGFP–KIND2 (**a**) or YPet–talin-1 (**b**), and Lifeact–mRuby and H2B–CFP. **c,d,** SIM images (**c**) of EGFP–KIND2 (green), the $\beta_1$ integrin activation-associated 12G10 epitope (red) and phalloidin (magenta) in interphase and mitotic (30 min after RO3306 release) HeLa cells. Co-localization of EGFP–KIND2 with the 12G10 epitope is shown in FAs (arrowheads) of interphase cells and in puncta in retraction fibres and round cell bodies of mitotic cells. To visualize EGFP–KIND2 signals in retraction fibres, the fluorescence was enhanced by increasing the exposure time. Boxes indicate areas of retraction fibres shown at high magnifications. The arrow indicates the direction of line profile analysis (**d**) of EGFP–KIND2 and 12G10 epitope (red). Each line-profile assessment was done three times. AU, arbitrary unit. **e,** SIM images of immunostained endogenous KIND2 (green), phalloidin (red) and pH3 (blue) in interphase and mitotic (30 min after RO3306 release) HeLa cells. Arrowheads indicate KIND2 in FAs. KIND2 in mitotic retraction fibres was visualized by increasing the exposure time. Boxes indicate retraction fibre areas shown at high magnifications. **f,** Schematic of the procedure, depicting the Z positions of round mitotic (30 min after RO3306 release) HeLa cell bodies: region 1, bottom stack; region 3, stack capturing the cortical end of retraction fibres; region 2, intermediate stack between region 1 and 3. Region 1 was recorded by TIRF microscopy and regions 2 and 3 by confocal microscopy. **g,** Representative image sections at different regions as depicted in **f** of round mitotic (30 min after RO3306 release) HeLa cell bodies immunostained for the 12G10 epitope or stably expressing EGFP–KIND2, YPet–talin-1 or EGFP. **h,** Quantification of adhesion assays of untreated and STLC-arrested HeLa cells on FN, VN, fetal bovine serum or BSA coatings. Mean ± s.d.; $n = 9$ independent experiments. P values calculated by unpaired Student's t-test (two-tailed). Scale bars, 1 μm (magnifications in **c** and **e**) or 10 μm (**a**–**c**,**e**,**g**).

## KIND2 phosphorylation is indispensable for normal mitosis.

To analyse the significance of KIND2-S181 phosphorylation, we measured protein stability in cycloheximide-treated HeLa cells expressing the following proteins: EGFP-tagged phospho-mimetic KIND2-S181D; the phospho-inhibitory mutants KIND2-S181A and KIND2-5SA, in which in addition to S181, four adjacent serine residues that are phosphorylated in EGFP–KIND2-WT and EGFP–KIND2-S181A (Supplementary Tables 2–4) were substituted for alanine residues; and KIND2-SWAP, in which the KIND2-F1-loop was replaced by the KIND3-F1-loop lacking CDK1 consensus motifs (Extended Data Fig. 4a). The KIND2-S181D mutation reduced protein stability in interphase, which was rescued by the proteasome inhibitor MG132 but not the lysosome inhibitor bafilomycin A1. This result indicates that phosphorylated KIND2 is degraded by the proteasome (Fig. 4a,b and Extended Data Fig. 4b). The protein stability of KIND2-S181A, KIND2-5SA and KIND2-SWAP remained unaffected in interphase (Fig. 4a,b) and stable in mitosis, which was more pronounced in KIND2-5SA and KIND2-SWAP cells than in KIND2-S181A cells (Fig. 4c,d). These results confirm our MS analysis and previous studies[46,47] showing that phosphorylated residues adjacent to the CDK1 consensus site can contribute to protein degradation.

To analyse the role of KIND2 degradation in cell division, we retrovirally transduced KIND1 and KIND2 double-floxed mouse fibroblasts with complementary DNAs encoding WT or mutant EGFP–KIND2 and then removed the *FERMT1* and *FERMT2* alleles by adenoviral *Cre* transduction[43]. Compared with KIND2-WT, expression of KIND2-S181D and KIND2-5SA or KIND2-SWAP reduced cell proliferation (Extended Data Fig. 4c) and increased cell numbers with 2N-4N aneuploid DNA content, >4N polyploid/aneuploid DNA content and apoptotic cell numbers represented by the sub-$G_0$/$G_1$ peak (<2N)[48] (Fig. 4e). Live-imaging of RO3306-released M-phase cells revealed that KIND2-S181D

fibroblasts accomplished mitotic rounding faster and required almost three times longer (195 ± 50 min; mean ± s.d.) to complete mitosis than KIND2-WT fibroblasts (73 ± 14 min) (Fig. 4f–h and Supplementary Videos 2 and 3). Unlike KIND2-S181D cells, the KIND2-5SA and KIND2-SWAP fibroblasts failed to undergo mitotic rounding and showed an increased rate of apoptosis. Moreover, even cells that were able to adopt a round-like shape and divide required 50% more time to complete mitosis (110 ± 34 min for KIND2-5SA and 114 ± 25 min for KIND-SWAP) compared with KIND2-WT cells (Fig. 4f–l and Supplementary Videos 4–6).

Next, we searched for an association between the abnormal mitotic rounding of mutant KIND2-expressing fibroblasts and the FA defects. Immunostaining of FN-seeded interphase fibroblasts revealed that in comparison to KIND2-WT fibroblasts, KIND2-S181D expression decreased FA numbers but did not change FA sizes (Extended Data Fig. 4d–f). By contrast, the expression of KIND2-5SA and KIND2-SWAP increased numbers and slightly decreased FA sizes (Extended Data Fig. 4d–f). In early M phase (5 and 10 min after RO3306 release; Fig. 4j,k), KIND2-WT fibroblasts gradually reduced in size and eventually lost their interphase FAs, which was accompanied by a gradual increase in cell rounding (Supplementary Video 2). In KIND2-S181D fibroblasts, the few FAs were rapidly disassembled and were absent by 5 min after RO3306 release, which resulted in a fulminant cell rounding (Fig. 4j,k and Supplementary Video 3). Conversely, KIND2-5SA and KIND2-SWAP fibroblasts retained FAs and the flat morphology in M phase (Fig. 4j,k and Supplementary Videos 4 and 5). Cell circularity measurements confirmed the transition from a flat interphase morphology to a round M-phase morphology in KIND2-WT and KIND2-S181D fibroblasts, whereas this transition failed in fibroblasts expressing KIND2-5SA or KIND2-SWAP (Fig. 4l). Importantly, the phospho-mimetic KIND1-S179D and the phospho-inhibitory K-S179A and KIND1-5SA (serine for alanine

**Fig. 3 | CDK1–cyclin B1 mediates KIND2 phosphorylation at mitotic entry. a,** Volcano plot of phospho-peptides of untreated versus STLC-arrested mitotic HeLa cells. KIND1-pS179 (K1 S179) and KIND2-pS181 (K2 S181) are highlighted in red, paxillin-pS106 and paxillin-pS109 (PXN S106/109) in blue. The black line indicates the significance cut-off (false discovery rate of 0.05, S0:1) estimated using Perseus software. **b,** Alignment of the mitotic phospho-motifs of KIND1 and KIND2 in different vertebrates. Asterisks indicate serine residues that were identified as phosphorylated by MS analysis. The phospho-motif is absent in KIND3. **c,** GFP immunoprecipitation (IP) from untreated and STLC-arrested HeLa cells expressing EGFP–KIND2 or EGFP only and analysed by WB for CDK1, cyclin B1 and cyclin A2. **d,** WB of KIND2-pS181 in untreated and STLC-arrested HeLa cells expressing endogenous KIND2 and overexpressing EGFP–KIND2-WT or EGFP–KIND2-S181A in the presence of 5 μg ml⁻¹ non-phosphorylated KIND2 peptide (GSGSIYSSPGLYSKT). **e,** WB of KIND2 and KIND2-pS181 in untreated and STLC-arrested HeLa cell lysates treated with or without alkaline and lambda phosphatases (PPs). **f,** Representative confocal images of KIND2-pS181 (green), phalloidin (red) and DAPI (blue) in HeLa cells at indicated cell cycle phase. The area with nucleus/chromosomes is magnified in the bottom panel. **g,** SIM images of KIND2-pS181 (green) and phalloidin (red) in round mitotic HeLa cell bodies and retraction fibres (boxes show high magnifications of retraction fibres; 30 min after RO3306 release). **h,** Quantification of the KIND2-pS181 signal at indicated cell cycle phases of HeLa cells. $n = 20$ cells for each cell cycle phase pooled from 3 independent experiments. All WB experiments were repeated at least three times, with similar results obtained. pH3 served as an indicator for mitosis and GAPDH as the loading control. Scale bars, 2.5 μm (magnifications in **g**) or 10 μm (**f,g**).

substitutions of S179 and the adjacent serine residues S172, S174, S176 and S184) substitutions affected protein stability and rounding in M phase similar to the corresponding KIND2 mutants (Extended Data Fig. 4g,h). Finally, we investigated whether the Rap1–talin axis can override the role of KIND2 in M-phase. Overexpression of constitutive active Rap1-G12V did not prevent the accelerated

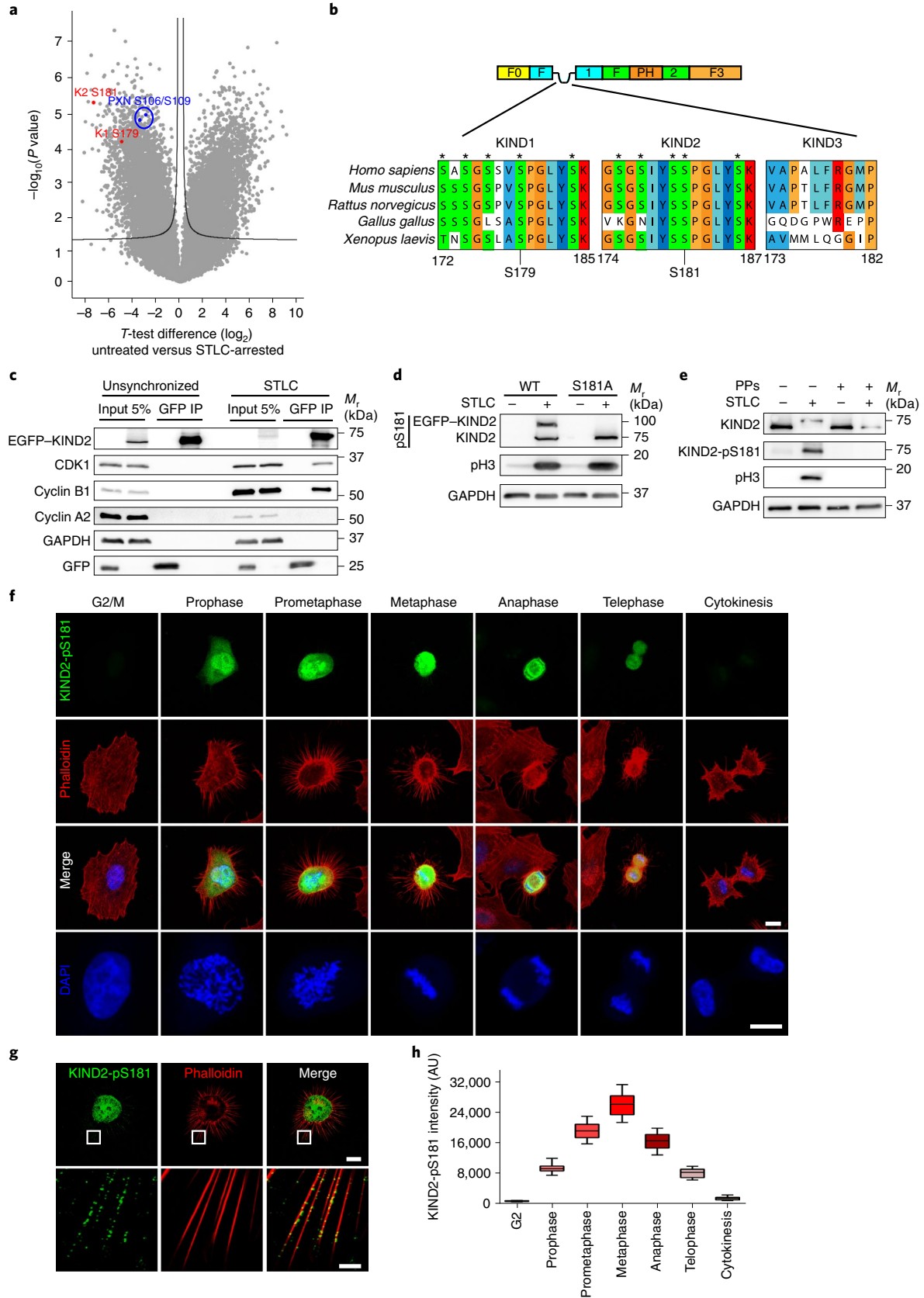

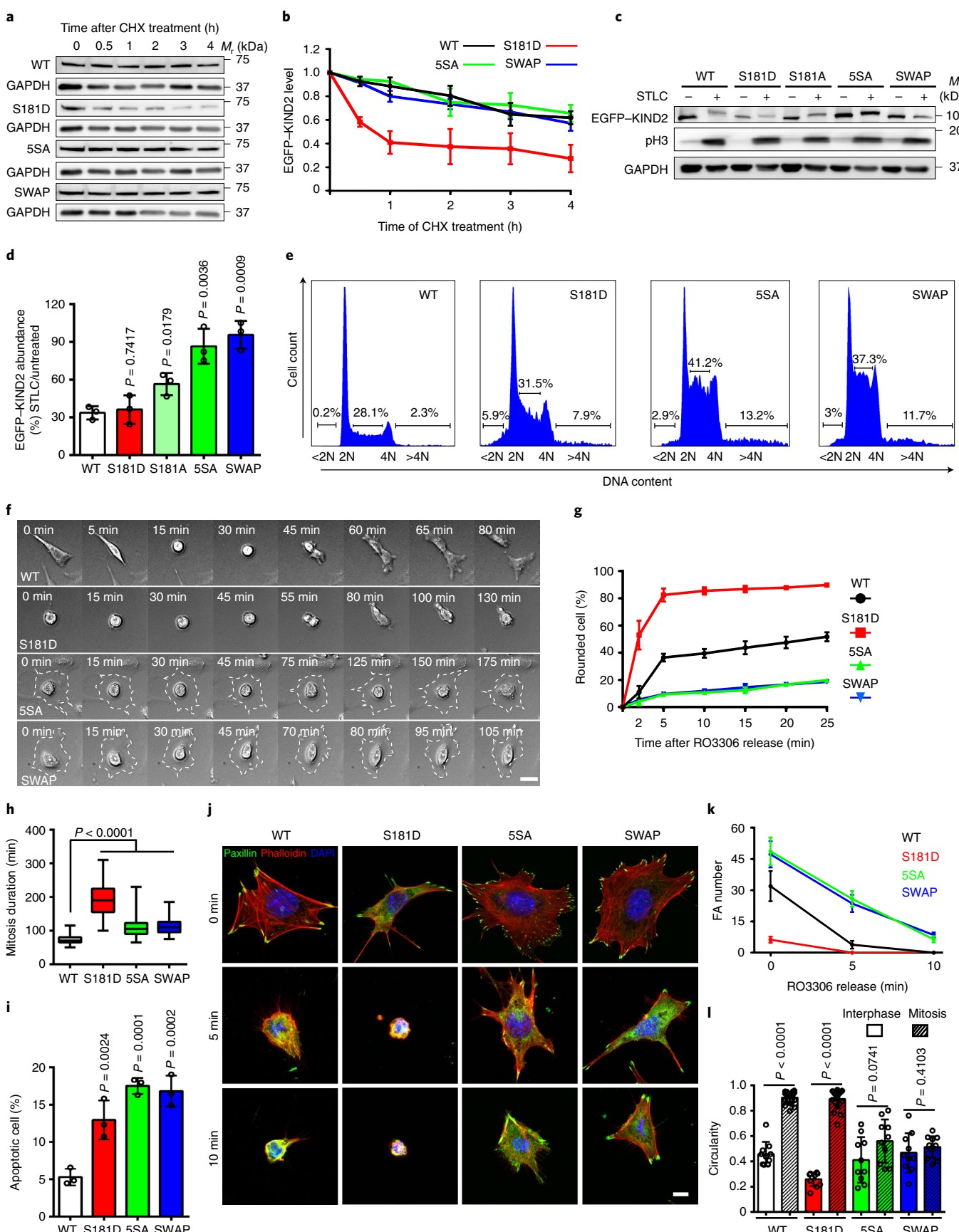

**Fig. 4 | KIND2 degradation is indispensable for mitotic rounding. a,b**, WB of GFP (**a**) and quantification (**b**) of EGFP–KIND2 stability in HeLa cells expressing EGFP-tagged KIND2-WT, KIND2-S181D, KIND2-5SA or KIND2-SWAP treated with 10 μg ml⁻¹ cycloheximide (CHX). Mean ± s.d.; $n = 3$ independent experiments. **c,d**, WB of GFP (**c**) and ratio (**d**) of EGFP–KIND2 levels in untreated and STLC-arrested HeLa cells expressing EGFP-tagged KIND2-WT, KIND2-S181D, KIND2-S181A, KIND2-5SA or KIND2-SWAP. Mean ± s.d.; $n = 3$ independent experiments. **e**, DNA contents of propidium-iodide-loaded untreated fibroblasts expressing EGFP-tagged KIND2-WT, KIND2-S181D, KIND2-5SA or KIND2-SWAP. Percentage of aneuploid cells is indicated. **f–h**, Montage of phase-contrast recordings (**f**), percentage of rounded fibroblasts (**g**) and M-phase duration (**h**) of fibroblasts expressing EGFP-tagged KIND2-WT, KIND2-S181D, KIND2-5SA or KIND2-SWAP after RO3306 release. Dashed outlines in **f** indicate cell margins. For **g**, data are the mean ± s.d.; 200 cells counted for each cell line, $n = 3$ independent experiments. For **h**, the time spans from cell margin retraction to completion of cytokinesis. $n$ (cell number) = 61 for WT, 51 for S181D, 53 for 5SA and 54 for SWAP pooled from 3 independent experiments. **i**, Percentage of apoptotic fibroblasts expressing EGFP-tagged KIND2-WT, KIND2-S181D, KIND2-5SA or KIND2-SWAP quantified 3 h after RO3306 release by staining for caspase 3 and caspase 7. Mean ± s.d.; 100 cells counted for each cell line, $n = 3$ independent experiments. **j,k**, Representative confocal images of immunostained paxillin (green), phalloidin (red) and DAPI (blue) (**j**) and quantification (**k**) of paxillin-positive FAs in fibroblasts expressing EGFP-tagged KIND2-WT, KIND2-S181D, KIND2-5SA or KIND2-SWAP at the indicated times after RO3306 release. Mean ± s.d.; $n = 10$ cells of each cell line quantified at each time point pooled from 2 independent experiments (**k**). **l**, Circularity of unsynchronized fibroblasts expressing EGFP-tagged KIND2-WT, KIND2-S181D, KIND2-5SA or KIND2-SWAP. Mean ± s.d.; interphase, $n = 10$ cells for each cell line; mitosis, $n = 20$ for WT and S181D, 10 for 5SA and 12 for SWAP pooled from 2 independent experiments. $P$ values in **d,h** and **i** were calculated by one-way analysis of variance (ANOVA) Dunnett's multiple comparison test (95% confidence interval (CI)). $P$ values in **l** were calculated by unpaired Student's $t$-test (two-tailed). All WB experiments were repeated three times, with similar results obtained. GAPDH served as loading control and pH3 as an indicator for mitosis. Scale bars, 10 μm (**j**) or 30 μm (**f**).

rounding of KIND2-S181D fibroblasts (Extended Data Fig. 4i–k), and simultaneous depletion of Rap1A and Rap1B in flat KIND2-5SA and KIND2-SWAP fibroblasts did not rescue the rounding defects (Extended Data Fig. 4l–n). These results indicate that CDK1–cyclin-B1-mediated KIND1 and/or KIND2 phosphorylation and degradation is indispensable for FA disassembly and cell rounding at mitotic entry and cannot be rescued by modulating the Rap1 GTPase activity.

Retraction fibres are important for adhesion of M-phase cells and the assembly of cortical landmarks that guide the assembly and positioning of the bipolar spindle parallel to the substrate[10,49], which is essential to capture, align and separate chromosomes[50]. To determine the number and length of retraction fibres, we analysed RO3306-released mitotic mouse fibroblasts expressing either stable or unstable KIND2 on FN-coated L-shaped micropatterns. Compared with KIND2-WT fibroblasts, there were fewer and shorter retraction fibres in mitotic KIND2-S181D fibroblasts and more in KIND2-5SA and KIND2-SWAP fibroblasts (Fig. 5a–c). The aberrant formation of mitotic retraction fibres in KIND2-S181D cells as well as the flat shape of KIND2-5SA and KIND2-SWAP fibroblasts produced severe mitotic spindle defects. Anti-α-tubulin and anti-γ-tubulin antibodies visualizing the mitotic spindle and centrosomes revealed that ~80% of RO3306-synchronized KIND2-WT fibroblasts displayed bipolar spindles with their poles arranged at an angle <15° between the spindle axis and substrate. By contrast, >50% of KIND2-S181D or KIND2-5SA and KIND2-SWAP fibroblasts showed aberrant spindles with supernumerary centrosomes that were randomly oriented in KIND2-S181D fibroblasts and at an angle of <15° in KIND2-5SA and KIND2-SWAP

fibroblasts. These effects were probably due to space confinement in the Z dimension of these flattened cells (Fig. 5d–f and Extended Data Fig. 5a,b). The consequences of the spindle defects included chromosome capture defects (Fig. 5d and Extended Data Fig. 5a,b), supernumerary centrosomes (Extended Data Fig. 5c,d), prolonged activation of the spindle assembly checkpoint (SAC) (revealed by aberrant Mad2 staining; Extended Data Fig. 5e,f) and increased numbers of γH2AX foci and polyploidy (Extended Data Fig. 5g–k) in KIND2-S181D as well as KIND2-5SA and KIND2-SWAP fibroblasts.

**The CUL9–FBXL10 complex promotes KIND2 ubiquitination in mitosis.** The degradation of KIND2-S181D by the proteasome in interphase suggests that E3-ligase-induced ubiquitination is triggered by KIND2 phosphorylation. This hypothesis was supported by MS, which identified 15 ubiquitination sites distributed along the entire KIND2 protein immunoprecipitated from STLC-arrested mitotic HeLa cells (Supplementary Table 5). To identify the ubiquitin conjugation enzymes, we screened a short interfering RNA (siRNA) library targeting >1,000 ubiquitin modifiers for increasing EGFP–KIND2-S181D fluorescence (cut-off > 1.5-fold) in HeLa cells (Extended Data Fig. 6a). The screen identified ten siRNA targets (FBXL10 (also known as KDM2B), CUL9, MID2, MTF2, NEDD4, RNF34, USP9Y, BMI1, USP54 and UBE2W) that increased EGFP–KIND2-S181D fluorescence. The HECT-type ubiquitin ligase Smad ubiquitination regulatory factor-1 (SMURF1), which can mediate proteasomal turnover of KIND2 in interphase[51], was not identified in our screen. FBXL10, CUL9 and MID2 emerged as most prominent hits (Extended Data Fig. 6a,b). Since MID2 is

**Fig. 5 | KIND2 degradation is indispensable for retraction fibre formation, spindle assembly and positioning. a**, Representative confocal images of phalloidin (red) and DAPI (blue) in interphase and mitotic (30 min after RO3306 release) mouse fibroblasts expressing EGFP-tagged KIND2-WT, KIND2-S181D, KIND2-5SA or KIND2-SWAP seeded on FN-coated L-shaped micropatterns. Boxes indicate areas of chromosomes shown at high magnifications. Scale bar, 10 μm. Scale bar of magnifications showing chromosomes, 5 μm. **b**, Quantification of retraction fibre length in mitotic mouse fibroblasts engineered and cultured as described in **a**. $n$ (number of retraction fibres) = 164 for WT, 21 for S181D, 128 for 5SA and 116 for SWAP pooled from 3 independent experiments. $P$ values calculated by Kruskal–Wallis Dunnett's multiple comparison test. **c**, Quantification of retraction fibres in mitotic mouse fibroblasts engineered and cultured as described in **a**. Mean ± s.d.; $n = 10$ cells for each cell line pooled from 3 independent experiments. $P$ values calculated by one-way ANOVA Dunnett's multiple comparison test (95% CI). **d**, Representative confocal images of α-tubulin (red), γ-tubulin (green) and DAPI (blue) in mitotic (30 min after RO3306 release) mouse fibroblasts expressing EGFP-tagged KIND2-WT, KIND2-S181D, KIND2-5SA or KIND2-SWAP. Maximal projections of Z-stack images beginning with Z-stacks capturing the first spindle from the bottom, the centrosomes and the last spindle at the top. Scale bar, 5 μm. **e**, Cartoon showing how spindle angles were calculated. The spindle angle ($\alpha°$) was defined as the angle between the spindle axis and substrate surface. Lengths of $a$ and $b$ represent the vertical and horizontal distances, respectively, between two spindle poles. The spindle angle ($\alpha°$) was calculated based on the equation shown on the top. **f**, Spindle angle distribution in mitotic mouse fibroblasts expressing EGFP-tagged KIND2-WT, KIND2-S181D, KIND2-5SA or KIND2-SWAP; $n = 50$ cells analysed for KIND2-WT and each mutant, pooled from 3 independent experiments.

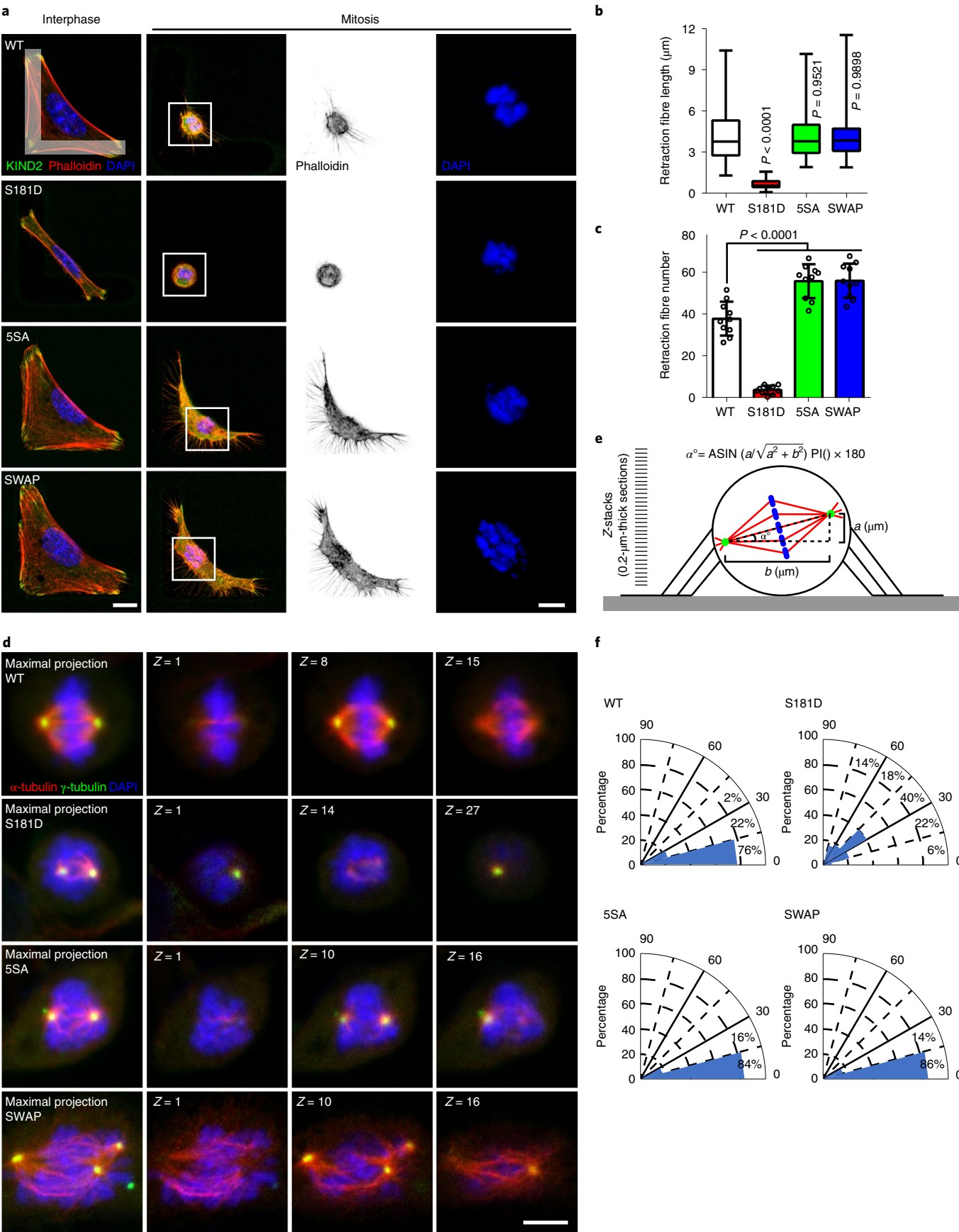

$$\alpha° = ASIN (a/\sqrt{a^2 + b^2}) \ PI() \times 180$$

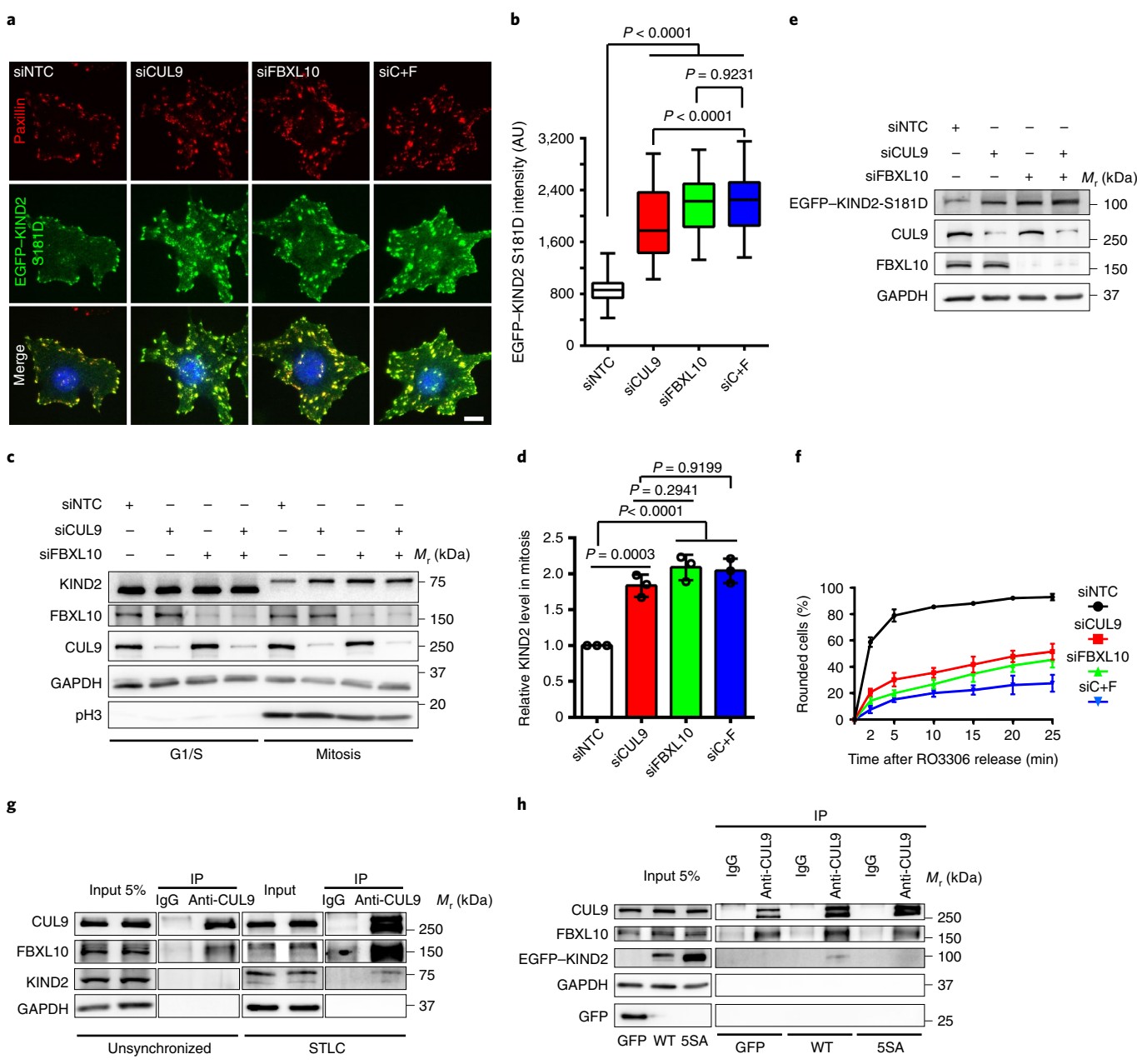

**Fig. 6 | The CUL9–FBXL10 complex promotes KIND2 degradation in mitosis. a**, Representative confocal microscopy images of HeLa cells expressing EGFP-tagged KIND2-S181D (green) transfected with non-targeting control siRNA (siNTC), siCUL9, siFBXL10 or siC+F and stained for paxillin (red) and DAPI (blue). Scale bar, 10 μm. **b**, Quantification of EGFP intensity in HeLa cells expressing EGFP-tagged KIND2-S181D transfected with siNTC, siCUL9, siFBXL10 or siC+F. n = 50 cells analysed for each transfection, pooled from 3 independent experiments. **c**, WB of endogenous KIND2 in D-THY-synchronized G1/S phase (0 h after release) and mitotic HeLa cells (9 h after release combined with mitotic shake-off) transfected with siNTC, siCUL9, siFBXL10 or siC+F. **d**, WB quantifications of endogenous KIND2 levels in D-THY synchronized G1/S phase (0 hr after release) and mitotic HeLa cells (9 hr after release combined with mitotic shake-off) transfected with siNTC, siCUL9, siFBXL10 or siC+F. Mean ± s.d.; n = 3 independent experiments. **e**, WB of EGFP, CUL9 and FBXL10 from mouse fibroblasts expressing EGFP-tagged KIND2-S181D and transfected with siNTC, siCUL9, siFBXL10 or siC+F. **f**, Percentage of rounded EGFP-tagged KIND2-S181D-expressing mouse fibroblasts transfected with siNTC, siCUL9, siFBXL10 or siC+F quantified after RO3306 release. Mean ± s.d.; n = 200 cells counted for each transfection, n = 3 independent experiments. **g**, IP of CUL9 from untreated and STLC-arrested mitotic HeLa cells followed by WB for CUL9, FBXL10 and KIND2. **h**, IP of CUL9 in STLC-arrested mitotic HeLa cells expressing EGFP only, EGFP-tagged KIND2-WT or KIND2-5SA followed by WB for CUL9, FBXL10 and GFP. P values in **b** and **d** were calculated by one-way ANOVA Dunnett's multiple comparison test (95% CI). All WB experiments were repeated three times, with similar results obtained. GAPDH served as the loading control and pH3 as an indicator for mitosis.

activated at mitotic exit[52,53], we decided to focus our analysis on CUL9 and FBXL10, which have been shown to prevent genome instability, aneuploidy and spontaneous tumour development in cells and mice[54–59].

Depletion of CUL9 or FBXL10 with specific siRNAs (siCUL9 or siFBXL10, respectively) increased the abundance of EGFP–KIND2-S181D-positive adhesion sites in interphase, increased endogenous KIND2 levels in M phase in HeLa cells (Fig. 6a–d)

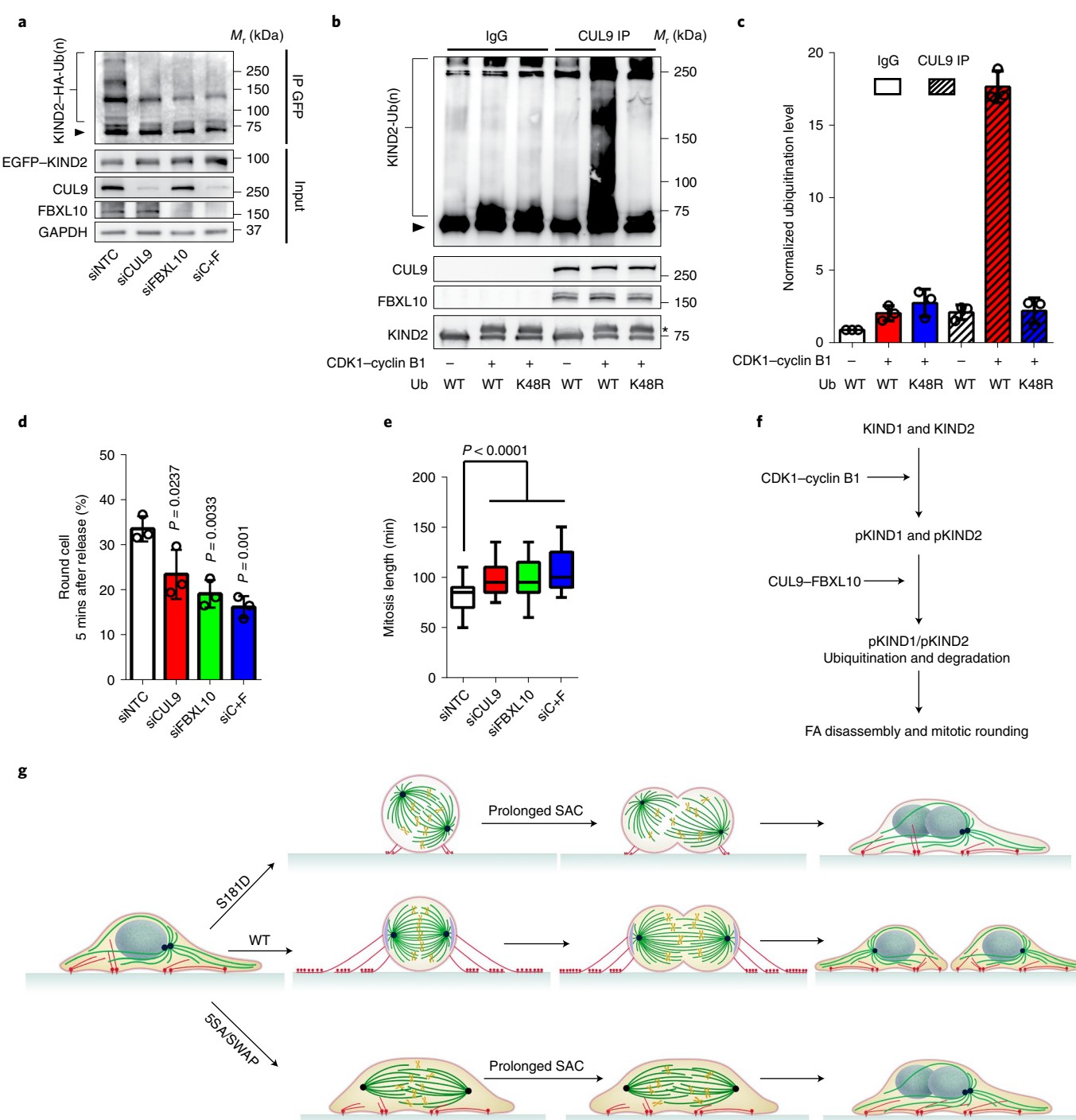

and stabilized the KIND2-S181D protein, which in turn decelerated mitotic rounding of KIND2-S181D-expressing mouse fibroblasts (Fig. 6e,f). The simultaneous depletion of CUL9 and FBXL10 (siC+F) had no additive effect, which suggests that both proteins function in the same pathway (Fig. 6a–f). MS of CUL9 and FBXL10 immunoprecipitates from M-phase HeLa cell lysates revealed an association with SKP1, RBX1, NEDD8 and KIND2 (Supplementary Tables 6 and 7), which indicates that these proteins form a canonical cullin-RING ligase (CRL) complex targeting KIND2 for ubiquitination[60,61]. Immunoprecipitation of CUL9 followed by immunoblotting confirmed an association with FBXL10 in interphase and mitosis and with phosphorylated KIND2 in STLC-arrested mitotic HeLa cells (Fig. 6g). The inability of the CUL9–FBXL10 complex to

precipitate KIND2-5SA from lysates of STLC-arrested HeLa cells (Fig. 6h) provides support for a phosphorylation-dependent interaction with the CUL9–FBXL10 complex.

Depletion of CUL9 and/or FBXL10 in STLC-arrested HeLa cells suppressed KIND2 polyubiquitination in M phase (Fig. 7a), and recombinant KIND2 phosphorylated by CDK1–cyclin B1 in vitro was readily modified by the CUL9–FBXL10 complex immunoprecipitated from HeLa cell lysates with WT but not K48R-substituted ubiquitin (Fig. 7b,c). The depletion of CUL9 and/or FBXL10 also curbed rounding of mitotic fibroblasts (Fig. 7d), and those cells that underwent rounding passed through a prolonged M phase (Fig. 7e and Extended Data Fig. 6c) and displayed mitotic spindle defects, chromosome misalignment and increased DNA damage (Extended

**Fig. 7 | The CUL9–FBXL10 complex ubiquitinates phosphorylated KIND2, leading to FA disassembly at mitotic entry. a**, GFP IP from STLC-arrested mitotic HeLa cells expressing EGFP-tagged KIND2-WT and HA-tagged ubiquitin (Ub) transfected with siNTC, siCUL9, siFBXL10 or siC+F followed by WB with anti-HA antibody. Whole-cell lysates (5%) were analysed by WB for KIND2, CUL9 and FBXL10. The arrowhead indicates non-ubiquitinated KIND2. GAPDH served as the loading control. **b**, In vitro ubiquitination of non-phosphorylated or CDK1–cyclin B1-phosphorylated recombinant KIND2 in the presence of WT-Ub or K48R-Ub by control rabbit IgG or anti-CUL9 precipitates and detected by anti-KIND2 WB experiments. The arrowhead indicates non-ubiquitinated KIND2 and the asterisk phosphorylated KIND2. IPs and recombinant KIND2 were analysed by WB for KIND2, CUL9 and FBXL10. **c**, Quantification of polyubiquitination measured by WB of in vitro ubiquitinated KIND2. Mean ± s.d.; $n = 3$ independent experiments. **d**, Percentage of rounded mouse fibroblasts transfected with siNTC, siCUL9, siFBXL10 or siC+F quantified 5 min after RO3306 release. Mean ± s.d.; $n = 100$ cells counted for each transfection, $n = 3$ independent experiments. **e**, M-phase duration of FN-seeded mouse fibroblasts transfected with siNTC, siCUL9, siFBXL10 or siC+F. The time spans from cell margin retraction to completion of cytokinesis. $n = 50$ cells for each transfection, pooled from 3 independent experiments. $P$ values in **d** and **e** were calculated by one-way ANOVA Dunnett's multiple comparison test (95% CI). All WB experiments were repeated three times, with similar results obtained. **f**, In early prophase, the CDK1–cyclin B1 complex phosphorylates KIND1 and KIND2 at the F1 loop and creates a phospho-degron. This recruits the CUL9–FBXL10 complex and results in the ubiquitination and proteasomal degradation of KIND1 and KIND2. **g**, KIND1 and KIND2 degradation induces FA disassembly, which is followed by the formation of retraction fibres and mitotic rounding. Cells expressing phospho-mimetic KIND2 (KIND2-S181D) display impaired retraction fibre formation, which leads to misorientation of mitotic spindles, prolonged SAC activation and either cell death or defects in chromosome segregation and aneuploidy. In cells expressing phospho-inhibitory KIND2 (KIND2-5SA or KIND2-SWAP), FA disassembly and mitotic cell rounding fail, which leads to defective spindle anchorage, aberrant chromosome capture and alignment, prolonged SAC activity and either cell death or defects in chromosome segregation and aneuploidy.

Data Fig. 6d–h). These findings indicate that CDK1–cyclin B1 generates a phospho-degron on KIND1 and KIND2, which recruit the CUL9–FBXL10 complex that in turn ubiquitinates and routes the protein to the proteasome for degradation. Furthermore, the observation that CUL9 and FBXL10 are present in the round body of M-phase HeLa cells but only FBXL10 in EGFP–KIND2-positive puncta in retraction fibres (Extended Data Fig. 6i,j) provides an explanation as to why KIND2 escapes degradation at these sites.

## Discussion

Integrin-containing FAs increase at S-phase entry in a CDK1-dependent and cyclin-A2-dependent manner[15] and disassemble at the onset of mitosis. How cells achieve FA disassembly at mitotic entry is unknown. In the current study, we showed that the CDK1–cyclin B1 complex binds and phosphorylates the integrin activators KIND1 and KIND2, which triggers their ubiquitination and degradation followed by the disassembly and remodelling of FAs that subsequently enables the change in cell morphology from flat to round.

KIND1 and KIND2 levels were substantially decreased in M phase of all epithelial and mesenchymal cells tested, irrespective of the ECM substrate to which they were attached. The remaining ~20% of the protein localized together with talin, paxillin and vinculin at punctiform integrin-containing attachment sites in retraction fibres and throughout the cytoplasm of the rounded, dividing cell. In search for a mechanistic explanation for the decrease in kindlin protein levels at mitotic entry, we identified several steps of a signalling pathway (Fig. 7f), in which the M-phase-specific CDK1–cyclin B1 complex directly binds KIND1 and KIND2 and phosphorylates a conserved proline-directed CDK1 consensus motif in the flexible and intrinsically disordered loop of the F1 domain. This then results in the recruitment of the CUL9–FBXL10 complex, modification with K48-linked polyubiquitin chains and proteasomal degradation of KIND1 and KIND2. In support of these findings, we observed that substitution of the CDK1-directed phosphosites in KIND2 with the phospho-inhibitory alanine or the depletion of CUL9 and/or FBXL10 inhibit KIND2 degradation, FA disassembly and mitotic rounding, thereby forcing cells to enter M phase with a flattened shape. By contrast, the substitution of S181 in KIND2 with the phospho-mimetic residue aspartate decreased protein stability in interphase and prematurely induced a spherical cell shape at mitotic entry with few and short retraction fibres that poorly attached the mitotic cell to the ECM substrate. The decreased protein stability of KIND2 induced by the S181D substitution indicates that the CDK1–cyclin-B1-mediated phosphorylation of S181 becomes an effective phospho-degron. Our phosphoproteome analysis also

identified additional serine residues adjacent to KIND1-S179 and KIND2-S181 that are phosphorylated in M phase, although to a lesser extent. This observation is in line with previous searches for CDK1 phosphosites[46,62,63], which reported that CDK1–cyclin B phosphorylates both proline-directed and non-proline-directed motifs. Indeed, degradation of KIND2 was further inhibited when serine residues adjacent to S181 were alanine-substituted (KIND2-5SA) or the KIND2-F1-flexible loop was exchanged with the corresponding KIND3 sequence (KIND2-SWAP), which lacks CDK1-directed phosphosites. The lack of the CDK1-mediated phospho-degron in KIND3 can probably be attributed to the intrinsically round-like shape of haematopoietic cells, which equips them a priori with a cellular geometry suited to divide in an error-free manner.

Despite the opposing consequences of stable and unstable kindlin protein for the geometry of mitotic cells, the ramifications for cells are similar and include spindle and chromosome capture defects, prolonged SAC activation, frequent chromosome segregation errors and cell death or aneuploidy (Fig. 7g). These mitotic defects were primarily caused by severe spindle abnormalities. Flat mitotic cells expressing the stable KIND2 protein have spindle–chromosome misalignment due to spatial restrictions in the Z-dimension[5,8], and round mitotic cells expressing instable KIND2 display spindle positioning defects due to the sparse and unstable retraction fibres and insufficient formation of cortical spindle attachment sites[64–66]. Notably, spindle defects and aneuploidy, which result in lagging chromosomes, increased replication stress and DNA double-strand breaks[67,68], also occur in cells and/or mice lacking CUL9 and FBXL10, respectively[56–59,69,70]. The observation that CUL9-deficient mice develop tumours in multiple organs suggests that ubiquitination of phosphorylated kindlin and degradation fails in mitosis, which leads to the mitotic defects and cancer.

Our findings show that kindlin degradation is essential to induce mitotic rounding. Whether additional pathways, such as Rap1 GTPase-mediated inactivation of talin, regulation of actomyosin or the activity control of further FA proteins, cooperate with kindlin degradation to induce FA disassembly and mitotic rounding is not clear and requires future investigations. An interesting candidate is paxillin, the phosphorylation of which (also induced in M phase) was shown to sensitively control integrin adhesion stability by recruiting proteins such as vinculin, PAK or β-Pix[71–73]. Irrespective of whether the activity of additional FA proteins changes at mitotic entry, the orchestration of kindlin degradation by the CDK1–cyclin B1 complex explains how interphase FAs disassemble and why loss of kindlin function promotes spindle abnormalities, aneuploidy and skin cancer in human Kindler syndrome[74].

## Online content

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

## Methods

**Antibodies and reagents.** The following antibodies were used for western blotting (WB), immunofluorescence (IF) and flow cytometry (FC): KIND2 (clone 3A3, MAB2617, mouse, Merck Millipore; 1:1,000 for WB, 1:200 for IF); KIND2-pS181 (home-made; 1:5,000 for WB; 1:400 for IF); KIND1 (home-made[35]; 1:1,000 for WB); talin-1 (T3287, Sigma; 1:1,000 for WB); ILK (3856, rabbit, Cell Signaling; 1:1,000 for WB); PINCH (612710, mouse, Transduction Laboratories; 1:1,000 for WB); vinculin (sc-5573, rabbit, Santa Cruz; 1:1,000 for WB; V9131, Sigma; 1:500 for IF); FAK (3285, rabbit, Cell Signaling; 1:1,000 for WB); paxillin (610051, mouse, Transduction Laboratories; 1:1,000 for WB, 1:300 for IF); $\beta_1$ integrin (9699, rabbit, Cell Signaling; 1:1,000 for WB; TS2/16, mouse, BioLegend; 1:200 for IF; 12G10, MAB2247, mouse, Merck Millipore; 1:200 for IF; clone HMbeta1-1, 102207, hamster, BioLegend; 1:200 for FC); $\beta_3$ integrin (M109-3, rat, Biozol; 1:200 for IF; clone VI-PL02, 336406, mouse, BioLegend; 1:200 for FC); $\beta_5$ integrin (3629, rabbit, Cell Signaling; 1:200 for IF; ALULA, gift from D. Sheppard, 1:200 for IF; clone KN52, 12-0497, mouse, eBioscience; 1:200 for FC); $\alpha_5$ integrin (clone 5H10-27, 557447, rat, PharMingen; 1:200 for FC); $\alpha_V$ integrin (clone NKI-M9, 327909, mouse, BioLegend; 1:200 for FC); zyxin (clone 164D4, 307011, mouse, Synaptic system; 1:200 for IF); phospho-histone H3 (53348, rabbit, Cell Signaling; 1:1,000 for WB; 1:500 for IF); cyclin A2 (4656, mouse, Cell Signaling; 1:1,000 for WB); cyclin B1 (4138, rabbit, Cell Signaling; 1:1,000 for WB); cyclin D1 (2978, rabbit, Cell Signaling; 1:1,000 for WB); cyclin E2 (4132, rabbit, Cell Signaling; 1:1,000 for WB); GAPDH (CB1001, mouse, Calbiochem; 1:10,000 for WB); $\alpha$-tubulin (2144, rabbit, Cell Signaling; 1:100 for IF); $\gamma$-tubulin (T6557, mouse, Sigma; 1:500 for IF); CUL9 (A300-098A, rabbit, Bethyl Laboratories; 1:1,000 for WB; 1:200 for IF); FBXL10 (09-864, rabbit, Merck Millipore; 1:1,000 for WB; 1:200 for IF); pS139-γH2AX (clone JBW301, 05-636, mouse, Millipore; 1:200 for IF); MAD2 (A300-301A, rabbit, Bethyl Laboratories; 1:200 for IF); haemagglutin (HA)-tag (rat, Roche; 1:1,000 for WB); GFP (A10262, chicken IgY, Invitrogen; 1:1,000 for WB); and GFP-Booster Alexa Fluor 488 (gb2AF488, Chromotek; 1:500 for IF).

The following chemicals were used: Alexa Fluor 647 phalloidin (A22287, Thermo Fisher; 1:1,000 for IF); phalloidin-TRITC (P1591, Sigma, 1:2,000 for IF); thymidine (sc-296542, Santa Cruz); STLC (164739, Sigma); RO3306 (sc-358700, Santa Cruz); BioTracker NucView 530 Red Caspase 3/7 Dye (SCT103, Sigma); N-ethylmaleimide (E-3876, Sigma); protease inhibitor cocktail (4693159001, Roche); phosphatase inhibitor cocktail (P5726 and P0044, Sigma); MG132 (474787, Sigma); bafilomycin A1 (BVT-0252, Adipogen); and cycloheximide (sc-3508A, Santa Cruz).

**Cell culture.** HeLa cells (American Type Culture Collection (ATCC), CCL-2), HEK293T cells (ATCC, CRL-3216), U2OS cells (ATCC, HTB-96), RPE1 cells (CRL-4000) and SV40 large T-immortalized double-floxed mouse KIND1 and KIND2 fibroblasts[75] were cultured in DMEM medium (Gibco) with 10% FBS (Gibco) and 1% penicillin–streptomycin (Gibco). HAP1 cells (Horizon Discovery, C859) were cultured in IMEM medium (Gibco) with 10% FBS and 1% penicillin–streptomycin. To generate KIND1 and KIND2 double-null fibroblasts expressing EGFP-tagged WT or mutant KIND2, double-floxed mouse KIND1 and KIND2 fibroblasts were retrovirally transduced with the respective cDNA, sorted by flow cytometry and adenovirally transduced with *Cre* to delete the floxed *Fermt1* and *Fermt2* alleles. To visualize actin and chromosome dynamics, EGFP–KIND2-expressing HeLa cells were lentivirally transduced with cDNAs encoding Lifeact–mRuby and H2B–CFP, respectively.

**Plasmids, constructs and siRNA.** Mouse *Fermt1* cDNA was cloned into pRetroQ-EGFP-C1 (Clontech) using the restriction sites BglII and XbaI for transient transfection. Mouse *Fermt2* cDNA was cloned into pRetroQ-EGFP-C1 (Clontech) using the restriction sites XhoI and BamHI for retrovirus-mediated expression. The phospho-mimetic and phospho-inhibitory mutants were generated using a QuikChange Site-Directed Mutagenesis kit according to the manufacturer's protocol (Agilent). For retrovirus-mediated expression of YPet–talin and paxillin–GFP, the mouse YPet–talin or paxillin–GFP cassette was cloned into pLPCX (Clontech) using the restriction sites XhoI and NotI. The plasmid pLenti.PGK. LifeAct-Ruby.W was a gift from R. Lansford (Addgene, plasmid 51009; http://n2t.net/addgene:51009; RRID: Addgene51009). The plasmid LV-CFP was a gift from E. Fuchs (Addgene, plasmid 25998; http://n2t.net/addgene:25998; RRID: Addgene25998). On-target Plus smart pool siRNA and non-targeting siRNA control were purchased from Dharmacon. The detailed sequences of siRNA are listed in Supplementary Table 8.

**Transient transfection and stable viral transduction.** Cells were transiently transfected with plasmids using Lipofectamine 2000 or with siRNA using Lipofectamine RNAiMAX according to the manufacturer's protocol (Invitrogen). To generate stable cell lines, VSV-G pseudotyped retroviral and lentiviral vectors were produced by transient transfection of HEK293T cells. Viral particles were concentrated from cell culture supernatant as previously described[75] and used for infection.

**Cell cycle synchronization.** HeLa cells were treated with 2.5 mM thymidine for 16 h, subsequently released for 10 h and then treated again with thymidine for 16 h.

Cells were then collected for biochemical analyses or fixed for immunostaining after 0, 4, 7, 9 or 13 h. To enrich mitotic HeLa cells, they were treated with 2 μM STLC overnight and collected by mitotic shake-off. To analyse the mitotic rounding process, cells were arrested at the G2/M border by treating them with 0.9 μM of the CDK1 inhibitor RO3306 for 20 h, washed twice with PBS and then cultured in complete medium.

**Flow cytometry.** Interphase cells were trypsinized and mitotic cells were collected by shake-off. For surface integrin profiling, cells were washed with cold PBS twice and incubated in 200 μl antibody solution (PBS with 2% BSA) for 1 h on ice in the dark. Cells were washed once with cold PBS and applied on a LSRFortessa X-20 Cell Analyzer. For DNA content analysis, cells were fixed with 70% ethanol for 3 h at −20 °C, washed twice with cold PBS and resuspended in 200 μl FxCycle PI/RNase staining solution, incubated for 20 min in the dark and applied on a LSRFortessa X-20 Cell Analyzer. Data were analysis with FlowJo software.

**MTT assay.** Mouse fibroblasts expressing EGFP-tagged KIND2-WT, KIND2-S181D, KIND2-5SA or KIND2-SWAP were seeded onto FN-coated 96-well plates (2,000 cells per well) and measured every 24 h. Cells were washed with PBS twice and then incubated in 100 μl serum-free medium containing 2.5 mg ml⁻¹ MTT (3-(4,5-dimethylthiazol-2-yl)-2,5-diphenyltetrazolium bromide) at 37 °C for 3 h. A total of 100 μl MTT solvent (4 mM HCl, 0.1% NP-40 in isopropanol) was then added in each well and the plates were wrapped in foil and shaken on an orbital shaker for 10 min. The absorbance was recorded at $\lambda = 590$ nm.

**Adhesion assay.** HeLa cells grown to 50% confluence were incubated in the absence or presence of 2 μM STLC overnight. Untreated cells were collected by trypsinization and STLC-arrested mitotic cells by mechanical shake-off. The cells were washed twice with PBS, seeded on flat-bottom 24-well plates coated with 5 μg ml⁻¹ FN, 5 μg ml⁻¹ VN, 10% FBS or 3% BSA in complete medium (150,000 per well), incubated for 30 min at 37 °C, and then vigorously washed with PBS. The remaining attached cells were fixed with 3% paraformaldehyde (PFA) and stained with 4,6-diamidino-2-phenylindole (DAPI), imaged using an EVOS FL Auto2 microscope with a ×10 objective and counted.

**Generation of polyclonal rabbit anti-KIND2-pS181 antibodies.** The phosphorylated peptide containing the CDK1 consensus motif of KIND2 (GSGSIYSS(H3O4P) PGLYSKT) was synthesized in-house. The peptide was conjugated with keyhole limpet haemocyanin using an Imject Maleimide-Activated mcKLH Spin kit (77666, Thermo Fisher) according to the manufacturer's protocol. The conjugated immunogen was then mixed with adjuvant (TiterMax Gold, Sigma) and injected into rabbits. The procedure was repeated twice every 3 weeks. Blood was collected 2 weeks after the last injection. IgG was purified using a Melon Gel IgG Purification kit (45212, Thermo Fisher) according to the manufacturer's protocol. For WB or immunostaining, the non-phosphorylated KIND2 peptide (GSGSIYSSPGLYSKT) was added to the antibody solution in a final concentration of 5 μg ml⁻¹ to prevent binding to non-phosphorylated KIND2.

**IF staining, live-cell imaging and microscopes.** Cells were seeded on coverslips coated with FN (10 μg ml⁻¹ in PBS), VN (10 μg ml⁻¹ in PBS) or 10% FBS (in PBS) or on FN-coated L-shaped micropatterns (CYTOO). For the majority of immunostainings, cells were washed twice with PBS, fixed with 3% PFA at room temperature (RT) for 15 min, washed twice with PBS and permeabilized with 0.1% Triton X-100 for 10 min. Samples were then blocked with 2% BSA for 1 h, incubated with primary antibodies in 2% BSA for 1 h at RT, washed three times with PBS, incubated with the secondary fluorophore-conjugated antibodies (1:500, Molecular Probes) for 1 h at RT, washed three times with PBS and mounted with Prolong glass antifade mountant (Thermo Fisher).

For staining the mitotic spindle, cells were fixed with cold methanol at −20 °C for 5 min and stained as described above. Images were collected at RT on a Zeiss LSM780 confocal laser scanning microscope or a SR1 Zeiss Elyra PS.1 super-resolution microscope equipped with a Zeiss Plan-APO ×63 or ×100 numerical aperture 1.46 oil-immersion objective.

For TIRF microscopy, cells were grown in a 35-mm glass bottom dish (Ibidi) coated with FN or VN. Live cells or immune-stained cells were imaged using a SR1 Zeiss Elyra PS.1 equipped with a Zeiss Plan-APO ×63 numerical aperture 1.46 oil-immersion objective in the TIRF mode.

For optical sectioning of mitotic cell bodies, cells were seeded on a FN-coated 35-mm glass bottom dish, treated with RO3306 for 16 h and released after 30 min from RO3306. Z-stack images with 0.2-μm intervals were collected from the Z-position of retraction fibres with a Zeiss Elyra PS.1 microscope equipped with a Zeiss Plan-APO ×100-numerical aperture 1.46 oil-immersion objective. Three to five images were stacked with maximum projection to show a section of the mitotic cells.

For quantifying KIND2-pS181 levels, HeLa cells were seeded on FN-coated coverslips, synchronized with D-THY, fixed at indicated the cell cycle phases after D-THY release and immunostained with anti-KIND2-pS181 antibody, phalloidin and DAPI and imaged on a Zeiss LSM780 confocal laser scanning microscope.

Z-sections covering the cell body were stacked with the projection type Sum Slices, and the anti-KIND2-pS181 signal was measured in Fiji ImageJ.

For monitoring mitotic rounding, cells were seeded on FN- or VN-coated 12-well plates (Falcon) and imaged on an EVOS FL Auto2 microscope (Thermo Fisher) enclosed within an environment that maintained 37 °C and 5% humidified $CO_2$. Images were captured every 5 min after RO3306 washout with a ×10 or ×20 objective.

For measuring cell circularity, cells were seeded on FN-coated coverslips, stained with phalloidin and DAPI and imaged on a Zeiss LSM780 confocal laser scanning microscope. The area ($A$) and the perimeter ($P$) of the cell were measured in Fiji ImageJ, and the circularity ($C$) was calculated using the following equation: $C = 4\pi \times A/P2$ (ref. [76]). At least ten cells of each cell line were measured.

Images from the Zeiss LSM780 confocal laser scanning microscope and the SR1 Zeiss Elyra PS.1 super-resolution microscope were acquired using ZEN software (black version).

**High-throughput siRNA screen.** A customized siRNA library targeting enzymes of the ubiquitination system was purchased from Dharmacon. HeLa cells stably expressing EGFP–KIND2-S181D were transfected with the siRNA using Lipofectamine RNAiMAX at a final concentration of 40 nM in 96-well cell carrier plates (Perkin Elmer) according to the manufacturer's protocol. After 48 h of transfection, cells were fixed with 3% PFA, stained with DAPI and imaged in an Opera Phenix High-Content Screening system (Perkin Elmer) with a ×20 objective. At least five random fields of view were collected per well, and the GFP fluorescence intensities of 800–1,200 cells were quantified per well using Harmony (v.4.9).

**Recombinant proteins.** Recombinant KIND2 was expressed and purified as previously described[43]. In brief, KIND2 was cloned into pCoofy17 to add an amino-terminal His10-Sumo tag and expressed in *Escherichia coli* Rosetta at 18 °C overnight. After purification by IMAC in high-salt TBS buffer (20 mM Tris, pH 7.5, 500 mM NaCl and 1 mM TCEP), the Sumo tag was removed by SenP2 (obtained from the in-house facility) digestion overnight at 4 °C. The protein was further purified by SEC using TBS (20 mM Tris, pH 7.5, 200 mM NaCl and 1 mM TCEP) containing 5% glycerol as running and storage buffer. Recombinant CDK1 and cyclin B1 were purchased from Sigma (SRP5009).

**Immunoprecipitation.** For GFP-based immunoprecipitations (IPs), cells were lysed in 10 mM Tris-HCl, 150 mM NaCl, 0.5 mM EDTA and 0.5% NP-40 complemented with protease and phosphatase inhibitors. The supernatant was collected and incubated with GFP nanotrap beads (Chromotek) for 3 h at 4 °C. To elute EGFP–KIND2 and associated proteins, the beads were washed with 10 mM Tris-HCl, 150 mM NaCl and 0.5 mM EDTA three times and boiled in 2× Laemmli buffer at 95 °C for 10 min. The eluted fraction was then run on SDS–PAGE followed by WB or MS analysis.

For antibody-based IPs, cells were lysed as described above, incubated with the respective antibody overnight at 4 °C, and then incubated with protein A/G agarose beads (Santa Cruz sc-2003) for 3 h at 4 °C. The beads were collected, washed and eluted as mentioned above.

**Kinase assay.** In a 50 µl reaction system, 2 µg of recombinant KIND2 was incubated with 100 ng recombinant CDK1–cyclin B1 in the presence of 1 mM ATP, 50 mM Tris-HCl, 10 mM $MgCl_2$, 0.1 mM EDTA and 2 mM DTT (pH 7.5) for 1 h at 37 °C. Subsequently, EDTA was added to final concentration of 20 mM to terminate the reaction.

**Ubiquitination assay.** HeLa cells expressing EGFP–KIND2 were transfected with pRK-HA-Ub-WT and cultured for 48 h. Subsequently, interphase or STLC-arrested cells were treated with 25 µM MG132 for 2 h and 50 µM N-ethylmaleimide for 30 min and lysed. EGFP–KIND2 was immunoprecipitated as described above and analysed by WB. Ubiquitin was detected with anti-HA antibodies.

For the in vitro ubiquitination assay, 10 µg of recombinant KIND2 or recombinant KIND2 phosphorylated by recombinant CDK1–cyclin B1 as described above was incubated in 50 µl ubiquitination buffer (Abcam, ab139472) containing 2.5 µM ubiquitin (WT ab139472 or K48R ab207972, Abcam), 100 nM E1, 2.5 µM E2 (Abcam, ab139472), 5 mM Mg-ATP and CUL9–FBXL10 complex immunoprecipitated from WT HeLa cells at 37 °C for 1 h. The reactions were quenched by adding 50 µl 2× Laemmli buffer, boiled at 95 °C for 10 min, resolved by SDS–PAGE and analysed by WB with anti-KIND2 antibodies.

**Phosphoproteomics analysis.** Cell pellets were incubated in 700 µl of lysis buffer (8 M urea, 30 mM 2-chloroacetamide (Sigma-Aldrich), 5 mM Tris (2-carboxyethyl) phosphine (TCEP; PierceTM, Thermo Fisher Scientific), 1% benzonase, 1 mM $MgCl_2$ in 100 mM Tris, pH 8.0) at 37 °C for 10 min. The lysates were sonicated using a Bioruptor Plus sonication system (Diogenode) for 10 × 30 s at high intensity and 10 × 30 s pauses and incubated at 37 °C for 10 min and sonicated once. The samples were diluted 1:4 with MS-grade water (VWR) and digested for 2 h at 37 °C with 1 µg of LysC and overnight at 37 °C with 3 µg trypsin (Promega). The peptide solution was then acidified with trifluoroacetic acid (Merck) to a final

concentration of 1%, followed by desalting via Sep-Pak C18 1cc vacuum cartridges (Waters). The cartridge was washed twice with 1 ml of acetonitrile and twice with 1 ml of 0.1 M acetic acid before sample loading. The sample was eluted with 0.5 ml 80% (v/v) acetonitrile and 0.1 M acetic acid in Milli-Q $H_2O$ and vacuum dried. Phosphorylated peptides were enriched with Fe (III)-NTA cartridges (Agilent Technologies) using the AssayMAP Bravo Platform (Agilent Technologies) as previously described[77,78].

**LC–MS/MS data acquisition and data analysis.** Liquid chromatography–tandem MS (LC–MS/MS) data were acquired using a Thermo Easy-nLC (Thermo Fisher Scientific) coupled to an Orbitrap mass spectrometer Q Exactive HF (Thermo Fisher Scientific). Peptides were separated on a 30-cm column (inner diameter: 75 µm; packed in-house with ReproSil-Pur C18-AQ 1.9-µm beads, Dr Maisch GmbH) at 60 °C. As gradient, the following steps were applied with increasing addition of buffer B (80% acetonitrile, 0.1% formic acid): linear increase from 30% over 120 min followed a linear increase to 60% over 10 min then to 95% over the next 5 min and finally buffer B was maintained at 95% for another 5 min. The mass spectrometer was operated in data-dependent mode with survey scans from 300 to 1,750 $m/z$ (resolution of 60,000 at $m/z = 200$), and up to 15 of the top precursors were selected and fragmented using higher energy collisional dissociation (HCD with a normalized collision energy of value of 28). The MS2 spectra were recorded at a resolution of 30,000 (at $m/z = 200$) for detection of phosphorylation. For total proteome measurements, the resolution was set to 15,000 (at $m/z = 200$). The AGC targets for MS and MS2 scans were set to $3 \times 10^6$ and $1 \times 10^5$, with a maximum injection time of 20 ms for MS1 and 50 ms for MS2 scans for the detection of phosphorylation and to 100 ms for MS1 and 25 ms for MS2 for total proteome measurements. Dynamic exclusion was set to 16 ms.

Raw data were processed using the MaxQuant computational platform[79] (v.1.6.7.0) with standard settings. The peak list was searched against the human proteome database (75,069 entries) with an allowed precursor mass deviation of 4.5 ppm and an allowed fragment mass deviation of 20 ppm. Cysteine carbamidomethylation was set as a static modification, and oxidation (on methionine) and acetylation (N-terminal) as variable modifications. Additionally, for the detection of phosphorylated and ubiquitinylated peptides, phosphorylation (on serine, threonine and tyrosine) and the GlyGly motif (on lysines) was set as a variable modification. Quantification of proteins across samples was achieved through the label-free quantification (LFQ) algorithm in MaxQuant.

**Statistics and reproducibility.** Statistical analyses were performed using GraphPad Prism v.6 (GraphPad Software). For comparison between two conditions, datasets were analysed with unpaired $t$-test. In multiple comparisons, datasets were first analysed by D'Agostino and Pearson omnibus normality test and further analysed by one-way analysis of variance (ANOVA) Dunnett's test if the normality test was passed or by Kruskal–Wallis test if the normality test was not passed. Results are expressed as the mean ± s.d. Box and whiskers plots in Figs. 3h, 4h, 5b and 6b,m, and Extended Data Figs. 4f and 6h show the median, the first and third quantiles as the box, and whiskers represent minimum to maximum values. The microscopy experiments in Figs. 2a–e, 3f,g, 4f and 5d, and Extended Data Figs. 1l, 2a–m, 3e and 6c,l,j were performed at least three times with similar results obtained. Experiments for quantifications were also carried out at least three times.

**Reporting Summary.** Further information on research design is available in the Nature Research Reporting Summary linked to this article.

## Data availability

Raw data of the proteomics analysis that support the findings of this study have been deposited in the ProteomeXchange Consortium repository under accession number PXD031829. Uncropped blots and statistical data of Figs. 1e, 2h, 3h, 4b,d,g–i,k,l, 5b,c,f, 6b,d,f and 7c,d,e, and Extended Data Figs. 1c, 3c,d, 4c,e f h k,n, 5b,d,f,h,i,k and 6e,g,h are provided in the source data. All other data supporting the finding of this study are available from the corresponding author on request. Source data are provided with this paper.

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

## Acknowledgements

We thank W. Ding for expert technical help; B. Steigenberger, M. Spitaler and G. Cardone for their support with MS and imaging experiments; J. Rech for help with the siRNA library screen; P. Krenn for cell sorting; G. Wang for help with statistics; and W. Zachariae and A. Sonnenberg for discussions, advice and carefully reading the manuscript. The work was supported by the ERC (grant agreement no. 810104 - Point), the DFG (SFB-863) and the Max Planck Society.

## Author contributions

N.-P.C. and R.F. conceived the study, designed the experiments and analysed the data. N.-P.C. performed the experiments. J.A. purified recombinant kindlin and performed the antibody ELISA. N.-P.C. and R.F. wrote the manuscript with input from J.A.

## Funding

## Competing interests

The authors declare no competing interests.

## Additional information

**Extended data** is available for this paper at https://doi.org/10.1038/s41556-022-00886-z.

**Correspondence and requests for materials** should be addressed to Nan-Peng Chen or Reinhard Fässler.

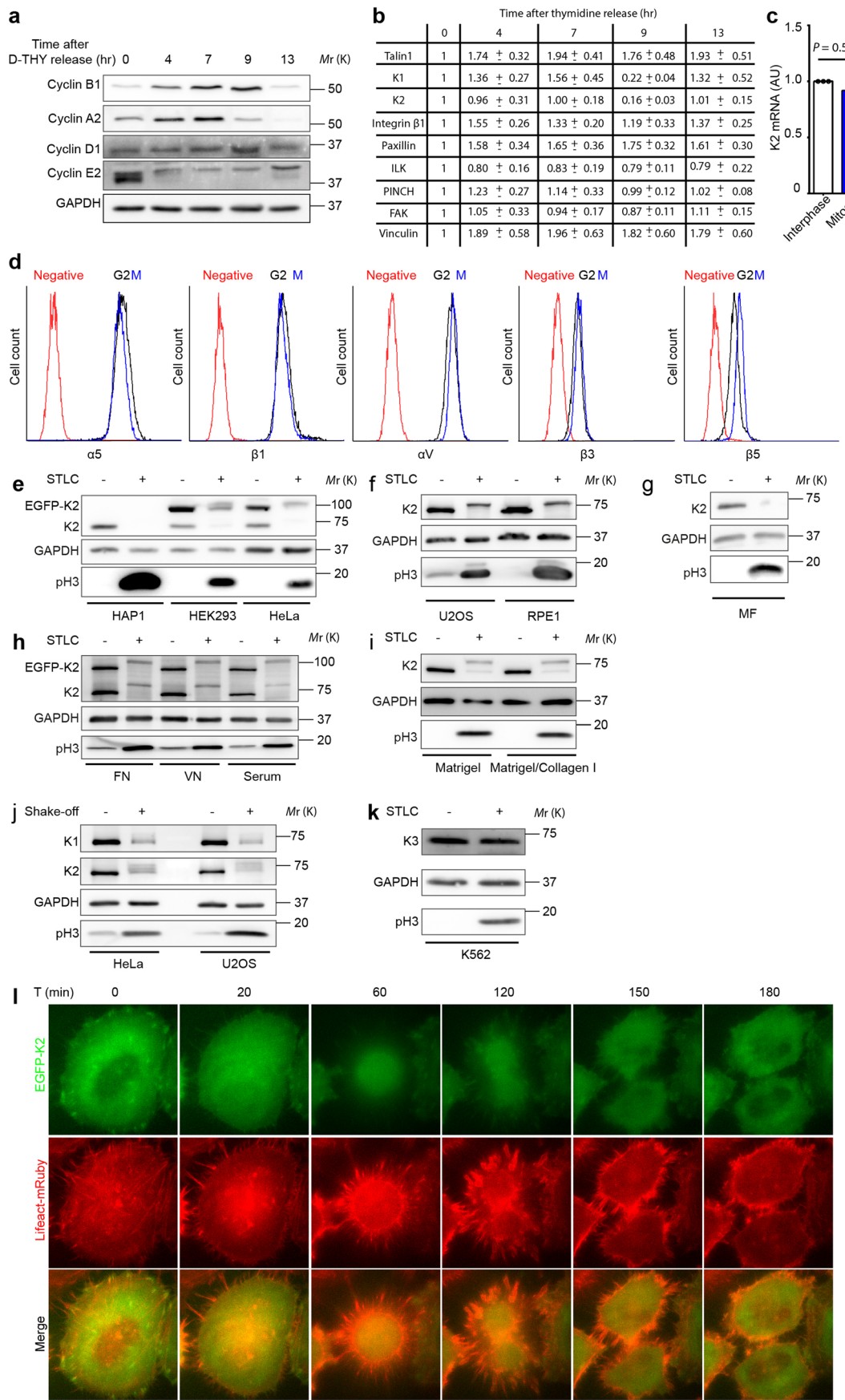

**Extended Data Fig. 1 | See next page for caption.**

**Extended Data Fig. 1 | Kindlin-1 (K1) and kindlin-2 (K2) are degraded in mitosis, related to Fig. 1. (a)** WB of indicated cyclins in HeLa cells released from D-THY treatment and harvested at indicated times. **(b)** Relative changes in GAPDH-normalized FA protein abundance in HeLa cells released from D-THY, harvested at indicated times and determined by WB. Mean ± SD; n = 3 independent experiments. Abundance of FA proteins was referenced to 0 hr after D-THY release.**(c)** K2 mRNA levels in D-THY-synchronized G2 phase (7 hr after release) and M-phase (9 hr after release combined with shake-off) HeLa cells quantified by RT-PCR. Mean ± SD; n = 3 independent experiments. ns: not significant by unpaired Student's t-test (two-tailed). **(d)** Surface levels of integrin subunits at G2 (black, 7 hr after D-THY release) and M-phase (red, collected by mitotic shake-off 9 hr after D-THY release) in HeLa cells determined by flow cytometry. The experiment has been repeated three times with similar results. (e-i) WB of EGFP-K2 and/or endogenous K2 with anti-K2 antibody in untreated and STLC-arrested mitotic HAP1, HEK293 and HeLa **(e)**, U2OS and RPE1 **(f)**, mouse fibroblast (MF; **g**) and HeLa cells seeded on **(h)** FN, VN, 10% FCS and **(i)** embedded in a 3D Matrigel or Matrigel/collagen I mixture (1:1). **(j)** WB of K1 and K2 in unsynchronized HeLa and U2OS cells harvested with and without mitotic shake-off. **(k)** WB of K3 in untreated and STLC-arrested mitotic K562 cells. **(l)** Montages of time-lapse live cell recordings generated by wide-field fluorescent microscopy showing the relative fluorescence of EGFP-K2 and lifeact-mRuby in HeLa cell at indicated times after RO3306 release. Scale bar, 10 μm. All WB experiments have been repeated three times with similar results. pH3 served as indicator for mitosis and GAPDH as loading control.

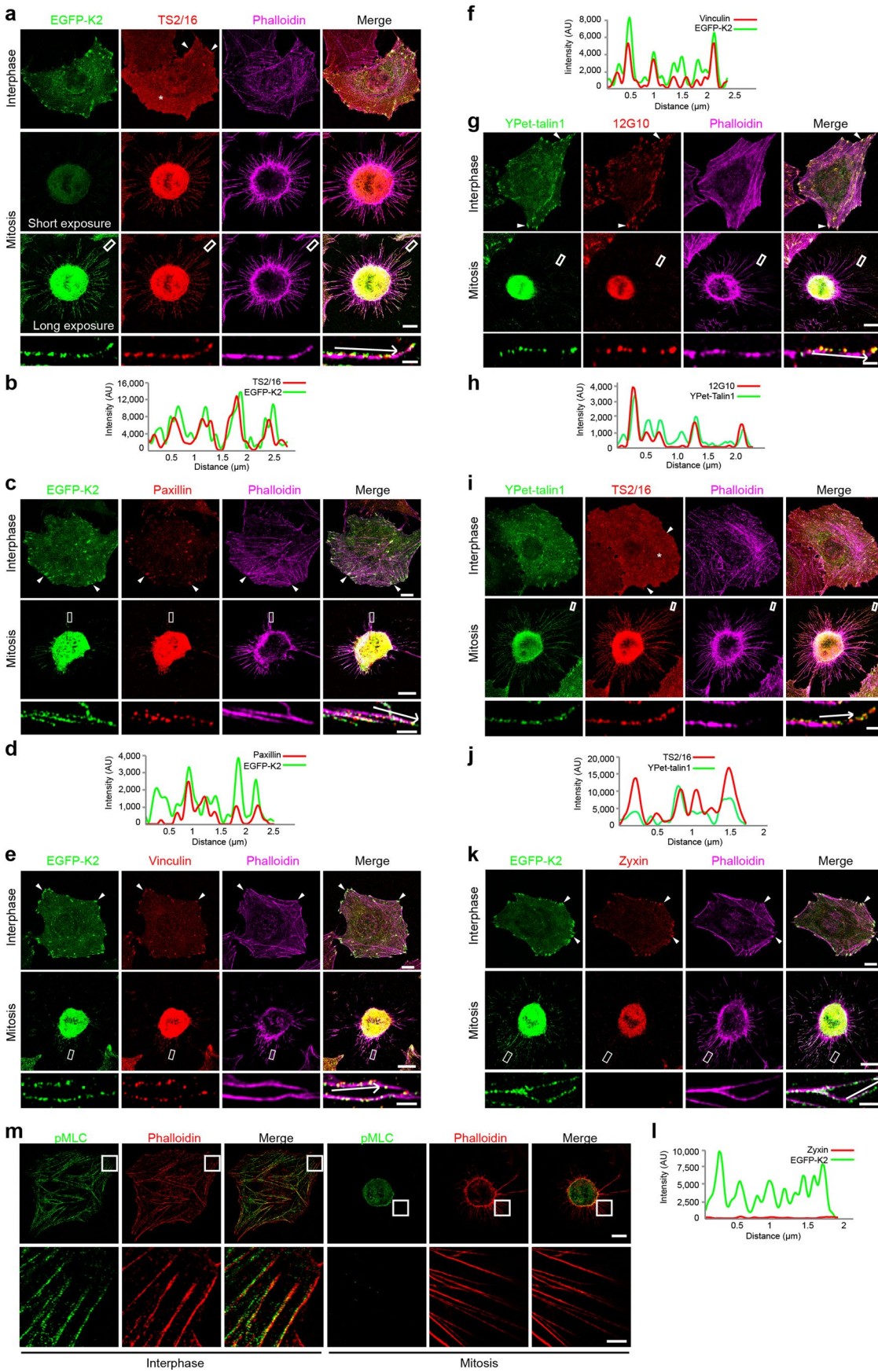

**Extended Data Fig. 2 | See next page for caption.**

**Extended Data Fig. 2 | Distribution of FA proteins in mitotic adhesion sites, related to Fig. 2.** (**a**–**f**) SIM images of EGFP-K2 (green) and phalloidin (magenta) in combination with (**a**) TS2/16-stained β1 integrin (red), (**c**) paxillin (red) or (**e**) vinculin (red) in interphase and mitotic HeLa cells (30 min after RO3306 release). Arrowheads indicate FAs and * plasma membrane. EGFP-K2 signals in retraction fibers (**a**) were enhanced by increasing exposure time. Boxes (**a,c,e**) indicate areas of retraction fibers shown at high magnifications. The arrows indicate direction of line profile analysis of (**b**) EGFP-K2 (green) and TS2/16 (red), (**d**) EGFP-K2 (green) and paxillin (red) and (**f**) EGFP-K2 (green) and vinculin (red). Scale bar, 10 μm. Scale bar of magnifications showing retraction fibers, 1 μm. (**g**–**j**) SIM images of YPet-talin1 (green) and phalloidin (magenta) in combination with (**g**) 12G10 (red) or (**i**) TS2/16 β1 integrin (red) in interphase and mitotic (30 min after RO3306 release) HeLa cells. Arrowheads (**g,i**) indicate FAs and *(**i**) plasma membrane. Boxes indicate areas of retraction fibers shown at high magnifications. The arrows indicate direction of line profile analysis of YPet-talin1 (green) together with (**h**) 12G10 (red) or (**j**) TS2/16 (red). Scale bar, 10 μm. Scale bar of magnifications showing retraction fibers, 1 μm. (**k,l**) SIM images of EGFP-K2 (green), zyxin (red) and phalloidin (magenta) in interphase and mitotic (30 min after RO3306 release) HeLa cells. Arrowheads indicate FAs. Boxes indicate areas of retraction fibers shown at high magnifications. The arrow indicates direction of line profile of (**l**) EGFP-K2 (green) and zyxin (red). Scale bar, 10 μm. Scale bar of magnifications showing retraction fibers, 1 μm. (**m**) SIM images of pMLC (green) and phalloidin (red) in interphase and mitotic (30 min after RO3306 release) HeLa cells. Boxes indicate areas of stress fibers of interphase cells and retraction fibers of mitotic cells shown at high magnifications. Scale bar, 10 μm. Scale bar of magnifications showing stress fibers and retraction fibers, 2 μm. Each line profile assessment has been done three times.

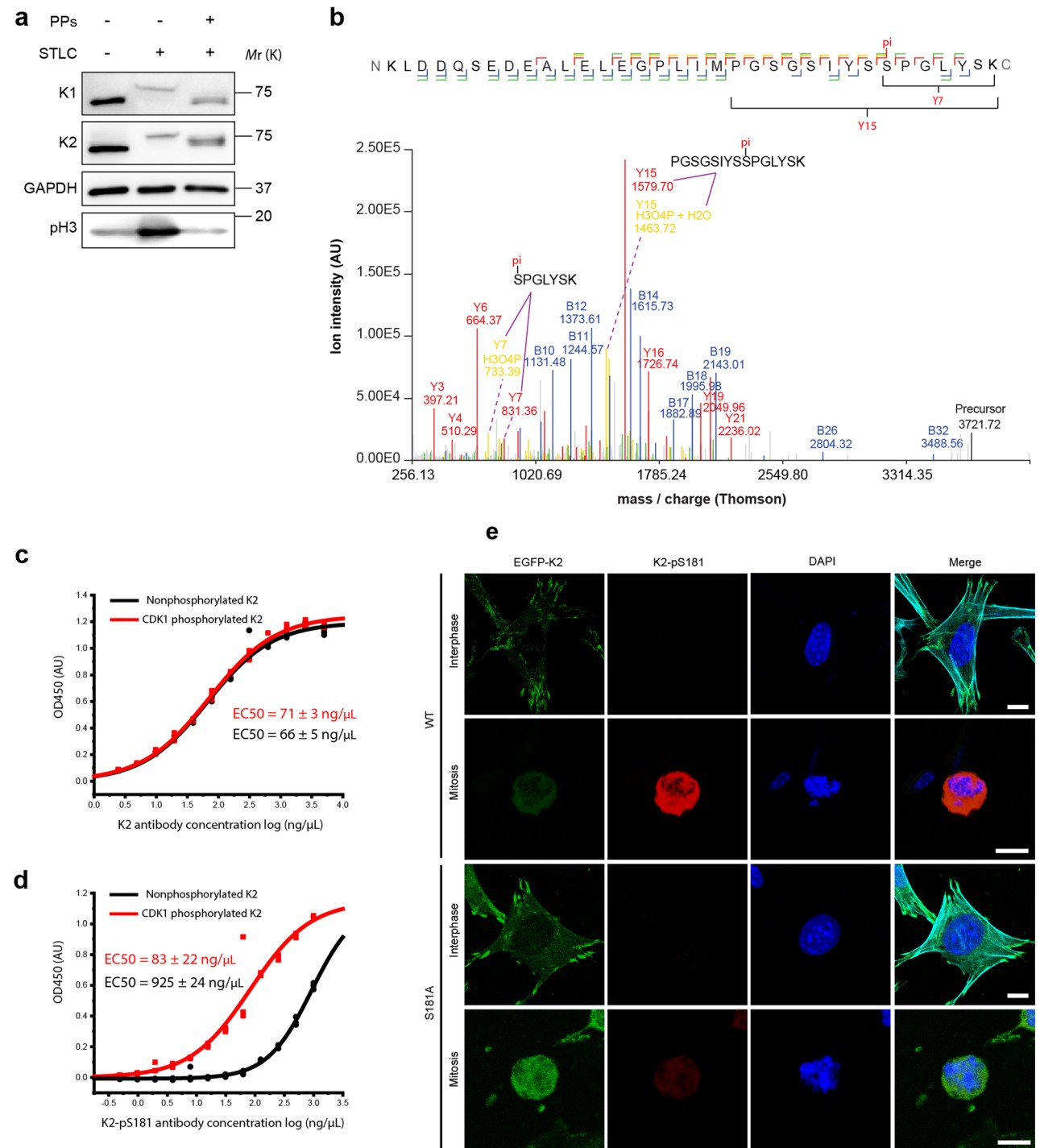

**Extended Data Fig. 3 | Validation of K1 and K2 phosphorylation in mitosis, related to Fig. 3. (a)** WB of K1 and K2 in untreated and STLC-arrested mitotic HeLa cells with and without alkaline- and lambda-phosphatases (PPs) treatment. pH3 served as indicator of mitosis and successful phosphatase treatment, and GAPDH as loading control. The experiment has been repeated three times with similar results. **(b)** MS2 spectrum acquired under HCD fragmentation conditions for the phosphorylated peptide (KKLDDQSEDEALELEGPLIMPGSGSIYSSPGLYSK) from CDK1-cyclin B1 phosphorylated recombinant K2. Full sequence coverage for the peptide achieved by combining b- and y-ion series (blue = b-ion, red = y-ion, green = b- or y-ion with neutral loss of H2O or NH3, yellow = b- or y-ion with neutral loss of H3O4P+H2O). The ions series y7 to y21, y23, y25 and the ion b31 and b32 include mass of phosphorylation. The y7-ion identifies S181 as the phosphorylated residue. (**c,d**) ELISA of immobilized K2 (black) and *in vitro* CDK1 cyclin-B1 phosphorylated K2 (red) titrated with (**c**) commercial anti-K2 (clone 3A3) and **d**) home-made anti-K2-pS181 antibody. n = 3 independent experiments. **(e)** Representative confocal images of EGFP-K2 (green), K2-pS181 (red) and DAPI (blue) in interphase and mitotic (30 min after RO3306 release) mouse fibroblasts expressing EGFP-tagged K2-WT or -S181A. Scale bar = 20 µm.

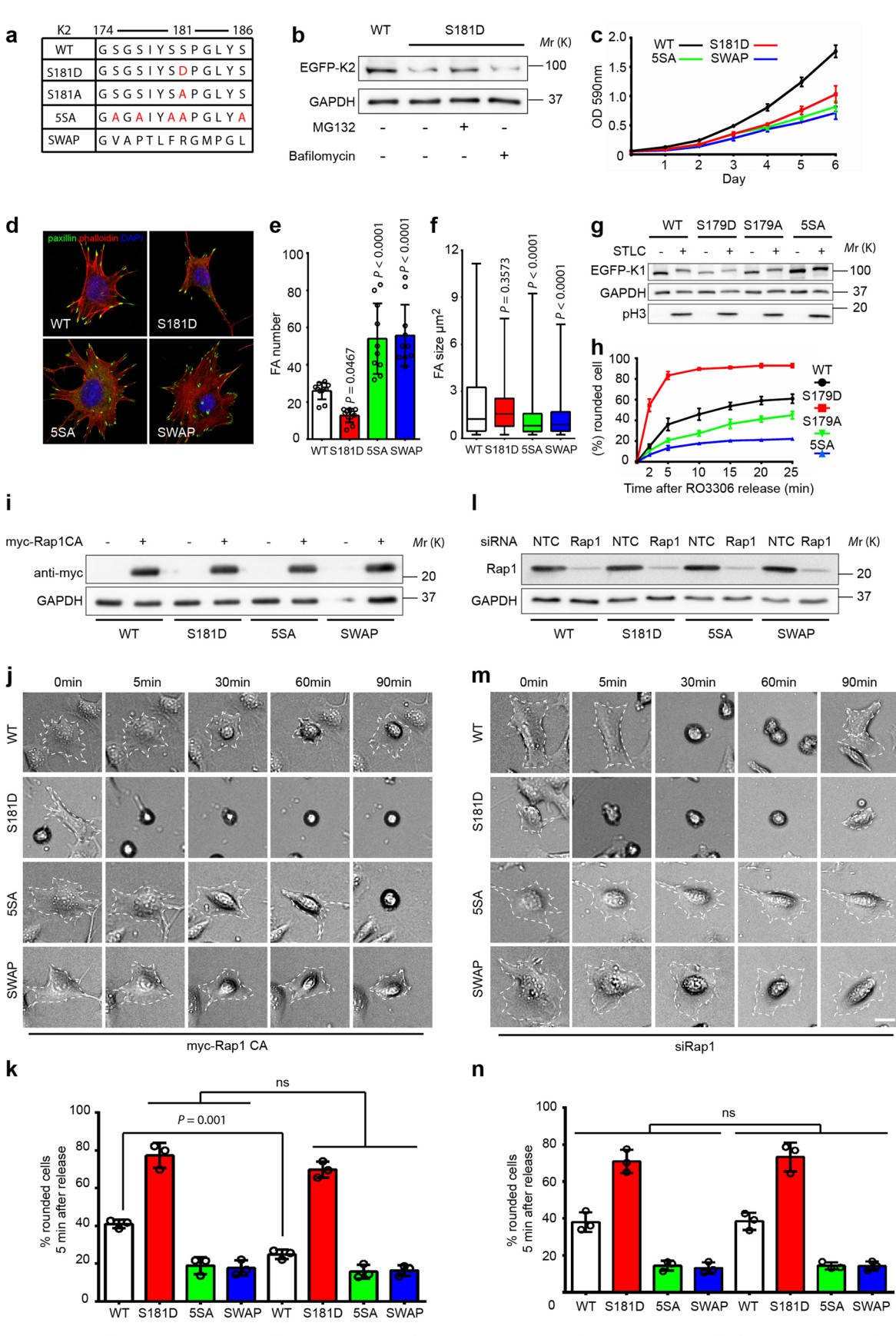

**Extended Data Fig. 4 | See next page for caption.**

**Extended Data Fig. 4 | K2 degradation induced by CDK1-cyclin B1-mediated phosphorylation, related to Fig. 4. (a)** Amino acid sequences from position 174 to 186 of K2-WT, K2-S181D, K2-S181A, K2-5SA and K2-SWAP. Amino acid substitutions are shown in red. **(b)** WB with anti-GFP antibody of untreated EGFP-K2-WT or K2-S181D expressing HeLa cells treated with MG132 (5 μM for 6 hr) or Bafilomycin A1 (1 μM for 6 hr). **(c)** MTT assay showing proliferation of mouse fibroblasts expressing EGFP-tagged K2-WT, K2-S181D, K2-5SA or K2-SWAP. Mean ± SD, n=3 independent experiments. **(d)** Representative confocal images of immunostained paxillin (green), phalloidin (red) and DAPI (blue) in FN-seeded interphase mouse fibroblasts expressing EGFP-tagged K2-WT, K2-S181D, K2-5SA or K2-SWAP. Scale bar, 10 μm. **(e)** Quantification of paxillin-positive FA numbers of FN-seeded interphase mouse fibroblasts expressing EGFP-tagged K2-WT, K2-S181D, K2-5SA or K2-SWAP. Mean ± SD; n = 11 cells for WT and S181D, 10 cells for 5SA and SWAP pooled from 3 independent experiments. **(f)** Quantification of paxillin-positive FA sizes in FN-seeded interphase mouse fibroblasts expressing EGFP-tagged K2-WT, K2-S181D, K2-5SA or K2-SWAP. n (FA number): 294 for K2-WT, 108 for K2-S181D, 426 for K2-5SA and 431 for K2-SWAP pooled from 3 independent experiments. **(g)** WB of GFP in untreated and STLC-arrested HeLa cell expressing EGFP-tagged K1-WT, K1-S179D, K1-S179A or K1-5SA. **(h)** Percentages of rounded mouse fibroblasts expressing EGFP-tagged K1-WT, K1-S179D, K1-S179A or K2-5SA quantified after RO3306 release. Mean ± SD; 200 cells counted for each cell line, n = 3 independent experiments. **(i)** WB with anti-myc antibody of mouse fibroblasts expressing EGFP-tagged K2-WT, K2-S181D, K2-5SA or K2-SWAP and retrovirally transduced with myc-tagged Rap1-G12V (myc-Rap1-CA). **(j)** Montage of phase contrast recordings from FN-seeded mouse fibroblasts expressing EGFP-tagged K2-WT, K2-S181D, K2-5SA or K2-SWAP and retrovirally transduced with myc-Rap1-CA. Time (min) starts at RO3306 release. Scale bar, 20 μm. **(k)** Percentages of rounded mouse fibroblasts expressing EGFP-tagged K2-WT, K2-S181D, K2-5SA or K2-SWAP and retrovirally transduced with myc-Rap1-CA quantified 5 min after RO3306 release. Mean ± SD; 200 cells counted for each cell line, n = 3 independent experiments. ns: not significant. **(l)** WB with anti-Rap1 antibody recognizing both Rap1A and Rap1B isoforms of mouse fibroblasts expressing EGFP-tagged K2-WT, K2-S181D, K2-5SA or K2-SWAP and transfected with non-targeting control (siNTC) siRNA or siRNAs targeting Rap1A and Rap1B mRNAs (siRap1). Cells were harvested 48 hr after transfection. **(m)** Montage of phase contrast recordings of FN-seeded mouse fibroblasts expressing EGFP-tagged K2-WT, K2-S181D, K2-5SA or K2-SWAP and transfected with siRap1. 32 hr after siRap1 transfection, cells were synchronized with RO3306 for 16 hr and then released for imaging. Time (min) starts at RO3306 release. Scale bar, 20 μm. **(n)** Percentages of rounded mouse fibroblasts expressing EGFP-tagged K2-WT, K2-S181D, K2-5SA or K2-SWAP and transfected with siRap1, and quantified 5 min after RO3306 release. Mean ± SD; 200 cells counted for each cell line, n = 3 independent experiments. ns: not significant. P values in (e, f, k, n) calculated by one-way ANOVA Dunnett's multiple comparison test (95% CI). All WB experiments have been repeated at least three times with similar results. pH3 served as an indicator for mitosis and GAPDH as loading control.

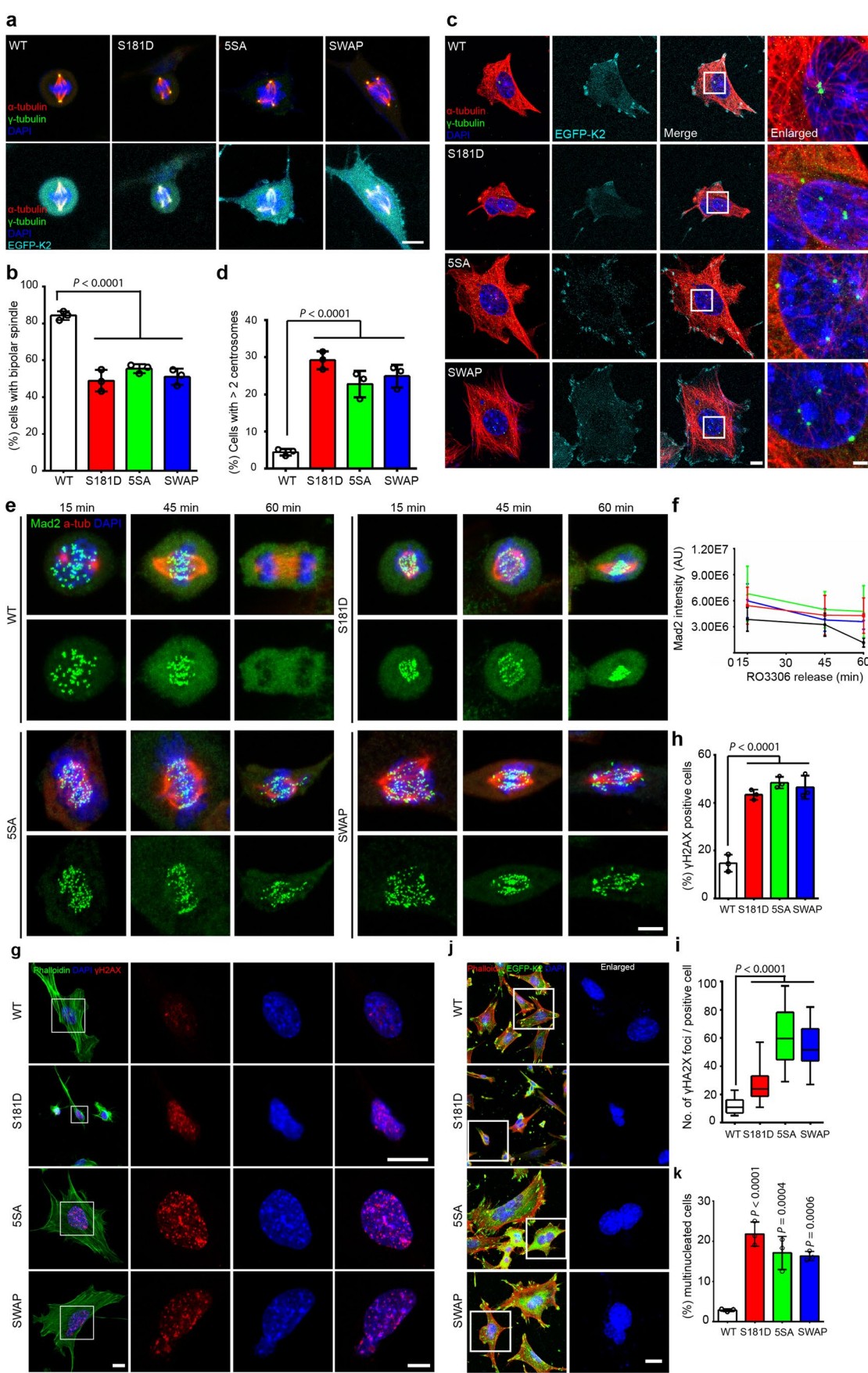

**Extended Data Fig. 5 | See next page for caption.**

**Extended Data Fig. 5 | Consequences of K2 phospho-mutants, related to Fig. 5.** (**a-d**) Representative confocal images of α-tubulin (red), γ-tubulin (green), DAPI (blue) and EGFP-K2 (cyan) in (**a**) mitotic (30 min after RO3306 release) and (**c**) interphase mouse fibroblasts expressing EGFP-tagged K2-WT, K2-S181D, K2-5SA or K2-SWAP. Boxes indicate areas of centrosomes shown at high magnifications. Quantifications of cells (mean ± SD; 100 cells were measured for each cell line, n = 3 independent experiments) with bipolar (**b**) and supernumerary centrosomes (**d**). Scale bar, 10 μm. Scale bar of magnifications showing centrosomes, 2.5 μm. (**e**) Representative confocal images of immunostained Mad2 (green), α-tubulin (red) and DAPI (blue) in mouse fibroblasts expressing EGFP-tagged K2-WT, K2-S181D, K2-5SA or K2-SWAP at indicated times after RO3306 release. Scale bar, 5 μm. (**f**) Quantification of the Mad2 fluorescence intensity in mouse fibroblasts expressing EGFP-tagged K2-WT K2-S181D, K2-5SA or K2-SWAP at indicated time points after RO3306 release. Mean ± SD; n (cell number) = 20 for WT, 13 for S181D, 10 for 5SA and 11 for SWAP at 15 min; 12 for WT, 16 for S181D, 10 for 5SA and 12 for SWAP at 45 min; 10 for WT, 12 for S181D, 10 for 5SA and 11 for SWAP at 60 min pooled from 3 independent experiment. (**g**) Representative confocal images of γH2AX (red), phalloidin (green) and DAPI (blue) in mouse fibroblasts expressing EGFP-tagged K2-WT, K2-S181D, K2-5SA or K2-SWAP. Boxes indicate nuclear areas shown at high magnifications. Scale bar: 10 μm (left), 5 μm (right). Note that nuclei of K2-S181D-expressing fibroblasts are smaller. (**h**) Percentages of γH2AX-positive cells with more than 5 foci in mouse fibroblasts expressing EGFP-tagged K2-WT, K2-S181D, K2-5SA or K2-SWAP. Mean ± SD; 200 cells quantified for each cell line, n = 3 independent experiments. (**i**) Quantification of the number of γH2AX-foci in foci-positive mouse fibroblasts expressing EGFP-tagged K2-WT, K2-S181D, K2-5SA or K2-SWAP. Mean ± SD, n = 50 cells pooled from 3 independent experiments. (**j**) Representative confocal images of phalloidin (red) and DAPI (blue) in mouse fibroblasts expressing EGFP-tagged K2-WT K2-S181D, K2-5SA or K2-SWAP. Boxes indicate areas of polyploid cells shown as DAPI at high magnifications. Scale bar, 10 μm. Scale bar of magnifications showing polyploid cells, 5 μm. (**k**) Percentages of polyploid mouse fibroblasts expressing EGFP-tagged K2-WT K2-S181D, K2-5SA or K2-SWAP. Mean ± SD; 200 cells were counted in each cell line, n = 3 independent experiments. P values in (b, d, f, h, i, k) calculated by one-way ANOVA Dunnett's multiple comparison test (95% CI).

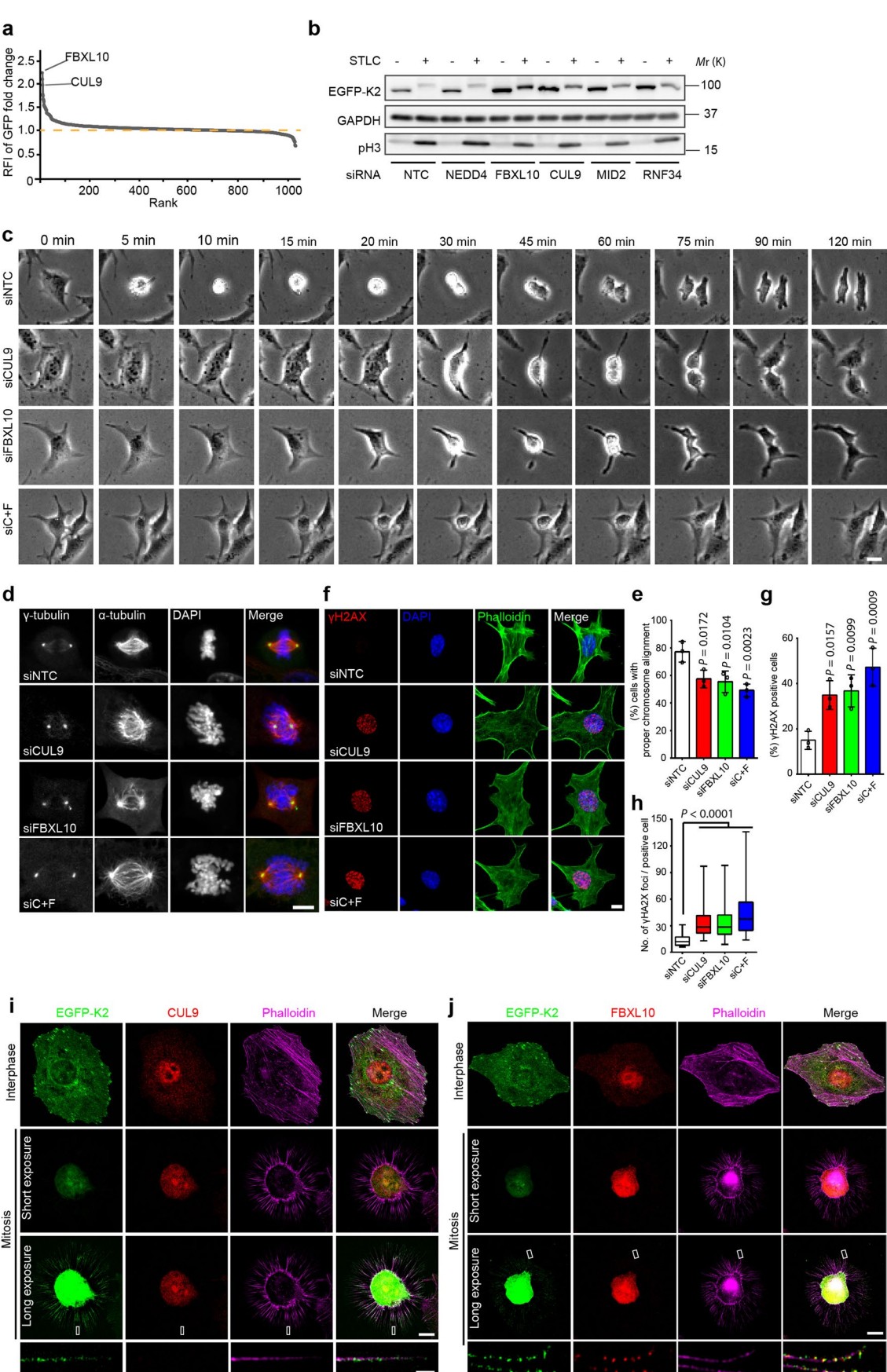

**Extended Data Fig. 6 | See next page for caption.**

**Extended Data Fig. 6 | Consequences of CUL9 and/or FBXL10 depletion, related to Figs. 6 and 7. (a)** Scatter plot based on fold change of EGFP fluorescent intensity in individual transfections of the siRNA screening. EGFP intensity of EGFP-K2-S181D expressing HeLa cells transfected with siNTC was set as 1 and used as reference for quantifications. 800–1200 cells were quantified in each well. CUL9 and FBXL10 genes are indicated. **(b)** WB with anti-GFP antibody of untreated and STLC-arrested mitotic HeLa cells expressing EGFP-tagged K2 and transfected with indicated siRNAs. pH3 served as an indicator for mitosis and GAPDH as loading control. The experiment has been repeated three times with similar results. **(c)** Montage of phase contrast recordings of FN-seeded mouse fibroblasts transfected with siNTC, siCUL9, siFBXL10 or siC+F. Cells were synchronized 32 hr after transfection with RO3306 for 16 hr and then released for imaging. Time (in min) starts at RO3306 release. Dashed outlines indicate cell margins. Scale bar, 20 μm. **(d)** Representative confocal images of α-tubulin (red), γ-tubulin (green) and DAPI (blue) in mitotic (30 min after RO3306 release) mouse fibroblasts transfected with siNTC, siCUL9, siFBXL10, or siC+F. Scale bar, 5 μm. **(e)** Percentages of mitotic (30 min after RO3306 release) mouse fibroblasts transfected with siNTC, siCUL9, siFBXL10 or siC+F with normal chromosome alignments. Mean ± SD; 50 cells were measured in each transfection, n = 3 independent experiments. **(f)** Representative confocal images of γH2AX (red), phallodin (green) and DAPI (blue) in interphase mouse fibroblasts transfected with siNTC, siCUL9, siFBXL10 or siC+F. Scale bar, 10 μm. **(g)** Percentages of γH2AX-positive cells with more than 5 foci in mouse fibroblasts transfected with siNTC, siCUL9, siFBXL10 or siC+F. Mean ± SD; 200 cells were quantified in each transfection, n = 3 independent experiments. **(h)** Quantification of number of γH2AX-foci in foci-positive mouse fibroblasts transfected with siNTC, siCUL9, siFBXL10 or siC+F. n = 40 cells for each transfection pooled from 3 independent experiments. **(i,j)** SIM images of EGFP-K2 (green), phalloidin (magenta) and **(i)** CUL9 (red) or **(j)** FBXL10 (red) in interphase and mitotic (30 min after RO3306 release) HeLa cells. Localization of K2 in mitotic retraction fibers was visualized by increasing the exposure time. Boxes indicate areas of retraction fibers shown at high magnifications. Scale bar, 10 μm; retraction fibers, 1 μm. P values in (e, g, h) calculated by one-way ANOVA Dunnett's multiple comparison test (95% CI).

|---|---|

# Reporting Summary

## Statistics

For all statistical analyses, confirm that the following items are present in the figure legend, table legend, main text, or Methods section.

| n/a | Confirmed | |
|---|---|---|
| ☐ | ☒ | The exact sample size (*n*) for each experimental group/condition, given as a discrete number and unit of measurement |
| ☐ | ☒ | A statement on whether measurements were taken from distinct samples or whether the same sample was measured repeatedly |
| ☐ | ☒ | The statistical test(s) used AND whether they are one- or two-sided<br>*Only common tests should be described solely by name; describe more complex techniques in the Methods section.* |
| ☒ | ☐ | A description of all covariates tested |
| ☐ | ☒ | A description of any assumptions or corrections, such as tests of normality and adjustment for multiple comparisons |
| ☐ | ☒ | A full description of the statistical parameters including central tendency (e.g. means) or other basic estimates (e.g. regression coefficient) AND variation (e.g. standard deviation) or associated estimates of uncertainty (e.g. confidence intervals) |
| ☐ | ☒ | For null hypothesis testing, the test statistic (e.g. *F*, *t*, *r*) with confidence intervals, effect sizes, degrees of freedom and *P* value noted<br>*Give P values as exact values whenever suitable.* |
| ☒ | ☐ | For Bayesian analysis, information on the choice of priors and Markov chain Monte Carlo settings |
| ☒ | ☐ | For hierarchical and complex designs, identification of the appropriate level for tests and full reporting of outcomes |
| ☒ | ☐ | Estimates of effect sizes (e.g. Cohen's *d*, Pearson's *r*), indicating how they were calculated |

*Our web collection on statistics for biologists contains articles on many of the points above.*

## Software and code

Policy information about availability of computer code

| Data collection | ZEN (Zeiss) for image collection on LSM780 confocal microscope (version 2.3 SP1 FP3, black) and SR1 Elyra PS.1 super-resolution microscope (version 2.1 SP1 FP3, black).<br>Visisystem (Visitron Systems) for image collection on wide field and TIRF microscope (Zeiss).<br>EVOS AutoFL system (Invitrogene) for image collection on EVOS live-cell microscope.<br>Harmony 4.9(Perkin Elmer) for image collection on Opera Phenix Plus High-Content Screening System.<br>FACsDIVA (BD, version 9.0) for data collection on LSRFortessa™ X-20 Cell Analyzer.<br>ImageQuant LAS 4000 for image collection of western blots on Fujifim LAS-4000<br>SoftMax Pro 7.1 for data collection for MTT and quantification of protein concentration on SpectraMax ABS Plus. |
|---|---|
| Data analysis | GraphPad Prism (version 6.0) was used for statistical analysis.<br>Flowjo (version 8.0) was used to analyze FACs data.<br>Fiji (Image J) was used for quantification of immunoblot bands, immunofluorescence and live-cell imaging signals.<br>Harmony (version 4.9) was used for quantification of high-through put siRNA screening.<br>Maxquant (version 1.6.7.0) was used for processing raw proteomic data.<br>Perseus software (version 1.5.5.3) was used to analyze proteomic data. |

For manuscripts utilizing custom algorithms or software that are central to the research but not yet described in published literature, software must be made available to editors and reviewers. We strongly encourage code deposition in a community repository (e.g. GitHub). See the Nature Portfolio guidelines for submitting code & software for further information.

## Data

Policy information about availability of data

All manuscripts must include a data availability statement. This statement should provide the following information, where applicable:

- Accession codes, unique identifiers, or web links for publicly available datasets
- A description of any restrictions on data availability
- For clinical datasets or third party data, please ensure that the statement adheres to our policy

Raw data of the proteomic analysis that support the findings of this study have been deposited in ProteomeXchange Consortium repository under Accession number PXD031829. Uncropped blots and statistical data of Fig. 1e; 2h; 3h; 4b, d, g, h, i, k, l; 5b, c, f; 6b, d, f; 7c, d, e and Extended Data Fig. 1c; 3c, d; 4c, e, f, h, k, n; 5b, d, f, h, i, k; 6e, g, h are provided in the source data. All other data supporting the finding of this study are available from the corresponding author on request.

# Field-specific reporting

Please select the one below that is the best fit for your research. If you are not sure, read the appropriate sections before making your selection.

☒ Life sciences          ☐ Behavioural & social sciences          ☐ Ecological, evolutionary & environmental sciences

For a reference copy of the document with all sections, see nature.com/documents/nr-reporting-summary-flat.pdf

# Life sciences study design

All studies must disclose on these points even when the disclosure is negative.

| | |
|---|---|
| Sample size | The sample sizes were determined based on experience on the reliable measurement of the experimental results according to the standards in previous publications (Stewart et al, Nature, 2011; Lancaster et al, Dev Cell, 2013; Jones et al, JCB, 2018; Monster et al, JCB, 2021) in the field and sufficient to represent the significance of difference in different conditions. No statistical method was used to predetermine the sample size. |
| Data exclusions | No data was excluded in this study. |
| Replication | The sample size of each experiment and the number of experimental replicates were indicated in figure legends or methods. All experiments were successfully replicated. |
| Randomization | Samples were allocated into groups according to the kindlin 2 mutation the cell line carried, or based on the cell cycle stages revealed by phalloidin and DAPI staining, or based on the different siRNA used in the transfection. Data for comparisons of rounding speed, spindle orientation, retraction fiber length and number, focal adhesion number and size, DNA double strand break and duration of mitotic checkpoint in different kindlin mutant fibroblasts or upon different siRNA depletion were acquired by randomly taken images of cells in the well or on the coverslips. Statistical data for quantification was collected over months or years, thus ensure reproducibility despite being gathered from different passages. |
| Blinding | Image acquisition from different cell lines or at different conditions were not blinded but relied on unbiased data collection from random regions in the wells or coverslips. |

# Reporting for specific materials, systems and methods

We require information from authors about some types of materials, experimental systems and methods used in many studies. Here, indicate whether each material, system or method listed is relevant to your study. If you are not sure if a list item applies to your research, read the appropriate section before selecting a response.

## Materials & experimental systems

| n/a | Involved in the study |
|---|---|
| ☐ | ☒ Antibodies |
| ☐ | ☒ Eukaryotic cell lines |
| ☒ | ☐ Palaeontology and archaeology |
| ☒ | ☐ Animals and other organisms |
| ☒ | ☐ Human research participants |
| ☒ | ☐ Clinical data |
| ☒ | ☐ Dual use research of concern |

## Methods

| n/a | Involved in the study |
|---|---|
| ☒ | ☐ ChIP-seq |
| ☐ | ☒ Flow cytometry |
| ☒ | ☐ MRI-based neuroimaging |

## Antibodies

| | |
|---|---|
| Antibodies used | The following antibodies were used for western blotting (WB), immunofluorescence (IF) and flow cytometry (FC): kindlin-2 (clone |

| Antibodies used | 3A3, MAB2617, mouse, Merck Millipore, 1:1000 for WB; 1:200 for IF); kindlin-2-pS181 (home-made, 1:5000 for WB; 1:400 for IF); kindlin-1 (home-made35, 1:1000 for WB); talin1 (T3287, Sigma, 1:1000 for WB); ILK (3856, rabbit, Cell signaling, 1:1000 for WB); PINCH (612710, mouse, Transduction Laboratories, 1:1000 for WB); vinculin (sc-5573, rabbit, Santa Cruz, 1:1000 for WB; V9131, Sigma, 1:500 for IF); FAK (3285, rabbit, cell signaling, 1:1000 for WB); paxillin (610051, mouse, Transduction Laboratories, 1:1000 for WB, 1:300 for IF); β1 integrin (9699, rabbit, cell signaling, 1:1000 for WB; TS2/16, mouse, Biolegend, 1:200 for IF; 12G10, MAB2247, mouse, Merck Millipore, 1:200 for IF; clone HMbeta1-1, 102207, hamster, BioLegend, 1:200 for FC); β3 integrin (M109-3, rat, Biozol, 1:200 for IF; clone VI-PL02, 336406, mouse, Biolegend, 1:200 for FC); β5 integrin (3629, rabbit, cell signaling, 1:200 for IF. ALULA, gift from Dr. Dean Sheppard, 1:200 for IF; clone KN52, 12-0497, mouse, ebioscience, 1:200 for FC); α5 integrin (clone 5H10-27, 557447, rat, PharMingen, 1:200 for FC); αV integrin (clone NKI-M9, 327909, mouse, Biolegend, 1:200 for FC); zyxin (clone 164D4, 307011, mouse, Synaptic system, 1:200 for IF); phospho-histone H3 (53348, rabbit, cell signaling, 1:1000 for WB; 1:500 for IF); cyclin A2 (4656, mouse, cell signaling, 1:1000 for WB); cyclin B1 (4138, rabbit, cell signaling, 1:1000 for WB); cyclin D1 (2978, rabbit, cell signaling, 1:1000 for WB); cyclin E2 (4132, rabbit, cell signaling, 1:1000 for WB); GAPDH (CB1001, mouse, Calbiochem, 1:10000 for WB); α-tubulin (2144, rabbit, cell signaling, 1:100 for IF); γ-tubulin (T6557, mouse, Sigma, 1:500 for IF); cullin9 (A300-098A, rabbit, bethyl, 1:1000 for WB; 1:200 for IF); FBXL10 (09-864, rabbit, Merck Millipore, 1:1000 for WB; 1:200 for IF); pS139-γH2AX (clone JBW301, 05-636, mouse, Millipore, 1:200 for IF); Mad2 (A300-301A, rabbit, Bethyl Laboratories, 1:200 for IF); hemagglutin (HA)-tag (rat, Roche, 1:1000 for WB); GFP (A10262, chicken IgY, Invitrogen, 1:1000 for WB); GFP-Booster Alexa Fluor 488 (gb2AF488, Chromotek, 1:500 for IF). |
|---|---|
| Validation | The application and dilution are listed above. The references of the used antibodies are the following. kindlin-2 (MAB2617, mouse, Merck Millipore, Theodosiou et al, eLife 2016. WB and IF); talin1 (T3287, Sigma, Theodosiou et al, eLife 2016; WB); ILK (3856, rabbit, Cell signaling, Hussain et al, J Neurosci. 2017. WB and IF); PINCH (612710, mouse, Transduction Laboratories, Donthamsetty et al, PLoS ONE. 2013. WB); vinculin (sc-5573, rabbit, Santa Cruz, O'Connor et al, Cancer research, 2013; V9131, Sigma, Feliciano et al, Nat Commun. 2021. WB and IF); FAK (3285, rabbit, cell signaling, Lundby et al, Cell. 2019. WB); paxillin (610051, mouse, Transduction Laboratories, Freeman et al, Science. 2020; WB and IF); β1 integrin (9699, rabbit, cell signaling, Matsumura et al, Nat Commun. 2016. WB; TS2/16, mouse, Biolegend, Freeman et al, Science. 2020. IF; 12G10, MAB2247, mouse, Merck Millipore, Nader et al, Nat Cell Biol. 2016. IF; clone HMbeta1-1, 102207, hamster, BioLegend, Baker et al, PNAS. 2012. FC); β3 integrin (336406, mouse, Biolegend, Hartmann et al, Cell Rep. 2019. FC); β5 integrin (clone KN52, 12-0497, mouse, ebioscience, Hansen et al, Nat Commun. 2016. FC); α5 integrin (clone 5H10-27, 557447, rat, PharMingen, Kharbili et al, Oncotarget. 2017. FC); αV integrin (clone NKI-M9, 327909, mouse, Biolegend, Debnath et al, Nature. 2018. FC); zyxin (clone 164D4, mouse, Synaptic system, Gill et al, EMBO J. 2015. WB); phospho-histone H3 (53348, rabbit, cell signaling, Ramos et al, Dev Cell, 2020. WB); cyclin A2 (4656, mouse, cell signaling, Bönisch et al, Nat Commun. 2017. WB); cyclin B1 (4138, rabbit, cell signaling, Kanakkanthara et al, Science. 2016. WB); cyclin D1 (2978, rabbit, cell signaling, Zhang et al, Nature. 2018. WB); cyclin E2 (4132, rabbit, cell signaling, Guo et al, Nat Cell Biol. 2016. WB); GAPDH (CB1001, mouse, Calbiochem, Gersch et al, Mol Cell. 2019: WB); α-tubulin (2144, rabbit, cell signaling, Bae et al, J Cell Biol. 2019. IF); γ-tubulin (T6557, mouse, Sigma, Douanne et al, Cell Rep. 2019: IF); cullin9 (A300-098A, rabbit, bethyl, Polajnar et al, EMBO Reports. 2017.WB); FBXL10 (09-864, rabbit, Merck Millipore, Han et al, Oncogene, 2016. WB); pS139-γH2AX (clone JBW301, 05-636, mouse, Millipore, Vermeij et al, Nature, 2016. WB and IF); Mad2 (A300-301A, rabbit, Bethyl Laboratories, Baron et al, eLife. 2016. IF); hemagglutin (HA)-tag (rat, Roche, Amodeo et al, J Cell Sci. 2018. WB); GFP (A10262, chicken IgY, Invitrogen, D'Souza et al, eLife. 2020. WB); GFP-Booster Alexa Fluor 488 (gb2AF488, Chromotek. Buchfellner et al, Plos One. 2016. IF). |

# Eukaryotic cell lines

Policy information about cell lines

| Cell line source(s) | HeLa cell  ATCC number: CCL-2 <br> HEK293T cell  ATCC number: CRL-3216 <br> U2OS cell  ATCC number: HTB-96 <br> RPE1 cell  ATCC number: CRL-4000 <br> HAP1 cell: Horizon Discovery C859 <br> Mouse K1/K2 double floxed fibroblast: generated in the lab |
|---|---|
| Authentication | HeLa, HEK293T, U2OS, RPE1 cell lines were purchased from ATCC and not authenticated in the study. HAP1 cells were purchased from Horizon Discovery and not authenticated in the lab. Mouse K1/K2 double floxed fibroblast was generate in our lab and previously described in eLife 2016;5:e10130. |
| Mycoplasma contamination | All cell lines used were tested negative for mycoplasma contamination |
| Commonly misidentified lines (See ICLAC register) | Not present in this study |

# Flow Cytometry

## Plots

Confirm that:

☒ The axis labels state the marker and fluorochrome used (e.g. CD4-FITC).

☐ The axis scales are clearly visible. Include numbers along axes only for bottom left plot of group (a 'group' is an analysis of identical markers).

☒ All plots are contour plots with outliers or pseudocolor plots.

☒ A numerical value for number of cells or percentage (with statistics) is provided.

## Methodology

| | |
|---|---|
| Sample preparation | Interphase cells were trypsinized and mitotic cells were harvested by shake-off. For surface integrin profiling, cells were washed with cold PBS twice and incubated in 200 μL antibody solution (PBS with 2% BSA) for 1 hr on ice in dark. Cells were washed once with cold PBS and applied on LSRFortessa™ X-20 Cell Analyzer. For DNA content analysis, cells were fixed with 70% ethanol for 3 hr at -20 °C, washed twice with cold PBS and resuspended in 200 μL FxCycle PI/RNase Staining Solution, incubated for 20 min in dark and applied on LSRFortessa™ X-20 Cell Analyzer. Data was analysis with Flowjo software. |
| Instrument | LSRFortessa™ X-20 Cell Analyzer (BD biosciences) |
| Software | FACsDIVA (version 9.0) and Flowjo (version 8.0) software |
| Cell population abundance | Ten thousand events were recorded in each analysis. The whole population of harvested cells were stained and analyzed. The purity of the cells in different cell cycle phases were determined by DNA content analysis and immunoblotting of cyclins as described in the method section. |
| Gating strategy | No gating strategy was used. |

☐ Tick this box to confirm that a figure exemplifying the gating strategy is provided in the Supplementary Information.

