## [Peer Review File · Nature Cell Biology]

Peer Review Information

Journal: Nature Cell Biology

Manuscript Title: CDK1-cyclin B1-induced kindlin degradation drives focal adhesion disassembly at mitotic entry

Corresponding author name(s): Reinhard Fässler

Reviewer Comments & Decisions:

Decision Letter, initial version:
--

Dear Dr Fässler,

Thank you for submitting your manuscript, "CDK1-cyclin B-induced kindlin degradation drives focal adhesion disassembly at mitotic entry", to Nature Cell Biology, and I apologize for the delay in communicating our decision to you. Having carefully assessed and discussed this work in the team of editors, we regret that it is not suitable for this journal.

We decline a substantial number of submissions without formal peer-review, so that they may be considered elsewhere without delay. These editorial decisions are based on considerations such as the potential interest of a study for a broad cell biology audience, the level of advance provided relative to the published literature, and the level of development of the dataset.

Our decision is not a criticism of the technical quality of your work. We recognize that demonstrating that CDK1-cyclin B induces degradation of kindlins at mitotic entry, which is needed for mitotic progression, will be compelling to researchers working in this field. However, after a detailed assessment of this work in relation to published studies in this area, in our view, these findings do not represent a sufficient conceptual advance for a general cell biological readership, to justify publication in Nature Cell Biology.

Although we regret that we cannot offer to publish your paper in Nature Cell Biology, you might wish to consider our sister journal *Nature Communications* as a potential venue for the publication of these results. *Nature Communications* publishes high quality and influential research across the full spectrum of the natural sciences. More information on the journal, the potential benefits of transfer and a link to transfer your paper, can be found at the bottom of this email. Please note that the editorial team at Nature Communications will consider your manuscript independently of our suggestion to transfer.

We are sorry that we could not be more positive on this occasion, but we appreciate the opportunity to

consider this work, and wish you success in seeking publication elsewhere.

Kind regards,
Melina

Melina Casadio, PhD
Senior Editor, Nature Cell Biology
ORCID ID: <https://orcid.org/0000-0003-2389-2243>

** *Nature Communications* is the Nature Research flagship Open Access journal. If you would like this work to be considered for publication there, you can easily transfer the manuscript by following the instructions below. It is not necessary to reformat your paper. Once all files are received, the editors at *Nature Communications* will assess your manuscript's suitability for potential publication; they aim to provide feedback quickly, with a median decision time of 8 days for first editorial decisions on suitability. The journal offers double blind and transparent peer review options. For information on journal metrics please visit our Nature journals metrics page. Our open access pages contain information about article processing charges, open access funding, and advice and support from Springer Nature.

** To transfer your manuscript to Nature Communications, or another Nature Portfolio journal, please use our manuscript transfer portal. If you transfer to Nature journals or to the Communications journals, you will not have to re-supply manuscript metadata and files, unless you wish to make modifications. This link can only be used once and remains active until used.

All Nature Portfolio journals are editorially independent, and the decision on your manuscript will be taken by their editors. For more information, please see our manuscript transfer FAQ page.

Note that any decision to opt in to In Review at the original journal is not sent to the receiving journal on transfer. You can opt in to In Review at receiving journals that support this service by choosing to modify your manuscript on transfer. In Review is available for primary research manuscript types only.

** For Nature Research general information and news for authors, see <http://npg.nature.com/authors>.

Decision Letter, first revision:

Dear Dr Fässler,

I hope you are well. I once again apologize for the delay due to a higher than regular volume of submissions and recent editorial team absences. Your manuscript, "CDK1-cyclin B1-induced kindlin degradation drives focal adhesion disassembly at mitotic entry", has now been seen by 3 referees, who are experts in integrin trafficking (referee 1); focal adhesions and mechanobiology (referee 2); and cell cycle, cyclins (referee 3). As you will see from their comments (attached below) they find this work of potential interest, but have raised substantial concerns, which in our view would need to be addressed with considerable revisions before we can consider publication in Nature Cell Biology.

Nature Cell Biology editors discuss the referee reports in detail within the editorial team, including the chief editor, to identify key referee points that should be addressed with priority, and requests that are overruled as being beyond the scope of the current study. To guide the scope of the revisions, I have listed these points below. We are committed to providing a fair and constructive peer-review process, so please feel free to contact me if you would like to discuss any of the referee comments further.

In particular, it would be essential to:

A) Clarify and distinguish between potential effects of phosphorylation and degradation (reviewers #1 and #3)

B) Clarify potential effects of Kindlin on cell cycle progression (Reviewers #1 and #3)

C) Assess focal adhesion component localization (Reviewers #1 and #2).

D) All other referee concerns pertaining to strengthening existing data, providing controls, methodological details, clarifications and textual changes, should also be addressed.

E) Finally please pay close attention to our guidelines on statistical and methodological reporting (listed below) as failure to do so may delay the reconsideration of the revised manuscript. In particular please provide:

We would be happy to consider a revised manuscript that would satisfactorily address these points, unless a similar paper is published elsewhere, or is accepted for publication in Nature Cell Biology in the meantime.

- ensure that it conforms to our format instructions and publication policies (see below and www.nature.com/nature/authors/).

- provide a point-by-point rebuttal to the full referee reports verbatim, as provided at the end of this letter.

- provide the completed Editorial Policy Checklist (found here <https://www.nature.com/authors/policies/Policy.pdf>), and Reporting Summary (found here <https://www.nature.com/authors/policies/ReportingSummary.pdf>). This is essential for

reconsideration of the manuscript and these documents will be available to editors and referees in the event of peer review. For more information see <http://www.nature.com/authors/policies/availability.html> or contact me.

Nature Cell Biology is committed to improving transparency in authorship. As part of our efforts in this direction, we are now requesting that all authors identified as 'corresponding author' on published papers create and link their Open Researcher and Contributor Identifier (ORCID) with their account on the Manuscript Tracking System (MTS), prior to acceptance. ORCID helps the scientific community achieve unambiguous attribution of all scholarly contributions. You can create and link your ORCID from the home page of the MTS by clicking on 'Modify my Springer Nature account'. For more information please visit www.springernature.com/orcid.

[REDACTED]

We would like to receive a revised submission within six months. We would be happy to consider a revision even after this timeframe, however if the resubmission deadline is missed and the paper is eventually published, the submission date will be the date when the revised manuscript was received.

We hope that you will find our referees' comments, and editorial guidance helpful. Please do not hesitate to contact me if there is anything you would like to discuss.

Best wishes,

Daryl

Daryl J.V. David, PhD

Senior Editor, Nature Cell Biology
Consulting Editor, Nature Communications
Nature Portfolio

Heidelberger Platz 3, 14197 Berlin, Germany
Email: daryl.david@nature.com
ORCID: <https://orcid.org/0000-0002-9253-4805>

Reviewers' Comments:

Reviewer #1:

Remarks to the Author:

Chen et al show that the key regulators of cell-matrix adhesion kindlin-1 and -2 are phosphorylated by CDK1/cyclinB1 as synchronized cells enter M phase, and this promotes ubiquitination and degradation via recruitment of an E3 ligase complex CUL9/FBXL10. Using phospho-mimetic (S181D) and phospho-null K2 mutants, the authors show that this phosphorylation and degradation is a critical step in mitotic rounding, and without it faithful segregation of chromosomes is abrogated. The manuscript combines cell biology, biochemistry, and mass spectrometry in a series of well controlled experiments to provide a new understanding of how cells downregulate cell-matrix adhesion to allow mitotic rounding.

Major comments:

It seems that almost all the cell cycle experiments involve use of synchronization, which as the authors point out was important in leading them to identify the specific effects on K1 and K2. However, synchronization can have profound effects on cell physiology, and it is therefore important to show that the phospho-degron is active in the absence of intervention. It should be relatively straightforward using cell cycle markers to demonstrate phosphorylation/loss of K1 and K2 in otherwise un-treated cells as they enter mitosis. Figure 1E shows K2 levels in live cells but the legend is not clear (normalization would suggest release after CDK inhibition?) and representative images should be shown to support the quantification (and are K2 levels in mitosis significantly different from interphase?).

The manuscript draws strong conclusions related to CDK1/cyclinB1. The authors are correct in their assertion that the global impact of CDK1/cyclinB1 on mitosis makes intervention and interpretation difficult, but at present timing post synchronization, in vitro phosphorylation and co-IP (Figure 3C) imply that it is the CDK1/cyclinB1 complex that phosphorylates kindlin-1/2. Knock down of cyclinB1 would demonstrate whether this CDK/cyclin pair are indeed necessary for the phosphorylation event in cells.

The fact that some kindlin remains in retraction fibers is interesting and helps explain the effect of phospho-mimetic substitutions in K2. However, it appears these experiments were performed using overexpressed K2 and it is important to confirm that endogenous kindlin persists in these structures (antibody staining, CRISPR knock-in). How does some phosphorylated kindlin escape degradation? Are retraction fibers inaccessible to the E3 ligase system?

The authors use double-null kindlin fibroblasts to reveal the impact of K2 mutants on cell cycle progression. What is the impact of loss of kindlin function on cell cycle progression? Do these cells show defects in mitotic rounding analogous to K2-S181D?

What is the effect of K2 phospho-mutants on formation of adhesion complexes in interphase and mitotic cells? Images in S5J and 6A are suggestive of changes, but because K1 has been implicated in spindle assembly by localizing to centrioles (Patel 2013 Nature comms) it is important to define the role of kindlin with respect to adhesion as indicated in the overall model.

Mass spectrometry revealed the presence of phosphorylation sites in K1 and K2 but follow up experiments only focus on K2. Given that the authors draw major conclusions around 'kindlin' and 'K1 and K2' it is important to confirm key findings with K1 phospho-mutants.

K2-S181A is less stable in mitosis than S5A, and the latter is used in experiments to show the effect of stabilized K2 on cell cycle progression. Are the 4 sites around S181 also dependent on CDK1/cyclinB1? In vitro phosphorylation can be promiscuous, and it is therefore important to show either the physiological effect of K2-S181A (where phosphorylation is convincingly shown to depend on CDK1) or that phosphorylation of surrounding Ser residues is similarly controlled by CDK1/cyclinB1 (e.g., inhibition/knock down).

It is notable that CUL9/FBXL10 knockdown rescue adhesion complexes and K2-S189D recruitment there. Does CUL9/ FBXL10 knockdown result in prolonged entry to mitosis in K2-S181 expressing cells?

Minor comments:

Fig 3H: Not clear how this quantification was performed

Figure 6H: Polyubiquitination would benefit from additional quantification

Reviewer #2:

Remarks to the Author:

In this manuscript, the molecular mechanisms underlying the role of kindling-1/2 in mitotic cell rounding are dissected. Upon mitotic entry, kindlin1&2 were shown to be significantly phosphorylated and degraded at the protein level. A small fraction of kindlin that remains were observed to localize to retraction fibers and co-localized with activated integrin. Biochemical interactions between kindlin and CDK1/cyclin B1 are demonstrated. Phosphorylation of serine S181 in kindlin2 was identified to be the main post-translational modification of kindlin during mitotic phases, and pS181-kindlin-2 was identified as the form of kindlin localized to the retraction fiber.

The authors then demonstrated that the phosphorylation of S181 (or phosphomimetic mutation thereof) decrease the stability of kindlin in a proteasome but not lysosome- dependent fashion. Cells expressing phosphomimetic S181D kindlin-2 was shown to undergo mitotic rounding faster while taking significantly longer time to exit Mitotic phase. In contrast, cells expressing non-phosphorylatable mutation (5SA or kindlin-3 swap), show a significant decrease in mitotic rounding capacity. Rap1 contribution to mitotic cell rounding was also shown to be insignificant, thus highlighting the importance of kindling phosphorylation-dependent degradation.

Upon examination of the mitotic cells, the authors showed that either phosphomimetic or non-phosphorylatable kindling-2 cells are prone to mitosis defect as shown by spindle and centrosome defect, markers of DNA damage gH2AX, % multi-nucleated cells, and apoptosis.

The authors then performed siRNA screening that implicated CUL9 and FBXL10. Mass spectrometric/immunoprecipitate analysis suggests kindlin2 could be ubiquitinated by cullin-ring ligase complex formed by CUL9/FBXL10. The authors then further confirmed that CDK1/Cyclin B1 dependent phosphorylation is requisite for CUL9/FBXL10 dependent degradation. Consistent with this, depletion of CUL9/FBXL10 decrease mitotic cell rounding.

Overall, this manuscript is a tour de force study in cell biology that advance our understanding the

multifaceted roles of kindlin, an important protein conventionally associated with cell adhesions, in mitosis. The current literature in mitotic cell rounding tend to focus on the actomyosin/cytoskeleton/mechanics aspects of the process, while biochemical understanding has hitherto been incomplete. This manuscript is potentially shedding new light onto how adhesion proteins such as kindlin may play particularly important role as master regulator of this process. I am supportive of its eventual publication in Nature Cell Biology after addressing a number of issues below.

1. Given the multifaceted nature of kindlin functions and previous literature, I suggest that the model (Fig. 7) and/or discussion may be revised to take into account a number of recent studies showing kindling to be directly localized in the centrosome and play additional roles in mitotic processes. These include "Patel, Hitesh, et al. "Kindlin-1 regulates mitotic spindle formation by interacting with integrins and Plk-1." Nature communications 4.1 (2013): 1-9." which may have been overlooked by the authors. This study as well as ref.33-34 showed that kindlin1/2 directly affect microtubules/spindle properties by either modifying acetylation or as being localized in centrosome. Thus, this raise the possibility that kindlin may contribute to mitosis via multiple mechanisms. In my opinion, this should merit some space in the discussion, which in the current form is focused primarily on kindlin and adhesions.

2. Interestingly, in Fig.1a, besides kindlin1,2 paxillin is also showing a significant shift on the gel. Given the roles of phosphopaxillin in interacting with vinculin, and the generally well-known role of vinculin in stabilizing integrin-based adhesions, perhaps the possibility that these proteins may (or may not) contribute to mitotic rounding should be addressed or discussed.

Reviewer #3:

Remarks to the Author:

In this study the authors have identified a role for phosphorylation and potentially degradation of focal adhesion components in cell rounding at mitosis. They show that the kindlin protein is phosphorylated in mitosis on serine-proline site that can be phosphorylated by cyclin B1-Cdk1, and that this correlates with a reduction in kindling protein in cells, and at focal adhesions in particular. The authors identify a potential ubiquitin ligase (Cul9-FBXL10) and show that mutating the phosphorylation sites or depleting Cul9 or FBXL10 leads to a reduction in ubiquitylation and problems in cell rounding, spindle morphology and chromosome segregation.

Overall, this is an extensive study that sheds an unexpected light on the mechanism of cell rounding at mitosis. Most of the data are clear and generated from experiments with appropriate controls. I think that the insights would be of interest to the readers of Nature Cell Biology, but before publication there are a number of specific concerns - particularly in the strength of conclusions drawn from indirect evidence - that should be addressed.

1) Line 139: the sites identified are not canonical CDK sites since they lack flanking basic amino acids. They are, however, potential cyclin-Cdk sites. The wording should be changed.

2) Line 139: it is of interest that Ser181 has an amino-terminal serine making this a potential Plk1-binding site (SS*P). It might benefit the authors to determine whether mutating Ser 180 also has an effect on the behaviour of the protein since Cyclin B1-Cdk1 and Plk1 often act in tandem, and the authors own data that mutating 5 serines has a more profound effect on stability.

3) Lines 143 - 144: the blots in Fig3c should be improved. There are no molecular mass markers and I

am concerned that there multiple bands in the 'letter boxes' presented for Cyclin A in the STLC-arrested cells.

Line 145: I would remove the statement that the absence of Cyclin A co-precipitation supports the lack of interphase phosphorylation because Cyclin A is also an important kinase for mitotic entry and has a number of mitosis-specific substrates.

Line 156: the citations for the oscillatory behaviour of cyclin B1-Cdk1 activity are inappropriate. Please change these to citations to papers that measure activity not protein level.

Line 158: the authors have made mutants that both cannot be phosphorylated on specific sites and are more stable; they have not made mutants that can be phosphorylated but not degraded. Without these separation of function mutants they cannot ascribe the effects of the mutations only to preventing degradation. The wording should be changed to reflect this.

Line 178: the authors note the increase in aneuploidy in Fig 4d but to this referee the major effect appears to be an accumulation of cells in S phase. The authors should comment on this.

Line 179: the authors use RO3306 block and release to assay the timing of cell rounding, but RO3306 blocks perturb the normal cell cycle. The authors should also measure cell rounding in untreated cells.

Line 189: as far as I can tell the authors have not assayed focal adhesion disassembly directly and are only inferring that kindlin phosphorylation (and potentially degradation) are indispensable for this. The wording should be changed to reflect this.

Line 208: Bub1 staining does not directly correlate with Spindle Assembly Checkpoint (SAC) activity; to conclude that SAC activity is prolonged the authors should stain for Mad2.

Lines 236-237: this referee is unable to see much of a difference between the smear representing ubiquitylation in the IgG vs Cul9 immunoprecipitates. These data do not convince me that Cul9 ubiquitylates kindlin 2.

Line 252 - 253: the authors should note that Cyclin A preferentially binds to Cdk2 in S phase and only later to Cdk1 (10.1016/j.molcel.2011.03.031).

Methods should be written concisely, but should contain all elements necessary to allow interpretation and replication of the results. As a guideline, Methods sections typically do not exceed 3,000 words. The Methods should be divided into subsections listing reagents and techniques. When citing previous

methods, accurate references should be provided and any alterations should be noted. Information must be provided about: antibody dilutions, company names, catalogue numbers and clone numbers for monoclonal antibodies; sequences of RNAi and cDNA probes/primers or company names and catalogue numbers if reagents are commercial; cell line names, sources and information on cell line identity and authentication. Animal studies and experiments involving human subjects must be reported in detail, identifying the committees approving the protocols. For studies involving human subjects/samples, a statement must be included confirming that informed consent was obtained. Statistical analyses and information on the reproducibility of experimental results should be provided in a section titled "Statistics and Reproducibility".

All Nature Cell Biology manuscripts submitted on or after March 21 2016 must include a Data availability statement at the end of the Methods section. For Springer Nature policies on data availability see <http://www.nature.com/authors/policies/availability.html>; for more information on this particular policy see <http://www.nature.com/authors/policies/data/data-availability-statements-data-citations.pdf>. The Data availability statement should include:

- Accession codes for primary datasets (generated during the study under consideration and designated as "primary accessions") and secondary datasets (published datasets reanalysed during the study under consideration, designated as "referenced accessions"). For primary accessions data should be made public to coincide with publication of the manuscript. A list of data types for which submission to community-endorsed public repositories is mandated (including sequence, structure, microarray, deep sequencing data) can be found here <http://www.nature.com/authors/policies/availability.html#data>.
- Unique identifiers (accession codes, DOIs or other unique persistent identifier) and hyperlinks for datasets deposited in an approved repository, but for which data deposition is not mandated (see here for details <http://www.nature.com/sdata/data-policies/repositories>).
- At a minimum, please include a statement confirming that all relevant data are available from the authors, and/or are included with the manuscript (e.g. as source data or supplementary information), listing which data are included (e.g. by figure panels and data types) and mentioning any restrictions on availability.
- If a dataset has a Digital Object Identifier (DOI) as its unique identifier, we strongly encourage including this in the Reference list and citing the dataset in the Methods.

We recommend that you upload the step-by-step protocols used in this manuscript to the Protocol Exchange. More details can found at www.nature.com/protocolexchange/about.

All imaging data should be accompanied by scale bars, which should be defined in the legend. Cropped images of gels/blots are acceptable, but need to be accompanied by size markers, and to retain visible background signal within the linear range (i.e. should not be saturated). The boundaries of panels with low background have to be demarked with black lines. Splicing of panels should only be considered if unavoidable, and must be clearly marked on the figure, and noted in the legend with a statement on whether the samples were obtained and processed simultaneously. Quantitative comparisons between samples on different gels/blots are discouraged; if this is unavoidable, it should only be performed for samples derived from the same experiment with gels/blots were processed in parallel, which needs to be stated in the legend.

- For line art, graphs, charts and schematics we prefer Adobe Illustrator (.AI), Encapsulated PostScript (.EPS) or Portable Document Format (.PDF). Files should be saved or exported as such directly from the application in which they were made, to allow us to restyle them according to our journal house style.
- We accept PowerPoint (.PPT) files if they are fully editable. However, please refrain from adding PowerPoint graphical effects to objects, as this results in them outputting poor quality raster art. Text used for PowerPoint figures should be Helvetica (preferred) or Arial.
- We do not recommend using Adobe Photoshop for designing figures, but we can accept Photoshop generated (.PSD or .TIFF) files only if each element included in the figure (text, labels, pictures, graphs, arrows and scale bars) are on separate layers. All text should be editable in 'type layers' and line-art such as graphs and other simple schematics should be preserved and embedded within 'vector smart objects' - not flattened raster/bitmap graphics.
- Some programs can generate Postscript by 'printing to file' (found in the Print dialogue). If using an application not listed above, save the file in PostScript format or email our Art Editor, Allen Beattie for advice (a.beattie@nature.com).

The total number of Supplementary Figures (not including the “unprocessed scans” Supplementary Figure) should not exceed the number of main display items (figures and/or tables (see our Guide to Authors and March 2012 editorial <http://www.nature.com/ncb/authors/submit/index.html#suppinfo>; <http://www.nature.com/ncb/journal/v14/n3/index.html#ed>). No restrictions apply to Supplementary Tables or Videos, but we advise authors to be selective in including supplemental data.

GUIDELINES FOR EXPERIMENTAL AND STATISTICAL REPORTING

REPORTING REQUIREMENTS – To improve the quality of methods and statistics reporting in our papers we have recently revised the reporting checklist we introduced in 2013. We are now asking all life sciences authors to complete two items: an Editorial Policy Checklist (found here <https://www.nature.com/authors/policies/Policy.pdf>) that verifies compliance with all required editorial policies and a reporting summary (found here <https://www.nature.com/authors/policies/ReportingSummary.pdf>) that collects information on experimental design and reagents. These documents are available to referees to aid the evaluation of the manuscript. Please note that these forms are dynamic 'smart pdfs' and must therefore be downloaded and completed in Adobe Reader. We will then flatten them for ease of use by the reviewers. If you would like to reference the guidance text as you complete the template, please access these flattened versions at <http://www.nature.com/authors/policies/availability.html>.

Author Rebuttal, first revision:

Reviewers' Comments:

Reviewer #1:

Remarks to the Author:

Chen et al show that the key regulators of cell-matrix adhesion kindlin-1 and -2 are phosphorylated by CDK1/cyclinB1 as synchronized cells enter M phase, and this promotes ubiquitination and degradation via recruitment of an E3 ligase complex CUL9/FBXL10. Using phospho-mimetic (S181D) and phospho-null K2 mutants, the authors show that this phosphorylation and degradation is a critical step in mitotic rounding, and without it faithful segregation of chromosomes is abrogated. The manuscript combines cell biology, biochemistry, and mass spectrometry in a series of well controlled experiments to provide a new understanding of how cells downregulate cell-matrix adhesion to allow mitotic rounding.

We thank the reviewer for the excellent comments and the support.

Major comments:

1 : It seems that almost all the cell cycle experiments involve use of synchronization, which as the authors point out was important in leading them to identify the specific effects on K1 and K2. However, synchronization can have profound effects on cell physiology, and it is therefore important to show that the phospho-degron is active in the absence of intervention. It should be relatively straightforward using cell cycle markers to demonstrate phosphorylation/loss of K1 and K2 in otherwise un-treated cells as they enter mitosis. Figure 1E shows K2 levels in live cells but the legend is not clear (normalization would suggest release after CDK inhibition?) and representative images should be shown to support the quantification (and are K2 levels in mitosis significantly different from interphase?).

A: The reviewer is right that cell synchronization can affect cell physiology. To examine whether K1 and K2 are also degraded in mitosis of normally cycling cells, we carried out two experiments. Firstly, we harvested mitotic HeLa and U2OS cells solely by shake-off, determined K1 and K2 levels by WB and found that K1 and K2 abundance was reduced and the apparent molecular weight increase (see revised manuscript - Extended Data Fig. 1j). Secondly, we immuno-stained unsynchronized HeLa cells with anti-K2 and anti-pH3 antibodies to confirmed a reduction of the K2 signal in M-phase (see Fig. 1 for reviewers at the end of the rebuttal).

The reviewer correctly noted that the EGFP-K2 fluorescence signal in Figure 1e was recorded after RO3306 washout. We revised the Fig. 1 legend. The revised manuscript shows now representative images at indicated time points to support the quantification (see revised manuscript - Extended Data Fig. 1l).

2: The manuscript draws strong conclusions related to CDK1/cyclinB1. The authors are correct in their assertion that the global impact of CDK1/cyclinB1 on mitosis makes intervention and interpretation difficult, but at present timing post synchronization, in vitro phosphorylation and co-IP (Figure 3C) imply that it is the CDK1/cyclinB1 complex that phosphorylates kindlin-1/2. Knock down of cyclinB1 would demonstrate whether this CDK/cyclin pair are indeed necessary for the phosphorylation event in cells.

A: CDK1-cyclin B1 is the master mitotic kinase and the engine that drives mitotic entry. It is well accepted that silencing cyclin B1 or inhibiting CDK1 causes G2/M arrest and prevents all mitosis-specific phosphorylation events including phosphorylation of K1 and K2. To test this perception, we inhibited the activity of CDK1-cyclin B1 in STLC-treated EGFP-K2-expressing HeLa cells by either siRNA silencing cyclin B1 or RO3306-mediated inhibition of CDK1. Both treatments strongly inhibited K1 and K2 degradation shown by WB, and phosphorylation of S181 and adjacent serines (including S175, S177 and S180) shown by phosphoproteomic analysis (see Fig. 2 for reviewers at the end of the rebuttal).

3: The fact that some kindlin remains in retraction fibers is interesting and helps explain the effect of phospho-mimetic substitutions in K2. However, it appears these experiments were performed using overexpressed K2 and it is important to confirm that endogenous kindlin persists in these structures (antibody staining, CRISPR knock-in). How does some phosphorylated kindlin escape degradation? Are retraction fibers inaccessible to the E3 ligase system?

A: To validate that also endogenous K2 is present in the mitotic retraction fibers, we co-stained HeLa cells for K2, pH3 and phalloidin. We clearly observed endogenous K2 puncta along the mitotic retraction fibers (see revised manuscript - Fig. 3e) resembling the pattern of EGFP-K2 in these sites. To explore how K2 escape degradation in mitotic retraction fibers, we investigated the localization of the E3 ligase complex in mitotic cells. To this end, we immunostained EGFP-K2-expressing HeLa cells with antibodies against CUL9 or FBXL10, always in combination with phalloidin. The majority of CUL9 and FBXL10 is cytosolic (see revised Extended Data Fig. 6i, j). However, FBXL10 but not CUL9 co-localized with EGFP-K2 puncta in mitotic retraction fiber (see revised Extended Data Fig. 6i, j). This finding indicates, as hypothesized by the reviewer, that CUL9 cannot access the retraction fibers and therefore, K2 escapes ubiquitination and degradation in these sites.

4: The authors use double-null kindlin fibroblasts to reveal the impact of K2 mutants on cell cycle progression. What is the impact of loss of kindlin function on cell cycle progression? Do these cells show defects in mitotic rounding analogous to K2-S181D?

A: The mutant K2-expressing mouse fibroblasts were not generated on top of the K1/K2 double-null state. Instead, we first introduced EGFP-tagged WT or mutant K2 into the K1/K2 double floxed mouse fibroblasts and subsequently removed the endogenous *Fermt1* and *Fermt2* alleles by adenovirally transducing Cre (See revised method section line 382-385). To better explain the process of generating these cell lines, we have changed the text in the result section: see revised lines 175-177 'we retrovirally transduced the K1/K2 double-floxed mouse fibroblasts with cDNAs encoding WT or mutant EGFP-K2 and subsequently removed the *Fermt1* and *Fermt2* alleles by adenoviral cre transduction'. We also analyzed the cell cycle progression of K1/K2-null mouse fibroblasts. Consistent with our published findings (Theodosiou et al., 2016), loss of K1/K2 generates round cells (see Fig. 3 for reviewers at the end of the rebuttal). After RO3306 washout, K1/K2-null fibroblasts fail to complete cytokinesis. As a consequence, double strand DNA breaks revealed by rH2AX staining are significantly

increased compared to the parental K1/K2 double floxed cells (see Fig. 3 for reviewers at the end of the rebuttal).

5: What is the effect of K2 phospho-mutants on formation of adhesion complexes in interphase and mitotic cells? Images in S5J and 6A are suggestive of changes, but because K1 has been implicated in spindle assembly by localizing to centrioles (Patel 2013 Nature comms) it is important to define the role of kindlin with respect to adhesion as indicated in the overall model.

A: We addressed this comment by quantifying number and size of paxillin-positive FAs in EGFP-tagged WT or mutant K2-expressing fibroblasts. Although the FA size was not significantly affected in the K2-S181D mutant, the FA number was reduced and the cells were smaller (see revised manuscript - Extended Data Fig. 4d-f). Interestingly, K2-5SA and K2-SWAP cells decreased FA size but increased the FA number and adopted a larger area compared to K2-WT cells (see revised manuscript - Extended Data Fig. 4d-f).

In K2-WT fibroblasts the FAs were efficiently disassembled already 5 min after RO3306 release and no FAs were detected 10 min after RO3306 release. This trend was more pronounced in K2-S181D fibroblasts, which lost their FAs already 5 min after RO3306 washout (see revised manuscript - Fig. 4j, k). In contrast, K2-5SA and K2-SWAP fibroblasts, which harbor more but slightly smaller FAs in interphase, kept their FAs after RO3306 washout (see revised manuscript - Fig. 4j, k). These data confirm the role of the K2 phosphorylation as a mean to regulate FA dynamics at mitotic entry.

6: Mass spectrometry revealed the presence of phosphorylation sites in K1 and K2 but follow up experiments only focus on K2. Given that the authors draw major conclusions around 'kindlin' and 'K1 and K2' it is important to confirm key findings with K1 phospho-mutants.

A: Both, K1 and K2 were degraded in mitosis as shown in Fig. 1a and Extended Fig. 3a. The sequence encompassing the phospho-degron in the F1 loops of K1 and K2 is highly conserved. To confirm our key findings also for K1, we expressed three K1 mutants in HeLa cells: phosphomimetic K1-S179D, phosphoinhibitory K1-S179A, and K1-5SA in which the S179 and adjacent serines (S172, S174, S176 and S184) were alanine-substituted. In line with the K2 findings, the protein stability of EGFP-K1-S179D in interphase HeLa cells was significantly reduced, whereas mitotic degradation of EGFP-tagged K1-S179A and K1-5SA was inhibited (see revised manuscript - Extended Data Fig. 4g). To measure mitotic rounding, we retrovirally transduced K1/K2 double floxed mouse fibroblasts with cDNAs encoding EGFP-tagged K1-WT, or K1-S179D/A or K1-5SA and then deleted the floxed *Fermt1* and *Fermt2* alleles by transducing Cre. This experiment revealed that K1-S179D expressing fibroblasts rounded up faster than K1-WT cells, while mitotic rounding in RO3306-released K1-S179A cells was delayed and almost blunted in K1-5SA cells (see revised manuscript - Extended Data Fig. 4h).

7: K2-S181A is less stable in mitosis than 5SA, and the latter is used in experiments to show the effect of stabilized K2 on cell cycle progression. Are the 4 sites around S181 also dependent on CDK1/cyclinB1? In vitro phosphorylation can be promiscuous, and it is therefore important to show

either the physiological effect of K2-S181A (where phosphorylation is convincingly shown to depend on CDK1) or that phosphorylation of surrounding Ser residues is similarly controlled by CDK1/cyclinB1 (e.g., inhibition/knock down).

A: To validate whether phosphorylation of the 4 serine residues adjacent to S181 depends on CDK1-cyclin B1, we compared EGFP-K2 phosphorylation in STLC-arrested mitotic cells treated with or without RO3306, or transfected with siRNA against cyclin B1. The results showed that inhibition/depletion of the CDK1-cyclin B1 complex blocked phosphorylations of S181 as well as S175, S177, S180.

Furthermore, K2 degradation was concomitantly inhibited by these treatments in mitosis (see Fig. 2 for reviewers at the end of this rebuttal).

8: It is notable that CUL9/FBXL10 knockdown rescue adhesion complexes and K2-S181D recruitment there. Does CUL9/ FBXL10 knockdown result in prolonged entry to mitosis in K2- S181D expressing cells?

A: This is an important request. We depleted CUL9 and /or FBXL10 in K2-S181D cells by siRNA and analyzed mitotic rounding after RO3306 release. The experiments revealed that the depletion of CUL9 and/or FBXL10 stabilized EGFP-K2-S181D and decelerated rounding of K2-S181D expressing cells (see revised Fig. 6e, f).

Minor comments:

Fig 3H: Not clear how this quantification was performed

A. The phalloidin signal of the confocal image was used to define the cell body and all the Z- sections capturing the cell body was stacked together to quantify the fluorescent signal of anti-K2- pS181 staining. We have added this information in the method section (see revised manuscript - lines 462-466).

Figure 6H: Polyubiquitination would benefit from additional quantification

A. We added the quantification (see revised manuscript - Fig. 6k).

Reviewer #2:

Remarks to the Author:

In this manuscript, the molecular mechanisms underlying the role of kindling-1/2 in mitotic cell rounding are dissected. Upon mitotic entry, kindlin1&2 were shown to be significantly phosphorylated and degraded at the protein level. A small fraction of kindlin that remains were observed to localize to retraction fibers and co-localized with activated integrin. Biochemical interactions between kindlin and CDK1/cyclin B1 are demonstrated. Phosphorylation of serine S181 in kindlin2 was identified to be the main post-translational modification of kindlin during mitotic phases, and pS181-kindlin-2 was identified as the form of kindlin localized to the retraction fiber.

The authors then demonstrated that the phosphorylation of S181 (or phosphomimetic mutation thereof) decrease the stability of kindlin in a proteasome but not lysosome- dependent fashion. Cells

expressing phosphomimetic S181D kindlin-2 was shown to undergo mitotic rounding faster while taking significantly longer time to exit Mitotic phase. In contrast, cells expressing non-phosphorylatable mutation (5SA or kindlin-3 swap), show a significant decrease in mitotic rounding capacity. Rap1 contribution to mitotic cell rounding was also shown to be insignificant, thus highlighting the importance of kindling phosphorylation-dependent degradation. Upon examination of the mitotic cells, the authors showed that either phosphomimetic or non-phosphorylatable kindling-2 cells are prone to mitosis defect as shown by spindle and centrosome defect, markers of DNA damage gH2AX, % multi-nucleated cells, and apoptosis. The authors then performed siRNA screening that implicated CUL9 and FBXL10. Mass spectrometric/immunoprecipitate analysis suggests kindlin2 could be ubiquitinated by cullin-ring ligase complex formed by CUL9/FBXL10. The authors then further confirmed that CDK1/Cyclin B1 dependent phosphorylation is requisite for CUL9/FBXL10 dependent degradation. Consistent with this, depletion of CUL9/FBXL10 decrease mitotic cell rounding. Overall, this manuscript is a tour de force study in cell biology that advance our understanding the multifaceted roles of kindlin, an important protein conventionally associated with cell adhesions, in mitosis. The current literature in mitotic cell rounding tend to focus on the actomyosin/cytoskeleton/mechanics aspects of the process, while biochemical understanding has hitherto been incomplete. This manuscript is potentially shedding new light onto how adhesion proteins such as kindlin may play particularly important role as master regulator of this process. I am supportive of its eventual publication in Nature Cell Biology after addressing a number of issues below.

We thank the reviewer for the enthusiastic comments on our work.

1. Given the multifaceted nature of kindlin functions and previous literature, I suggest that the model (Fig. 7) and/or discussion may be revised to take into account a number of recent studies showing kindlin to be directly localized in the centrosome and play additional roles in mitotic processes. These include "Patel, Hitesh, et al. "Kindlin-1 regulates mitotic spindle formation by interacting with integrins and Plk-1." Nature communications 4.1 (2013): 1-9." which may have been overlooked by the authors. This study as well as ref.33-34 showed that kindlin1/2 directly affect microtubules/spindle properties by either modifying acetylation or as being localized in centrosome. Thus, this raise the possibility that kindlin may contribute to mitosis via multiple mechanisms. In my opinion, this should merit some space in the discussion, which in the current form is focused primarily on kindlin and adhesions.
- A. *Thanks for reminding us of the missing references. We agree with the reviewer that these studies together with our data suggest that kindlin may contribute to the flawless mitosis via multiple mechanisms. We have included the references and revised the discussion. See revised lines 334-335: 'A minor fraction of kindlin escapes degradation in M-phase and distributes in the mitotic round cell body, pericentriolar material where it may assist the assembly of the mitotic spindle'.*

2. Interestingly, in Fig.1a, besides kindlin1,2 paxillin is also showing a significant shift on the gel. Given the roles of phosphopaxillin in interacting with vinculin, and the generally well-known role of vinculin in stabilizing integrin-based adhesions, perhaps the possibility that these proteins may (or may not) contribute to mitotic rounding should be addressed or discussed.

A. We agree with the reviewer that discussing the potential roles of paxillin phosphorylation in mitotic rounding is important. We revised the discussion, see lines 331-334: 'An interesting candidate is paxillin, whose phosphorylation, also induced in M-phase, was shown to sensitively control integrin adhesion stability by recruiting proteins such as vinculin, PAK or β -Pix'.

Reviewer #3:

Remarks to the Author:

In this study the authors have identified a role for phosphorylation and potentially degradation of focal adhesion components in cell rounding at mitosis. They show that the kindlin protein is phosphorylated in mitosis on serine-proline site that can be phosphorylated by cyclin B1-Cdk1, and that this correlates with a reduction in kindling protein in cells, and at focal adhesions in particular. The authors identify a potential ubiquitin ligase (Cul9-FBXL10) and show that mutating the phosphorylation sites or depleting Cul9 or FBXL10 leads to a reduction in ubiquitylation and problems in cell rounding, spindle morphology and chromosome segregation. Overall, this is an extensive study that sheds an unexpected light on the mechanism of cell rounding at mitosis. Most of the data are clear and generated from experiments with appropriate controls. I think that the insights would be of interest to the readers of Nature Cell Biology, but before publication there are a number of specific concerns - particularly in the strength of conclusions drawn from indirect evidence - that should be addressed.

We thank the reviewer for the excellent review and support.

1) Line 139: the sites identified are not canonical CDK sites since they lack flanking basic amino acids. They are, however, potential cyclin-Cdk sites. The wording should be changed.

A: We changed 'canonical substrate motif' to 'potential proline-directed CDK1 consensus motif'.

2) Line 139: it is of interest that Ser181 has an amino-terminal serine making this a potential Plk1-binding site (SS*P). It might benefit the authors to determine whether mutating Ser 180 also has an effect on the behaviour of the protein since Cyclin B1-Cdk1 and Plk1 often act in tandem, and the authors own data that mutating 5 serines has a more profound effect on stability.

A: This is an interesting and important comment. We carried out two experiments to address the role of PLK1 for K2 degradation. Firstly, we synchronized EGFP-K2 expressing HeLa cells with RO3306 and released them into medium with or without the selective PLK1 inhibitor BI2536 for 30 min. WBs showed that PLK1 inhibition in mitosis did not affect EGFP-K2 and endogenous K2 levels (see Fig. 4 for reviewers at the end of the rebuttal). Secondly, we generated EGFP-tagged K2-S180D or K2-S180A expressing mouse fibroblasts and measured mitotic rounding. Neither the expression of K2-S180D nor K2-S180A

affected the rounding process (see Fig. 4 for reviewers at the end of the rebuttal). These data indicate that PLK1 does not play decisive roles for K2 degradation in mitosis. Interestingly, the CDK1 motif in K1 is VS*P, which is not a polo-box binding motif. Still, K1 follows similar degradation pattern as K2, which indirectly supports the notion that PLK1 is not directly involved in K2 degradation in M-phase.

- 3) Lines 143 - 144: the blots in Fig3c should be improved. There are no molecular mass markers and I am concerned that these multiple bands in the 'letter boxes' presented for Cyclin A in the STLC-arrested cells.**

A: We improved the blots and indicated the molecular weights in Fig. 3c (See revised manuscript - Fig. 3C replaced for the old blot).

- 4) Line 145: I would remove the statement that the absence of Cyclin A co-precipitation supports the lack of interphase phosphorylation because Cyclin A is also an important kinase for mitotic entry and has a number of mitosis-specific substrates.**

A: We agree and changed the sentence to 'In contrast, cyclin A2 neither associated with EGFP-K2 in untreated nor STLC-arrested cells (Fig. 3c)'.

- 5) Line 156: the citations for the oscillatory behaviour of cyclin B1-Cdk1 activity are inappropriate. Please change these to citations to papers that measure activity not protein level.**

A: We changed the citations.

- 6) Line 158: the authors have made mutants that both cannot be phosphorylated on specific sites and are more stable; they have not made mutants that can be phosphorylated but not degraded. Without these separation of function mutants they cannot ascribe the effects of the mutations only to preventing degradation. The wording should be changed to reflect this.**

A: To separate the functions of phosphorylation and degradation, we attempted to identify the ubiquitination sites on K2. GFP-IP based ubiquitome analysis identified 15 ubiquitination sites on K2 that were distributed from position 56 to 555 in mitotic HeLa cells (see revised supplementary Table S5 which is also attached at the end of this rebuttal). We did not initiate a K2 mutational analysis by substituting lysine residues for arginine, as we were advised that single lysine substitutions usually lead to ubiquitination of secondary sites. Furthermore, the identified lysine residues localize to functionally important domains, which would make the characterization of these mutations and assignment to degradation difficult. Finally, we are also not sure whether our MS analysis identified all potential ubiquitination sites. Many more of them are published (Akimov et al., 2018). Therefore, we followed the reviewer's advice and toned the wording in our revised manuscript to 'K2 phosphorylation is indispensable for rounding, retraction fiber formation and spindle assembly and positioning'. We also depleted CUL9 and/or FBXL10 in K2-S181D cells and found that K2 was stabilized and mitotic rounding strongly suppressed (see revised manuscript - Fig. 6e, f). This experiment suggests that the

effects of the phospho-mimicking K2-S181D substitution on mitotic rounding are tightly associated with K2 degradation.

7) Line 178: the authors note the increase in aneuploidy in Fig 4d but to this referee the major effect appears to be an accumulation of cells in S phase. The authors should comment on this.

A. As noted by the reviewers, flow cytometry showed that the numbers of S-phase cells are increased in mutant K2-expressing fibroblasts. Furthermore, the numbers of apoptotic cells indicated by the sub-G1 peak, were also elevated in the mutant K2-expressing fibroblasts. Interestingly, in mutant K2-expressing fibroblasts the cell numbers in the G1 and G2/M peaks were decreased and outside these two peaks increased. This DNA content distribution was shown to indicate cell populations with increased aneuploidy (Dey, 2018). We favor this interpretation and believe that the accumulation of mutant K2-expressing cells with 2N-4N DNA content is not due to increased DNA replication, as their proliferation rate is significantly suppressed. We followed the reviewers' advice and changed the text in the revised manuscript to reflect these observations (see revised lines 179-180) and also marked the cell population with <2N (Sub-G1) and 2N-4N (S-phase) DNA contents in the revised Fig. 4e.

8) Line 179: the authors use RO3306 block and release to assay the timing of cell rounding, but RO3306 blocks perturb the normal cell cycle. The authors should also measure cell rounding in untreated cells.

A: To measure mitotic rounding in unsynchronized EGFP-tagged WT or mutant K2 expressing fibroblasts, we calculated the circularity in interphase and mitosis using the equation $4\pi \times \text{area}/\text{perimeter}^2$ (Monster et al., 2021). The circularity of K2-WT and K2-S181D cells increased to ~0.90. However, the circularity of K2-5SA and K2-SWAP fibroblasts did not change from interphase to mitosis (see revised manuscript - Fig. 4l). These data demonstrate that also in unsynchronized cell populations, disruption of K2 phosphorylation prevents cell rounding in M-phase.

9) Line 189: as far as I can tell the authors have not assayed focal adhesion disassembly directly and are only inferring that kindlin phosphorylation (and potentially degradation) are indispensable for this. The wording should be changed to reflect this.

A: The reviewer is correct. We did not directly measure FA dynamics at the mitotic entry in wildtype and mutant K2 expressing fibroblasts. We therefore measured FA dynamics in EGFP-tagged WT or mutant K2 expressing fibroblasts at early mitotic entry. FAs in K2-WT fibroblasts were almost fully disassembled and converted into small dots 5 min after RO3306 release. These adhesive dots vanished 10 min after RO3306 release. This trend was more pronounced in K2-S181D mutant, which lost all of their FAs 5 min after RO3306 release (see revised manuscript -Fig. 4j, k). In contrast, FAs persisted in mitotic K2-5SA and K2-SWAP fibroblasts (see revised manuscript - Fig. 4j, k) confirming the role of K2 phosphorylation in regulating FA dynamics at mitotic entry.

- 10) Line 208: Bub1 staining does not directly correlate with Spindle Assembly Checkpoint (SAC) activity; to conclude that SAC activity is prolonged the authors should stain for Mad2.**

A: We agree with the reviewer that Bub1 staining does not directly reflect SAC activity. We repeated Mad2 staining and found similar staining patterns as with Bub1, which corroborates the prolonged SAC activity in mutant K2 expressing fibroblasts (see revised manuscript - Extended Fig5.e, f).

- 11) Lines 236-237: this referee is unable to see much of a difference between the smear representing ubiquitylation in the IgG vs Cul9 immuno-precipitates. These data do not convince me that Cul9 ubiquitylates kindlin 2.**

A: We realized that the smear representing ubiquitylation in the IgG precipitates was due to the excessive amount of recombinant E1 and E2 ligases that was used in the experimental reactions. We carefully optimized the experimental conditions and repeated the in vitro ubiquitination assay with less amount of recombinant E1 and E2 ligases. The results could be considerably improved. The old blots have been replaced with the improved new ones. Furthermore, the extent of ubiquitination has been quantified (See revised manuscript - Fig. 6j, k).

- 12) Line 252 - 253: the authors should note that Cyclin A preferentially binds to Cdk2 in S phase and only later to Cdk1 (10.1016/j.molcel.2011.03.031).**

A: This statement referred to a previous study (Jones et al., 2018), which demonstrated that depletion of either cyclin A2 or CDK1, or chemical inhibition of CDK1 by RO3306 reduced FA size in S phase (Fig. 2, Jones et al, 2018). In this study it was also shown that inhibition of CDK2 by SNS032 had no effect on FA size in S phase (Fig. S2A, B; Jones et al, 2018). Although cyclin A2 associates with CDK2 in S phase, the finding by Jones et al. (JCB, 2018) excludes the CDK2-cyclin A2 complex in regulating FA size. We only want to refer to this paper when we write 'in S phase, CDK1 associates with cyclin A2 to increase FA size'.

References

- Akimov, V., Barrio-Hernandez, I., Hansen, S.V.F., Hallenborg, P., Pedersen, A.-K., Bekker-Jensen, D.B., Puglia, M., Christensen, S.D.K., Vanselow, J.T., Nielsen, M.M., et al. (2018). UbiSite approach for comprehensive mapping of lysine and N-terminal ubiquitination sites. *Nat. Struct. Mol. Biol.* 25, 631–640.
- Dey, P. (2018). Flow cytometry: basic principles, procedure and applications in pathology. In *Basic and advanced laboratory techniques in histopathology and cytology*, (Singapore: Springer Singapore), pp. 171–183.
- Jones, M.C., Askari, J.A., Humphries, J.D., and Humphries, M.J. (2018). Cell adhesion is regulated by CDK1 during the cell cycle. *J. Cell Biol.* 217, 3203–3218.
- Monster, J.L., Donker, L., Vliem, M.J., Win, Z., Matthews, H.K., Cheah, J.S., Yamada, S., de Rooij, J., Baum, B., and Glocerich, M. (2021). An asymmetric junctional mechanoresponse coordinates mitotic rounding with epithelial integrity. *J. Cell Biol.* 220.

Theodosiou, M., Widmaier, M., Böttcher, R.T., Rognoni, E., Veelders, M., Bharadwaj, M., Lambacher, A., Austen, K., Müller, D.J., Zent, R., et al. (2016). Kindlin-2 cooperates with talin to activate integrins and induces cell spreading by directly binding paxillin. *Elife* 5, e10130.

Fig. 1 for reviewers. a. Representative confocal images of interphase and mitotic HeLa cells stained for K2 in combination with phalloidin and pH3. Scale bar: 10 μ m. b. Quantification of the endogenous K2 fluorescent signal in interphase and mitotic cells. 10 cells were quantified in each condition. Statistics analyzed by unpaired t-test. **** P < 0.0001.

Fig. 2 for reviewers. a. WB of EGFP-K2, K2 and cyclin B1 in STLC-arrested EGFP-K2-expressing HeLa cells either treated with or without RO3306, or siRNA-depleted for cyclin B1. b. Intensity of the phosphorylation at indicated serine residues of EGFP-K2 in STLC-arrested EGFP-K2-expressing HeLa cells either treated with or without RO3306, or siRNA-depleted for cyclin B1.

Fig. 3 for reviewers. a. Montage of phase contrast recordings of FN-seeded mouse K1/K2 double floxed (K1/K2 dflox) and K1/K2-null (K1/K2 dKO) fibroblasts after RO3306 release. Scale bar: 30 μ m. b. γ H2AX staining of mouse K1/K2 dflox and K1/K2 dKO fibroblasts. Scale bar: left side, 10 μ m; right side, 6 μ m for K1/K2 dflox cells and 3 μ m for K1/K2 dKO cells.

Fig. 4 for reviewers. a. Unsynchronized or RO3306-released HeLa cells expressing EGFP-K2 and treated with or without the Plk1 inhibitor BI2536 for 30 min were lysed and then immunoblotted with anti-K2 antibody. b. Percentages of rounded mouse fibroblasts expressing EGFP-tagged K2- WT, or K2-S180D or K2-S180A quantified after RO3306 release (mean \pm SD; 200 cells counted for each cell line, n = 3 independent experiments).

Position	Localization probabilities	Score	GlyGly (K) Probabilities	Charge
56	0.940337	113.62	LVEKLDVK(0.06)K(0.94)	3
152	0.999893	45.408	K(1)K(1)K(1)LDDQSEDEALELEGPLIMPGSGSIYSS	4
153	0.999893	45.408	K(1)K(1)K(1)LDDQSEDEALELEGPLIMPGSGSIYSS	4
154	0.999893	45.408	K(1)K(1)K(1)LDDQSEDEALELEGPLIMPGSGSIYSS	4
238	0.999998	56.404	MFK(1)PQALLDK	3
275	1	85.377	FK(1)YYSFFDLNPK	2
285	1	84.753	YYSFFDLNPK(1)YDAIR	3
380	0.5	85.288	TSTILGDITSIPELADYIK(0.5)VFK(0.5)	2
383	0.5	85.288	TSTILGDITSIPELADYIK(0.5)VFK(0.5)	2
478	0.999798	48.823	GK(1)TMADSSYNLEVQNILSFLK	2
528	1	42.828	MQHLNPDQPQLIPDQITTDVNPECLVSPRYLK(1)K(5
529	1	42.828	MQHLNPDQPQLIPDQITTDVNPECLVSPRYLK(1)K(5
531	0.49999	57.21	YK(0.5)SK(0.5)QITARILEAHQNVAQMSLIEAK	3
533	0.49999	57.21	YK(0.5)SK(0.5)QITARILEAHQNVAQMSLIEAK	3
555	1	89.468	ILEAHQNVAQMSLIEAK(1)MR	4

Table 1 for reviewers. Ubiquitination sites in K2 immunoprecipitated from STLC- arrested mitotic HeLa cells.

Decision Letter, second revision:

4th February 2022

Dear Dr. Fässler,

Thank you for submitting your revised manuscript "CDK1-cyclin B1-induced kindlin degradation drives focal adhesion disassembly at mitotic entry" (NCB-F46289B). It has now been seen by the original referees and their comments are below. The reviewers find that the paper has improved in revision, and therefore we'll be happy in principle to publish it in Nature Cell Biology, pending minor revisions to satisfy the referees' final requests and to comply with our editorial and formatting guidelines.

As the current version of your manuscript is in a PDF format, please email us a copy of the file in an editable format (Microsoft Word or LaTeX)-- we can not proceed with PDFs at this stage.

Thank you again for your interest in Nature Cell Biology Please do not hesitate to contact me if you have any questions.

Sincerely,
Daryl

Daryl J.V. David, PhD

Senior Editor, Nature Cell Biology
Consulting Editor, Nature Communications
Nature Portfolio

Heidelberger Platz 3, 14197 Berlin, Germany
Email: daryl.david@nature.com
ORCID: <https://orcid.org/0000-0002-9253-4805>

Reviewer #1 (Remarks to the Author):

The authors have done a very good job in answering my comments, I'm happy to recommend this manuscript for publication.

Reviewer #2 (Remarks to the Author):

I have reviewed the revised manuscripts. The authors have satisfactorily addressed my comments raised for the original submission. I support its acceptance to Nature Cell Biology.

Reviewer #3 (Remarks to the Author):

The authors have addressed my concerns and I am happy to recommend publication.

11th February 2022

Dear Dr. Fässler,

Thank you for your patience as we've prepared the guidelines for final submission of your Nature Cell Biology manuscript, "CDK1-cyclin B1-induced kindlin degradation drives focal adhesion disassembly at mitotic entry" (NCB-F46289B). Please carefully follow the step-by-step instructions provided in the attached file, and add a response in each row of the table to indicate the changes that you have made. Please also check and comment on any additional marked-up edits we have proposed within the text. Ensuring that each point is addressed will help to ensure that your revised manuscript can be swiftly handed over to our production team.

We would like to start working on your revised paper, with all of the requested files and forms, as soon as possible (preferably within one week). Please get in contact with us if you anticipate delays.

In recognition of the time and expertise our reviewers provide to Nature Cell Biology's editorial process, we would like to formally acknowledge their contribution to the external peer review of your manuscript entitled "CDK1-cyclin B1-induced kindlin degradation drives focal adhesion disassembly at mitotic entry". For those reviewers who give their assent, we will be publishing their names alongside the published article.

Nature Cell Biology offers a Transparent Peer Review option for new original research manuscripts submitted after December 1st, 2019. As part of this initiative, we encourage our authors to support increased transparency into the peer review process by agreeing to have the reviewer comments, author rebuttal letters, and editorial decision letters published as a Supplementary item. When you

submit your final files please clearly state in your cover letter whether or not you would like to participate in this initiative. Please note that failure to state your preference will result in delays in accepting your manuscript for publication.

Cover suggestions

As you prepare your final files we encourage you to consider whether you have any images or illustrations that may be appropriate for use on the cover of Nature Cell Biology.

Nature Cell Biology has now transitioned to a unified Rights Collection system which will allow our Author Services team to quickly and easily collect the rights and permissions required to publish your work. Approximately 10 days after your paper is formally accepted, you will receive an email in providing you with a link to complete the grant of rights. If your paper is eligible for Open Access, our Author Services team will also be in touch regarding any additional information that may be required to arrange payment for your article.

Please note that *Nature Cell Biology* is a Transformative Journal (TJ). Authors may publish their research with us through the traditional subscription access route or make their paper immediately open access through payment of an article-processing charge (APC). Authors will not be required to make a final decision about access to their article until it has been accepted. Find out more about Transformative Journals

Authors may need to take specific actions to achieve compliance with funder and institutional open access mandates. For submissions from January 2021, if your research is supported by a funder that requires immediate open access (e.g. according to Plan S principles) then you should select the gold OA route, and we will direct you to the compliant route where possible. For authors selecting the subscription publication route our standard licensing terms will need to be accepted, including our self-archiving policies. Those standard licensing terms will supersede any other terms that the author or any third party may assert apply to any version of the manuscript.

For information regarding our different publishing models please see our Transformative

Journals page. If you have any questions about costs, Open Access requirements, or our legal forms, please contact ASJournals@springernature.com.

[REDACTED]

Best regards,

Nyx Hills
Staff
Nature Cell Biology

On behalf of

Daryl J.V. David, PhD

Senior Editor, Nature Cell Biology
Consulting Editor, Nature Communications
Nature Portfolio

Heidelberger Platz 3, 14197 Berlin, Germany
Email: daryl.david@nature.com
ORCID: <https://orcid.org/0000-0002-9253-4805>

Reviewer #1:

Remarks to the Author:

The authors have done a very good job in answering my comments, I'm happy to recommend this manuscript for publication.

Reviewer #2:

Remarks to the Author:

I have reviewed the revised manuscripts. The authors have satisfactorily addressed my comments raised for the original submission. I support its acceptance to Nature Cell Biology.

Reviewer #3:

Remarks to the Author:

The authors have addressed my concerns and I am happy to recommend publication.

Final Decision Letter:

Dear Dr Fässler,

I am pleased to inform you that your manuscript, "CDK1-cyclin B1-induced kindlin degradation drives focal adhesion disassembly at mitotic entry", has now been accepted for publication in Nature Cell Biology.

Please note that *Nature Cell Biology* is a Transformative Journal (TJ). Authors may publish their research with us through the traditional subscription access route or make their paper immediately open access through payment of an article-processing charge (APC). Authors will not be required to make a final decision about access to their article until it has been accepted. Find out more about Transformative Journals

If your paper includes color figures, please be aware that in order to help cover some of the additional cost of four-color reproduction, Nature Research charges our authors a fee for the printing of their color figures. Please contact our offices for exact pricing and details.

If you have not already done so, we strongly recommend that you upload the step-by-step protocols used in this manuscript to the Protocol Exchange (www.nature.com/protocolexchange), an open online resource established by Nature Protocols that allows researchers to share their detailed experimental know-how. All uploaded protocols are made freely available, assigned DOIs for ease of citation and are fully searchable through nature.com. Protocols and the Nature and Nature research journal papers in which they are used can be linked to one another, and this link is clearly and prominently visible in the online versions of both papers. Authors who performed the specific experiments can act as primary authors for the Protocol as they will be best placed to share the methodology details, but the Corresponding Author of the present research paper should be included as one of the authors. By uploading your Protocols to Protocol Exchange, you are enabling researchers to more readily reproduce or adapt the methodology you use, as well as increasing the visibility of your protocols and papers. You can also establish a dedicated page to collect your lab Protocols. Further information can be found at www.nature.com/protocolexchange/about

You can use a single sign-on for all your accounts, view the status of all your manuscript submissions and reviews, access usage statistics for your published articles and download a record of your refereeing activity for the Nature journals.

With kind regards,
Daryl

Daryl J.V. David, PhD

Senior Editor, Nature Cell Biology
Consulting Editor, Nature Communications

Nature Portfolio

Heidelberger Platz 3, 14197 Berlin, Germany
Email: daryl.david@nature.com
ORCID: <https://orcid.org/0000-0002-9253-4805>

** Visit the Springer Nature Editorial and Publishing website at www.springernature.com/editorial-and-publishing-jobs for more information about our career opportunities. If you have any questions please click here.**